# Budgeted Active Experimentation for Treatment Effect Estimation from Observational and Randomized Data

Jiacan Gao [*4]  Xinyan Su [*3]  Mingyuan Ma [5]  Yiyan Huang [6]  Xiao Xu [3]  Xinrui Wan [3]  Tianqi Gu [3]  Enyun Yu [3]
Jiecheng Guo [3]  Zhiheng Zhang[✉][1 2]

## Abstract

Estimating heterogeneous treatment effects is central to data-driven decision-making, yet industrial applications often face a fundamental tension between limited randomized controlled trial (RCT) budgets and abundant but biased observational data (OBS) collected under historical targeting policies. Although observational logs offer the advantage of scale, they may suffer from severe policy-induced imbalance and overlap violations, rendering standalone estimation unreliable. We propose a *budgeted active experimentation* framework that iteratively collects informative randomized samples for causal effect estimation via active sampling. By leveraging observational signals, we develop an acquisition function targeting uplift estimation uncertainty, domain discrepancy, and overlap deficits to select the most informative units for randomized experiments. We establish finite-sample deviation bounds, asymptotic normality via martingale CLTs, and minimax lower bounds showing near-optimality in the linear representation setting. Experiments on synthetic datasets support our theoretical findings, and further extensions to industrial neural network-based uplift modeling scenarios show that active sampling can improve sample efficiency over random sampling under limited RCT budgets.

---

[*]Equal contribution in alphabetical order. [1]School of Statistics and Data Science, Shanghai University of Finance and Economics, Shanghai, China [2]Institute of Big Data Research, Shanghai University of Finance and Economics, Shanghai, China [3]Didi Chuxing, Beijing, China [4]School of Statistics, East China Normal University, Shanghai, China [5]School of Mathematics and Statistics, Beijing Jiaotong University, Beijing, China [6]School of Computing and Information Technology, Great Bay University, Guangdong, China. Correspondence to: Zhiheng Zhang <zhangzhiheng@mail.shufe.edu.cn>.

*Proceedings of the 43rd International Conference on Machine Learning*, Seoul, South Korea. PMLR 306, 2026. Copyright 2026 by the author(s).

## 1. Introduction

Estimating heterogeneous treatment effects (HTE), such as the conditional average treatment effect (CATE), is a central problem in causal inference and data-driven decision making. Accurate HTE estimation underpins personalized marketing, recommendation systems, clinical decision support, and public policy design, where decisions must be tailored to individual characteristics rather than population averages. In large-scale real-world systems, however, HTE estimation is constrained by a fundamental asymmetry in data sources: randomized controlled trials (RCTs) provide unbiased and reliable causal signals, but are expensive, slow to deploy, and severely limited in sample size; observational data (OBS), in contrast, are abundant and high-dimensional, but are generated under historical targeting policies that induce selection bias, policy-driven imbalance, and violations of overlap (Hatt et al., 2022; Colnet et al., 2024).

This asymmetry has motivated a growing literature on learning treatment effects from multiple data sources. Classical work focuses on *causal identification* from observational data under strong assumptions such as ignorability and positivity, while more recent approaches seek to *combine* OBS and RCT data through reweighting or doubly robust estimators (Cheng & Cai, 2021). In industrial settings, however, these approaches face a critical limitation: historical policies are often near-deterministic, creating regions of the covariate space where observational data provide essentially no counterfactual information. In such regimes, directly using OBS outcomes for identification is potentially unreliable, while running large-scale uniform RCTs is prohibitively costly.

This tension naturally shifts the focus from estimation to *experiment design*. Rather than asking how to extract causal effects from biased observational logs, a more operational question is: *given abundant observational data and a strict budget for randomized experiments, where should one run RCTs to most efficiently learn heterogeneous treatment effects?* This question lies at the intersection of causal inference and active learning, but differs fundamentally from classical active learning: here, querying a unit does not merely reveal a label, but requires actively assigning a treat-

ment and observing a causal outcome.

Existing active learning approaches have explored this problem from two largely separate perspectives. One line of work studies active or budgeted sampling *within experimental settings* (Zhang et al., 2025; Kato et al., 2024; Ghadiri et al., 2023), where treatments can be freely assigned but the RCT sample size is limited. Related work on sequential experimental design for A/B testing further studies how to adaptively allocate treatments to improve estimation efficiency under finite-sample imbalance and model misspecification (Wen et al., 2026). Another line focuses on active learning *within observational studies* (Wen et al., 2025; Gao et al., 2025), where treatment assignments are fixed and the budget corresponds to labeling or outcome acquisition. In contrast, the practically relevant regime where *observational data and experimental design coexist*— abundant biased logs alongside a small, adaptively collected RCT— has received comparatively little theoretical and methodological attention. In particular, it remains unclear how observational data should guide adaptive experiment design while retaining finite-sample validity and statistical inference guarantees.

In this paper, we propose a *budgeted active experimentation framework* that addresses this gap by cleanly separating the roles of observational and randomized data. The core idea is that *observational data inform experiment design, while randomized experiments provide causal estimation*. Specifically, observational logs are used to learn a shared representation, to diagnose overlap deficits induced by historical policies, and to identify covariate regions under-represented in the current experimental sample. These signals are combined into a multi-criteria acquisition function that actively selects which units to enroll into RCTs. The final estimator, however, is identified purely by randomized experiments, ensuring robustness to arbitrary confounding in the observational source.

This design induces a nontrivial statistical regime: the experimental data are adaptively collected, non-i.i.d., and depend on past outcomes. We show that, despite adaptivity, randomization induces a martingale structure that protects unbiasedness, while active sampling shapes the information matrix and governs statistical efficiency. Perhaps counterintuitively, we prove that even the most aggressive active experimentation strategy cannot beat a $\sqrt{d/B}$ rate in general, where $d$ is the effective dimension of heterogeneity and $B$ is the RCT budget. Active sampling cannot improve the minimax rate for this class, but it can improve finite-sample constants through better conditioning of the experimental design. Our main contributions are summarized as follows:

**(i)** We propose a principled framework for budgeted active experimentation that integrates observational and randomized data by separating experiment design from causal

identification.

**(ii)** We provide theoretical guarantees for adaptively collected RCT data, including finite-sample deviation bounds, asymptotic normality via martingale CLTs, and minimax lower bounds establishing near-optimality.

**(iii)** We clarify the precise role of observational data in causal learning: OBS improves *where* to randomize, not *what* is identified, yielding a robust and interpretable path to cost-efficient HTE estimation.

The remainder of the paper is organized as follows. Section A reviews related work on active learning and treatment effect estimation. Section 2 formalizes the two-source causal learning problem. Section 3 presents the proposed active experimentation algorithm. Section 4 develops the theoretical guarantees under adaptive, non-i.i.d. experimentation. Section 5 reports empirical results on large-scale real-world data. Section 6 concludes with implications and future directions.

**Conflict of Interest Disclosure**. This work is a collaborative research outcome between DiDi Chuxing and Shanghai University of Finance and Economics, and the use of DiDi Chuxing's industrial data was reviewed and approved through DiDi Chuxing's internal data governance and compliance process.

## 2. Problem Formulation

**Notation.** Uppercase letters denote random variables and lowercase denote their realizations. Let $X \in \mathcal{X} \subseteq \mathbb{R}^d$ be covariates, $T \in \{0, 1\}$ a binary treatment, and $Y \in \{0, 1\}$ a binary outcome (the formulation extends to any bounded $Y \in [0, 1]$). For any distribution $\mathcal{Q}$, we write $\mathbb{E}_{\mathcal{Q}}[\cdot]$ and $\mathbb{P}_{\mathcal{Q}}(\cdot)$ for expectation and probability under $\mathcal{Q}$. We adopt the Neyman–Rubin potential outcomes framework (Rubin, 2005). Each unit has potential outcomes $Y(1)$ and $Y(0)$ under treatment and control, respectively. The observed outcome follows $Y = Y(T)$.

**Assumption 2.1.** (SUTVA). (i) *No interference*: a unit's potential outcomes are unaffected by other units' assignments. (ii) *Consistency*: $Y = T\,Y(1) + (1 - T)\,Y(0)$.

For the *target estimand and evaluation metric*, our estimand is the **conditional average treatment effect (CATE)** (Shalit et al., 2017; Zhong et al., 2022; Gao et al., 2025):

$$\tau(x) = \mathbb{E}[\,Y(1) - Y(0)\,|\,X = x\,], \quad x \in \mathcal{X}. \quad (1)$$

Define $\mu_t(x) \triangleq \mathbb{E}[Y(t) \mid X = x]$ for $t \in \{0, 1\}$, so that $\tau(x) = \mu_1(x) - \mu_0(x)$. We evaluate generalization over a *target population* whose covariate marginal distribution is $\mathbb{P}_X$. Given an estimator $\hat{\tau}$, we evaluate its performance using the *Precision in Estimation of Heterogeneous Effect* (PEHE) risk (Hill, 2011; Wen et al., 2025; Gao et al., 2025).

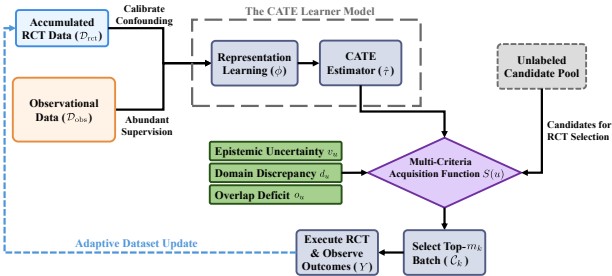

*Figure 1.* **Overview of the proposed Budgeted Active Experimentation framework for OBS-RCT fusion.** We first train a CATE learner using abundant observational logs, then iteratively select a batch of units from an unlabeled candidate pool $\mathcal{D}_{\text{pool}}$ for randomized experiments under a fixed budget. At each iteration, a multi-criteria acquisition function $S(u)$ scores and ranks candidates by balancing three signals: **epistemic uncertainty** ($v_u$), **domain discrepancy** ($d_u$), and **overlap deficit** ($o_u$). We select the top-ranked units to run RCTs, add the new outcomes to the labeled set, and update the learner to reduce CATE estimation error.

In particular, we report its square root, defined as:

$$\mathcal{R}(\hat{\tau}) \triangleq \sqrt{\mathbb{E}_{X \sim \mathbb{P}_X}[(\hat{\tau}(X) - \tau(X))^2]}. \qquad (2)$$

We have access to two-source data:

(i) **Observational log (OBS).** An observational dataset $\mathcal{D}_{\text{obs}} = \{(X_i^{\text{obs}}, T_i^{\text{obs}}, Y_i^{\text{obs}})\}_{i=1}^{n_{\text{obs}}} \overset{\text{i.i.d.}}{\sim} \mathbb{P}_{\text{obs}}$, collected under a historical (possibly non-random) assignment policy. Let the induced observational propensity be $e_{\text{obs}}(x) \triangleq \mathbb{P}_{\text{obs}}(T^{\text{obs}} = 1 \mid X^{\text{obs}} = x)$, which may be highly imbalanced and even (nearly) deterministic in some regions. We write $\mathbb{P}_{\text{obs}}$ as shorthand for $\mathbb{P}_{\mathbb{P}_{\text{obs}}}$.

(ii) **Unlabeled candidate pool (RCT-POOL).** An unlabeled pool $\mathcal{D}_{\text{pool}} = \{X_j^{\text{pool}}\}_{j=1}^{n_{\text{pool}}} \overset{\text{i.i.d.}}{\sim} \mathbb{P}_X$, from which we may *query* a small subset of units to run randomized experiments. Units in $\mathcal{D}_{\text{pool}}$ receive no treatment unless queried.

Because OBS treatments follow a historical policy, global positivity in $\mathcal{D}_{\text{obs}}$ (i.e., $0 < e_{\text{obs}}(x) < 1$ for all $x$) may fail. We therefore *do not* assume global overlap for OBS; instead, we will use randomized experiments on queried units to obtain counterfactual information in weak(non)-overlap regions.

Moreover, we view $\mathcal{D}_{\text{pool}} = \{X_j^{\text{pool}}\}_{j=1}^{n_{\text{pool}}}$ as an unlabeled sample from the *target* covariate marginal $\mathbb{P}_X$, whereas $\mathcal{D}_{\text{obs}} = \{(X_i^{\text{obs}}, T_i^{\text{obs}}, Y_i^{\text{obs}})\}_{i=1}^{n_{\text{obs}}}$ is drawn from a different joint law $\mathbb{P}_{\text{obs}}$ induced by a historical (possibly near-deterministic) targeting policy. Accordingly, the covariate marginals need not match: in general $\mathbb{P}_{X^{\text{obs}}} \neq \mathbb{P}_X$ and $\text{supp}(\mathbb{P}_{X^{\text{obs}}}) \subseteq \text{supp}(\mathbb{P}_X)$, reflecting selection bias and potential overlap violations in the OBS log. Our design therefore treats $\mathcal{D}_{\text{pool}}$ as the population whose risk is evaluated, while using $\mathcal{D}_{\text{obs}}$ only to guide where to randomize (and not for causal identification in weak-overlap regions).

**Active Experimentation Protocol (Adaptive RCT).** We sequentially run experiments for $K$ rounds (or until the budget is exhausted). Let $\mathcal{D}_{\text{rct}}^{(0)} = \emptyset$ initially. At round $k$, an adaptive strategy selects an index set $\mathcal{C}_k \subseteq [n_{\text{pool}}]$ from the remaining pool. For each selected unit with covariate $X$, we assign treatment $T^{\text{rct}} \sim \text{Bern}(p_k(X))$, $p_k : \mathcal{X} \to [f_{\min}, f_{\max}]$, where the randomization probability $p_k(\cdot)$ is *known* and satisfies $0 < f_{\min} \leq f_{\max} < 1$. We then observe $Y^{\text{rct}}$ and record the quadruple $(X, T^{\text{rct}}, Y^{\text{rct}}, p_k(X))$. The experimental dataset is updated as:

$$\mathcal{D}_{\text{rct}}^{(k)} \leftarrow \mathcal{D}_{\text{rct}}^{(k-1)} \cup \{(X_j^{\text{pool}}, T_j^{\text{rct}}, Y_j^{\text{rct}}, p_k(X_j^{\text{pool}})) : j \in \mathcal{C}_k\}. \qquad (3)$$

Since $\mathcal{C}_k$ and $p_k(\cdot)$ may depend on the history, $\mathcal{D}_{\text{rct}}^{(k)}$ is adaptively collected and thus non-i.i.d.

**Assumption 2.2.** (Randomization and positivity in RCT). Conditioned on the history up to round $k$ and covariates $X$, the RCT assignment is randomized:

$$T^{\text{rct}} \perp (Y(0), Y(1)) \mid (X, \text{history}),$$
$$\mathbb{P}(T^{\text{rct}} = 1 \mid X, \text{history}) = p_k(X) \in [f_{\min}, f_{\max}]. \qquad (4)$$

**Assumption 2.3.** (Outcome invariance across sources). The conditional potential outcome distributions are shared across sources: for each $t \in \{0, 1\}$ and $x \in \mathcal{X}$,

$$\mathbb{P}_{\text{obs}}(Y(t) \mid X = x) = \mathbb{P}_X(Y(t) \mid X = x). \qquad (5)$$

Equivalently, the target CATE $\tau(x)$ is common to both sources.

**Assumption 2.4.** (Ignorability for OBS; used when leveraging OBS causally). When we use OBS for causal identification (e.g., via propensity-based corrections), we assume *conditional ignorability* under the OBS distribution $\mathbb{P}_{\text{obs}}$:

$$(Y(0), Y(1)) \perp T^{\text{obs}} \mid X^{\text{obs}}. \qquad (6)$$

**Objective.** Let $\pi$ denote an *adaptive experiment design* that generates the sequence $\{(\mathcal{C}_k, p_k)\}_{k \geq 1}$ based on the observed history. Given a total experimental budget $B$ (the number of queried units), we aim to design $\pi$ and a corresponding estimator $\hat{\tau}_\pi$ such that the target risk (2) is minimized. The estimator $\hat{\tau}_\pi$ is defined as the RCT-identified CATE estimator obtained from the adaptively collected randomized dataset $\mathcal{D}_{\text{rct}}^{(K)}$ under the design policy $\pi$:

$$\hat{\tau}_\pi = \mathcal{A}_{\text{est}}\left(\mathcal{D}_{\text{rct}}^{(K)}; \phi\right), \qquad (7)$$

where $\phi$ is a representation or design feature map that may be learned from pre-randomization data, including $\mathcal{D}_{\text{obs}}$ and the unlabeled candidate pool. In this core formulation, $\mathcal{D}_{\text{obs}}$ affects the design policy $\pi$ and the representation $\phi$, but

does not directly enter the identifying estimating equations for $\hat{\tau}_\pi$. The objective is therefore

$$\min_\pi \ \mathcal{R}(\hat{\tau}_\pi) \quad \text{s.t.} \quad \sum_{k=1}^{K} |\mathcal{C}_k| \leq B. \tag{8}$$

Equivalently, one may consider the sample complexity $N_{\text{RCT}}(\varepsilon)$: the minimum budget required to achieve $\mathcal{R}(\hat{\tau}_\pi) \leq \varepsilon$ with high probability.

Assumptions 2.1–2.4 are standard in causal learning but play distinct roles here: (i) SUTVA ensures a well-defined unit-level causal model; (ii) Assumption 2.2 guarantees unbiased identification from queried RCT samples and prevents extreme-variance estimators via $[f_{\min}, f_{\max}]$; (iii) Assumption 2.3 enables pooling information across sources (diagnosable by comparing covariate distributions and outcome model residuals across domains); (iv) Assumption 2.4 is only needed if OBS is used for causal correction—if it is questionable, OBS can be used more conservatively (e.g., for representation / warm-starting), while identification and calibration rely primarily on the randomized samples.

## 3. Methodology

We study a two-source setting where (i) a large observational log $\mathcal{D}_{\text{obs}}$ provides abundant but potentially biased and weak-overlap supervision, and (ii) an unlabeled pool $\mathcal{D}_{\text{pool}}$ represents the target population from which we may enroll a small number of units into randomized experiments. Our objective is to leverage a limited experimental budget $B$ to acquire an adaptively designed RCT dataset $\mathcal{D}_{\text{rct}}$, and then learn a final CATE estimator $\hat{\tau}_\pi$.

The key difficulty is *where* to run RCTs. Running experiments uniformly at random wastes budget on regions where OBS already provides reliable information, while failing to repair regions where OBS is systematically uninformative (e.g., extreme propensities) or where the learned model extrapolates. We therefore cast experiment design as a *pool-based active learning* problem for causal effect estimation: each queried unit provides *both* a randomized treatment assignment and an outcome label, which can directly reduce CATE error.

Algorithm 1 summarizes our procedure. Below we explain the intuition behind each module, focusing on three canonical sources of CATE estimation error that are particularly severe in OBS–RCT fusion:
(i) **Epistemic uncertainty** of the CATE model due to finite data and high-dimensional covariates;
(ii) **Domain discrepancy** between the biased OBS log and the actively collected RCT sample (and thus the target pool);
(iii) **Overlap deficit** in OBS, where near-deterministic historical targeting makes counterfactual information essentially absent.

---

**Algorithm 1** Active Sampling for OBS-RCT Fusion

---

**Require:** Observational Data $\mathcal{D}_{\text{obs}}$, Unlabeled Pool $\mathcal{D}_{\text{pool}}$, Max Query Batch Size $M$, Budget $B$, Max Rounds $K$.
**Ensure:** Final CATE Estimator $\hat{\tau}_\pi$.
 1: **Initialize:** $\mathcal{D}_{\text{rct}} \leftarrow \emptyset$, $k \leftarrow 1$.
 2: **Representation Learning:** Train feature encoder $\phi : \mathcal{X} \rightarrow \mathcal{H}$ on $\mathcal{D}_{\text{obs}} \cup \mathcal{D}_{\text{pool}}$.
 3: **while** $|\mathcal{D}_{\text{rct}}| < B$ and $k \leq K$ **do**
 4: $\quad m_k \leftarrow \min(M, B - |\mathcal{D}_{\text{rct}}|)$
 5: $\quad$ **1. Update Scoring Components:**
 6: $\quad\quad$ Train domain classifier $g_\xi(\phi(x))$ to discriminate $\mathcal{D}_{\text{pool}}$ (label 1) from $\mathcal{D}_{\text{obs}} \cup \mathcal{D}_{\text{rct}}$ (label 0).
 7: $\quad\quad$ Estimate propensity $\hat{e}_{\text{obs}}(\phi(x)) \approx \mathbb{P}_{\text{obs}}(T^{\text{obs}} = 1 \mid \phi(x))$
 8: $\quad$ **2. Acquisition Scoring:**
 9: $\quad$ **for** candidate $u \in \mathcal{D}_{\text{pool}}$ **do**
10: $\quad\quad v_u \leftarrow \text{Uncertainty}(\phi(u))$ {CATE disagreement or ridge leverage}
11: $\quad\quad d_u \leftarrow \sigma(g_\xi(\phi(u)))$ $\quad$ {$\sigma(\cdot)$: Sigmoid Function}
12: $\quad\quad o_u \leftarrow 2 \cdot |\hat{e}_{\text{obs}}(\phi(u)) - 0.5|$
13: $\quad\quad$ Compute Score: $\quad\quad$ {$\eta(\cdot)$: Rank Function}
14: $\quad\quad S(u) \leftarrow \alpha \cdot \eta(v_u) + \beta \cdot \eta(d_u) + \gamma \cdot \eta(o_u)$
15: $\quad$ **end for**
16: $\quad$ **3. Selection & Experimentation:**
17: $\quad\quad$ Select candidates $\mathcal{C}_k \leftarrow \text{Top-}m_k(S(\cdot), \mathcal{D}_{\text{pool}})$.
18: $\quad$ **for** $X_i \in \mathcal{C}_k$ **do**
19: $\quad\quad$ Perform RCT: Assign $T_i^{\text{rct}} \sim \text{Bern}(p_k(X_i))$, observe $Y_i^{\text{rct}}$.
20: $\quad\quad \mathcal{D}_{\text{rct}} \leftarrow \mathcal{D}_{\text{rct}} \cup \{(X_i, T_i^{\text{rct}}, Y_i^{\text{rct}}, p_k(X_i))\}$.
21: $\quad$ **end for**
22: $\quad$ **4. Update:** $\mathcal{D}_{\text{pool}} \leftarrow \mathcal{D}_{\text{pool}} \setminus \mathcal{C}_k$; $\quad k \leftarrow k + 1$.
23: **end while**
24: **Return** $\hat{\tau}_\pi = \mathcal{A}_{\text{est}}(\mathcal{D}_{\text{rct}}; \phi)$.

---

Each error source is addressed by an explicit design choice: covariate shift is handled by querying from the unlabeled pool $\mathcal{D}_{\text{pool}}$ to restore coverage of the target marginal $\mathbb{P}_X$; weak or missing overlap is mitigated by targeted randomization rather than reweighting observational outcomes; and adaptivity bias is controlled by design-based randomization guarantees. Here, *uncertainty* refers not only to estimation variance conditional on a fixed design, but also to design-induced uncertainty arising from which regions of the covariate space receive randomized evidence. Our active strategy reduces this uncertainty by allocating experimental budget to covariate regions with the largest marginal contribution to CATE risk, yielding tighter error bounds for a fixed RCT cost. As a result, uncertainty is actively shaped—and provably reduced—by the experimental design, rather than passively inherited from OBS–RCT fusion. Our acquisition score explicitly combines proxies for these three error sources, so that each RCT query yields maximal marginal value toward reducing the target PEHE risk.

To mitigate instability in high-dimensional industrial settings, we employ a shared Multilayer Perceptron (MLP) encoder, defined as $\phi : \mathcal{X} \rightarrow \mathcal{H}$, to map raw covariates into a dense latent representation. This shared backbone serves as a common foundation for (i) propensity estimation, (ii) CATE modeling, and (iii) domain discrimination, so that all components operate in a unified lower-dimensional space.

$v_u$: **CATE learner disagreement for epistemic uncertainty.** Since a single CATE learner may provide over-confident estimates in regions with limited evidence, we quantify epistemic uncertainty through the disagreement among a set of plausible CATE learners. For neural models, this can be approximated by ensemble or MC-dropout variance ($E$: # of stochastic passes):

$$v_u = \mathrm{Var}\left(\{\hat{\tau}_j(\phi(u))\}_{j=1}^{E}\right). \qquad (9)$$

For the linear pseudo-outcome ridge estimator, the same uncertainty has a closed-form leverage-score proxy. At round $k$, we use

$$v_u = \phi(u)^\top \left(V_\lambda^{(k-1)}\right)^{-1} \phi(u),$$
$$V_\lambda^{(k-1)} = \lambda I_d + \sum_{i \in \mathcal{D}_{\mathrm{rct}}^{(k-1)}} \phi(X_i)\phi(X_i)^\top. \qquad (10)$$

A large $v_u$ indicates limited evidence around $u$, either through disagreement among plausible CATE learners or, in the linear ridge case, through poor coverage in the current RCT design. Querying $u$ is therefore expected to reduce epistemic uncertainty.

$d_u$: **Domain discrimination for representativeness and distributional alignment.** To prevent the active set from drifting solely towards decision boundaries and to mitigate the selection bias inherent in $\mathcal{D}_{\mathrm{obs}}$, we explicitly enforce distributional alignment. We employ a domain classifier $g_\xi$ operating on the representation space, trained to distinguish the target pool $\mathcal{D}_{\mathrm{pool}}$ from the current training set $\mathcal{D}_{\mathrm{current}} = \mathcal{D}_{\mathrm{obs}} \cup \mathcal{D}_{\mathrm{rct}}$. The score is defined as the predicted probability of belonging to the target pool (label 1):

$$d_u \triangleq \mathbb{P}(\mathrm{domain} = 1 \mid \phi(u); \xi) = \sigma(g_\xi(\phi(u))) \qquad (11)$$

where $g_\xi(\cdot)$ denotes the logit output and $\sigma(\cdot)$ is the sigmoid function. Theoretically, $g_\xi$ acts as a density-ratio estimator where the logit approximates $\log(p_{\mathrm{pool}}(\phi)/p_{\mathrm{current}}(\phi))$. Consequently, prioritizing high $d_u$ targets samples in regions under-represented by the current source data, minimizing covariate shift and ensuring robust generalization.

$o_u$: **Propensity-based overlap deficit from OBS.** To mitigate estimator instability caused by extreme observational propensities (positivity violations), we explicitly target regions lacking common support. We utilize a propensity

model $\hat{e}_{\mathrm{obs}}(\phi(u)) \approx \mathbb{P}_{\mathrm{obs}}(T^{\mathrm{obs}} = 1 \mid \phi(u))$, pre-trained on $\mathcal{D}_{\mathrm{obs}}$, to define the deficit score:

$$o_u \triangleq 2 \cdot |\hat{e}_{\mathrm{obs}}(\phi(u)) - 0.5|. \qquad (12)$$

Samples with high $o_u$ correspond to regions where the historical policy was deterministic. Prioritizing these instances for RCT labeling effectively "repairs" the overlap deficit by injecting counterfactual information exactly where the observational signal is weakest.

**Rank-Normalized Multi-Criteria Acquisition Function.** The three signals $(v_u, d_u, o_u)$ can have very different scales and may change over rounds. Instead of brittle scale-dependent normalization, we use a simple *rank-based* map $\eta(\cdot)$, computed over the current pool:

$$\eta(a_u) \triangleq \frac{1}{|\mathcal{D}_{\mathrm{pool}}|} \sum_{u' \in \mathcal{D}_{\mathrm{pool}}} \mathbb{I}(\psi(a_{u'}) \leq \psi(a_u)) \in [0,1],$$
$$(13)$$

where $\psi(\cdot)$ is a monotone transform used for ranking. Since the constructed signals $v_u$, $d_u$, and $o_u$ are all non-negative and positively correlated with the informativeness of a sample, we simply use identity transform $\psi(a) = a$ for ranking. For each pool candidate $u$, we compute acquisition score:

$$S(u) \triangleq \alpha \cdot \eta(v_u) + \beta \cdot \eta(d_u) + \gamma \cdot \eta(o_u), \qquad (14)$$

and select the top-$m_k$ candidates. $\alpha, \beta$, and $\gamma$ balance the trade-off between uncertainty , discrepancy, and counterfactual coverage. To ensure scale consistency across these heterogeneous metrics, we apply rank normalization prior to weighted aggregation.

### 3.1. Adaptive RCT execution and dataset update

After selecting $\mathcal{C}_k$, we conduct randomized experiments for each $X_i \in \mathcal{C}_k$. We allow a covariate-dependent randomization policy $f_k(\cdot)$ (e.g., to incorporate operational constraints), and enforce positivity by clipping:

$$p_i \triangleq \mathrm{Clip}(f_k(X_i), f_{\min}, f_{\max}), \qquad (15)$$

where $\mathrm{Clip}(z, a, b) \triangleq \min\{\max\{z, a\}, b\}$. We then draw $T_i \sim \mathrm{Bern}(p_i)$, observe $Y_i$, and update as in (3). Storing $p_i$ is essential for downstream learning/inference when randomization probabilities are not constant. Finally, we remove $\mathcal{C}_k$ from $\mathcal{D}_{\mathrm{pool}}$ and repeat until the budget is exhausted. We discuss the optimal choice of $p$ in Proposition B.15.

## 4. Theoretical Analysis

This section gives theoretical guarantees for the *estimation component* of Algorithm 1 under adaptively collected RCT data. Our goal is to state three results precisely:

(i) **Finite-sample validity**: a nearly unbiased effect estimator, together with a non-asymptotic error bound that remains valid under *adaptive, non-i.i.d.* RCT sampling.

(ii) **Asymptotic inference**: a central limit theorem (CLT) that establishes asymptotic normality and supports confidence intervals.

(iii) **Fundamental limits**: a minimax lower bound showing that the achieved rate is information-theoretically optimal up to logarithmic factors. The central observation is that randomization protects unbiasedness, whereas active selection shapes the information matrix and therefore the error rate. We first formalize an analyzable estimator that follows the RCT protocol in Algorithm 1 and is standard in modern causal ML: it regresses a *pseudo-outcome* onto a representation.

**A tractable estimator for adaptively collected RCT samples** Let $B$ denote the total RCT budget and index queried units in chronological order $t = 1, \ldots, B$. Write the $t$-th queried sample as $(X_t, T_t, Y_t, p_t)$ where $p_t \in [f_{\min}, f_{\max}]$ is the (known) assignment probability used when unit $t$ was experimented. Let $\{\mathcal{F}_t\}_{t \geq 0}$ be the filtration where $\mathcal{F}_t$ contains $\mathcal{D}_{\mathrm{obs}}$, the full covariate pool $\mathcal{D}_{\mathrm{pool}}$, and all RCT data up to time $t$. The adaptive querying policy, through $\mathcal{C}_k$ and $f_k$, implies that both $X_t$ and $p_t$ may depend on the past filtration $\mathcal{F}_{t-1}$. For each queried unit, we define the RCT pseudo-outcome as

$$\widetilde{Y}_t \triangleq \frac{T_t Y_t}{p_t} - \frac{(1 - T_t)Y_t}{1 - p_t}. \tag{16}$$

Intuitively, $\widetilde{Y}_t$ converts a single randomized observation into a noisy but unbiased label for the treatment effect at $X_t$. Let $\phi(x) \in \mathbb{R}^d$ be a fixed feature map (e.g., the learned embedding $\phi$ in Algorithm 1). For theory, we analyze a realizable linear CATE model in this representation: $\tau(x) = \langle \theta_\star, \phi(x) \rangle$, $\theta_\star \in \mathbb{R}^d$.

This linear specification isolates the statistical effect of *adaptive sampling under limited budgets*. (When $\tau(\cdot)$ is not exactly linear, the bounds below become bounds on the best linear approximation plus an approximation error term.) Given $\{(X_t, T_t, Y_t, p_t)\}_{t=1}^B$, we estimate $\theta_\star$ by (optionally regularized) least squares on pseudo-outcomes:

$$\widehat{\theta}_\lambda \triangleq \arg\min_{\theta \in \mathbb{R}^d} \sum_{t=1}^B \left( \widetilde{Y}_t - \langle \theta, \phi(X_t) \rangle \right)^2 + \lambda \|\theta\|_2^2, \tag{17}$$

and output $\widehat{\tau}_\lambda(x) \triangleq \langle \widehat{\theta}_\lambda, \phi(x) \rangle$.

Algorithm 1 specifies *how the $X_t$'s are chosen* (active sampling) and *how $p_t$ is assigned* (bounded randomization). Algorithm 2 specifies a transparent estimator for analyzing the statistical consequences of that adaptive design. In practice, one may replace the linear regressor by a neural CATE learner; the key objects in the proofs below are (i) the unbiased pseudo-outcome, and (ii) an information matrix that quantifies how well the queried points cover the representation space.

---

**Algorithm 2** Orthogonalized Linear CATE Estimation on Adaptive RCT Data

---

1: **Input:** Adaptive RCT data $\{(X_t, T_t, Y_t, p_t)\}_{t=1}^B$, feature map $\phi(\cdot)$, ridge $\lambda \geq 0$.
2: **Output:** CATE estimator $\widehat{\tau}_\lambda(\cdot)$.
3: **for** $t = 1, \ldots, B$ **do**
4:   Compute pseudo-outcome $\widetilde{Y}_t \leftarrow \frac{T_t Y_t}{p_t} - \frac{(1 - T_t)Y_t}{1 - p_t}$.
5: **end for**
6: Form $V_\lambda \leftarrow \lambda I_d + \sum_{t=1}^B \phi(X_t)\phi(X_t)^\top$ and $b \leftarrow \sum_{t=1}^B \phi(X_t)\widetilde{Y}_t$.
7: Solve $\widehat{\theta}_\lambda \leftarrow V_\lambda^{-1} b$.
8: Return $\widehat{\tau}_\lambda(x) \leftarrow \langle \widehat{\theta}_\lambda, \phi(x) \rangle$.

---

**Role of observational data in the estimator.** It is important to clarify that the causal identification of $\tau(x)$ is carried exclusively by randomized experiments. Observational data are *not* directly used in the estimating equations. Instead, $\mathcal{D}_{\mathrm{obs}}$ enters the procedure in two structural ways: (i) it defines the representation $\phi(\cdot)$ in which the CATE is approximated, and (ii) it shapes the adaptive experiment design that determines the queried covariates $\{X_t\}_{t=1}^B$. All theoretical guarantees are therefore *design-adaptive but identification-robust*: they remain valid even when the observational assignment mechanism is arbitrarily confounded.

### 4.1. Unbiasedness and finite-sample error bounds

We now state the first main result: despite *adaptive*, non-i.i.d. selection of $X_t$, the RCT pseudo-outcome yields a martingale estimating equation. This leads to a finite-sample self-normalized deviation bound for the ridge estimator with $\lambda > 0$, and recovers conditional unbiasedness of the OLS estimator when $\lambda = 0$ and the queried design is fixed in advance or generated independently of the RCT outcomes.

**Assumption 4.1.** (Predictable adaptive design). For each $t$, the queried covariate $X_t$ and assignment probability $p_t$ are $\mathcal{F}_{t-1}$-measurable.

**Assumption 4.2.** (RCT randomization and boundedness). Conditioned on $(X_t, p_t, \mathcal{F}_{t-1})$, treatment is randomized as $T_t \sim \mathrm{Bern}(p_t)$ and $T_t \perp (Y_t(0), Y_t(1)) \mid (X_t, p_t, \mathcal{F}_{t-1})$. Outcomes are bounded: $Y_t(0), Y_t(1) \in [0, 1]$ almost surely, and $p_t \in [f_{\min}, f_{\max}]$ with $0 < f_{\min} \leq f_{\max} < 1$.

**Assumption 4.3.** (Linear realizability in representation space). There exists $\theta_\star \in \mathbb{R}^d$ such that $\tau(x) = \langle \theta_\star, \phi(x) \rangle$ for all $x$. Moreover, $\|\theta_\star\|_2 \leq S$ and $\|\phi(x)\|_2 \leq L$ for all $x$.

**Lemma 4.4** (Unbiased pseudo-outcome under adaptive sampling). *Under Assumptions 4.1–4.2,*

$$\mathbb{E}\left[ \widetilde{Y}_t \mid X_t, p_t, \mathcal{F}_{t-1} \right] = \tau(X_t), \qquad t = 1, \ldots, B. \tag{18}$$

$$|\widetilde{Y}_t| \leq \max\left\{ \frac{1}{f_{\min}}, \frac{1}{1 - f_{\max}} \right\} \triangleq L_p.$$

Equation (18) shows that selection need not be random. If treatment is randomized for each selected unit, the queried unit gives an unbiased single-sample estimate of its own CATE. For example, we may target hard coupon users, but a randomized coupon/no-coupon decision still yields a valid causal contrast for that user. The lemma does not make the observational estimator or neural training unbiased. It relies only on RCT randomization and bounded outcomes. Adaptive selection changes which $X_t$'s we query, not whether the queried contrast is valid.

Although the learner selects $X_t$ adaptively based on past observations, the unbiasedness statement in (18) is conditional on $(X_t, p_t, \mathcal{F}_{t-1})$. Thus, the adaptive rule affects the distribution of $X_t$, but does not invalidate unbiasedness because $T_t$ is randomized after selection and is conditionally independent of potential outcomes. Technically, $\widetilde{Y}_t - \tau(X_t)$ forms a martingale difference sequence. Under this preparation, we introduce the finite-sample bound as follows:

**Theorem 4.5** (Finite-sample martingale validity and deviation bound). *Assume 4.1–4.3. Let*

$$V_\lambda \triangleq \lambda I_d + \sum_{t=1}^{B} \phi(X_t)\phi(X_t)^\top, \qquad (19)$$

*and let $\widehat{\theta}_\lambda$ be defined in (17). Then:*

*(i) Martingale estimating equation. Let $\epsilon_t = \widetilde{Y}_t - \tau(X_t)$ and $\phi_t = \phi(X_t)$. Then*

$$\mathbb{E}[\phi_t \epsilon_t \mid \mathcal{F}_{t-1}, X_t, p_t] = 0, \qquad t = 1, \ldots, B.$$

*Consequently, $\mathbb{E}\left[\sum_{t=1}^{B} \phi_t \epsilon_t\right] = 0$. If, in addition, the queried covariates $\{X_t\}_{t=1}^{B}$ and assignment probabilities $\{p_t\}_{t=1}^{B}$ are fixed in advance, or are generated independently of the RCT outcomes, then the OLS estimator with $\lambda = 0$ is conditionally unbiased given the design.*

*(ii) High-probability self-normalized bound. Fix $\delta \in (0,1)$ and any $\lambda > 0$. There exists a constant $\sigma \le 2L_p$ depending only on $f_{\min}, f_{\max}$ and the boundedness of $Y$ such that, with probability at least $1 - \delta$,*

$$\|\widehat{\theta}_\lambda - \theta_\star\|_{V_\lambda} \le \underbrace{\sigma\sqrt{2 \log \frac{\det(V_\lambda)^{1/2}}{\det(\lambda I_d)^{1/2}\delta}}}_{\text{stochastic term}} + \underbrace{\sqrt{\lambda}\, S}_{\text{regularization bias}}. \qquad (20)$$

*Consequently, for every $x \in \mathcal{X}$,*

$$\left|\widehat{\tau}_\lambda(x) - \tau(x)\right| \le \beta_B(\delta)\sqrt{\phi(x)^\top V_\lambda^{-1}\phi(x)}, \qquad (21)$$

*where*

$$\beta_B(\delta) \triangleq \sigma\sqrt{2 \log \frac{\det(V_\lambda)^{1/2}}{\det(\lambda I_d)^{1/2}\delta}} + \sqrt{\lambda}S. \qquad (22)$$

Theorem 4.5 characterizes how the experimental budget controls estimation error under *adaptive experimentation*. In particular, the error is governed by the *information matrix* $V_\lambda$, which is constructed from the queried covariates. The only price paid for adaptivity is that the $X_t$'s are not i.i.d.; the deviation bound still holds because the noise is a martingale difference. It is easy to confuse (21) with classical i.i.d. regression bounds. The crucial difference is: *the $X_t$'s can be adversarially chosen by the learner itself, based on past outcomes.* Theorem 4.5 remains valid in that setting; it does not require i.i.d. sampling of $X_t$.

For very small $B$, $V_0$ can be ill-conditioned. This motivates our statement of (20) for ridge $\lambda > 0$: the bound separates a stochastic term from an explicit regularization bias. The tradeoff is therefore easy to inspect in practice. When $V_0$ is poorly conditioned, increasing $\lambda$ stabilizes the estimate, while the bound quantifies the bias introduced by this choice.

**Corollary 4.6** (Finite-sample PEHE bound). *Under the conditions of Theorem 4.5, with probability at least $1 - \delta$,*

$$\mathcal{R}(\widehat{\tau}_\lambda) \le \beta_B(\delta) \cdot \sqrt{\mathbb{E}_{X \sim \mathbb{P}_X}\left[\phi(X)^\top V_\lambda^{-1}\phi(X)\right]}. \quad (23)$$

*In particular, if the queried design is* well-conditioned *in the sense that $V_0 \succeq \kappa B\, I_d$ for some $\kappa > 0$, and $\Sigma_X \triangleq \mathbb{E}_{\mathbb{P}_X}[\phi(X)\phi(X)^\top] \preceq L^2 I_d$, then (taking $\lambda = 0$ for simplicity),*

$$\mathcal{R}(\widehat{\tau}_0) \le \beta_B(\delta) \cdot \sqrt{\frac{\text{Tr}(\Sigma_X)}{\kappa B}} \lesssim \frac{L\,\sigma}{\sqrt{\kappa}}\sqrt{\frac{d \log(B/\delta)}{B}}. \quad (24)$$

Corollary 4.6 shows that the PEHE risk is governed by an *integrated leverage* term $\mathbb{E}_{\mathbb{P}_X}[\phi(X)^\top V_\lambda^{-1}\phi(X)]$. Active sampling is useful precisely because it can shape $V_\lambda$: selecting points that increase eigenvalues of $V_\lambda$ reduces this term. This provides a theoretical rationale for using an uncertainty proxy (Algorithm 1, $v_u$): in linear models, predictive variance is proportional to $\phi(u)^\top V_\lambda^{-1}\phi(u)$, and ensemble disagreement is a practical surrogate for that quantity.

### 4.2. Asymptotic normality under adaptive sampling

Finite-sample concentration ensures "small error with high probability." For statistical inference (e.g., confidence intervals), we further need a CLT. The non-i.i.d. nature of active sampling prevents a direct appeal to the classical i.i.d. CLT, but a martingale CLT applies.

**Assumption 4.7.** (Stabilizing design and moments). As $B \to \infty$, the normalized information matrix converges in probability:

$$\frac{1}{B}\sum_{t=1}^{B} \phi(X_t)\phi(X_t)^\top \xrightarrow{p} \Sigma_\pi, \quad \Sigma_\pi \succ 0. \quad (25)$$

Moreover, the conditional variances stabilize:

$$\frac{1}{B}\sum_{t=1}^{B}\mathbb{E}\left[\left(\widetilde{Y}_t - \tau(X_t)\right)^2 \phi(X_t)\phi(X_t)^\top \,\Big|\, \mathcal{F}_{t-1}\right] \xrightarrow{p} \Omega_\pi,$$
(26)

and a Lindeberg condition holds for the martingale differences $\{\widetilde{Y}_t - \tau(X_t)\}_{t\geq 1}$.

**Theorem 4.8** (Asymptotic normality (martingale CLT)). *Suppose Assumptions 4.1–4.7 hold and $\lambda = 0$. Then*

$$\sqrt{B}\left(\widehat{\theta}_0 - \theta_\star\right) \xrightarrow{d} \mathcal{N}\left(0, \Sigma_\pi^{-1}\Omega_\pi\Sigma_\pi^{-1}\right).$$
(27)

*Consequently, for any fixed $x \in \mathcal{X}$,*

$$\sqrt{B}\left(\widehat{\tau}_0(x) - \tau(x)\right) \xrightarrow{d} \mathcal{N}\left(0, \phi(x)^\top\Sigma_\pi^{-1}\Omega_\pi\Sigma_\pi^{-1}\phi(x)\right).$$
(28)

Theorem 4.8 says: *even though the RCT data are collected adaptively and are non-i.i.d.,* the final estimator still admits classical $\sqrt{B}$-asymptotics. In practice, this supports uncertainty quantification: one can estimate the sandwich variance $\Sigma_\pi^{-1}\Omega_\pi\Sigma_\pi^{-1}$ from data and form approximate confidence intervals for $\tau(x)$ at business-critical segments (e.g., new users or high-value users).

Notably, the technical challenge for constructing CLT beyond i.i.d is the martingale structure created by randomization: $\widetilde{Y}_t - \tau(X_t)$ is conditionally mean-zero given the past. Assumption 4.7 requires that the design does not degenerate (the information matrix stabilizes), which is exactly the failure mode active learning must avoid. This also clarifies why our acquisition score includes representativeness/shift terms ($d_u$): they discourage pathological designs where $\Sigma_\pi$ becomes nearly singular.

### 4.3. Minimax lower bound and (near) optimality

Upper bounds by themselves do not establish optimality. We therefore compare our guarantees against a minimax lower bound. This comparison leads to a tight conclusion: *even with perfect adaptivity and access to unlimited observational logs and unlabeled pools, one cannot beat the $\sqrt{d/B}$ rate in general.* Consider the linear class $\mathcal{F}_{\mathrm{lin}}(S) \triangleq \{\tau_\theta(x) = \langle\theta, \phi(x)\rangle : \theta \in \mathbb{R}^d, \|\theta\|_2 \leq S\}$. Assume $\mathbb{P}_X$ puts equal mass on $d$ covariate $\{x^{(1)}, \ldots, x^{(d)}\}$ such that $\phi(x^{(j)}) = e_j$ (the $j$-th standard basis). This captures a realistic situation where the population is a mixture of $d$ distinct segments and CATE differs by segment.

**Theorem 4.9** (Minimax lower bound for active RCT under bounded randomization). *Under the hard instance described above, suppose that $B \geq c_0 d$ and that the parameter radius satisfies $S \geq c_S d/\sqrt{B}$ for sufficiently large universal constants $c_0, c_S > 0$. Then there exists a universal constant $c > 0$ such that, for any adaptive querying policy $\pi$ possibly*

*using $\mathcal{D}_{\mathrm{obs}}$ and $\mathcal{D}_{\mathrm{pool}}$, and any estimator $\widehat{\tau}$ based on $B$ RCT samples,* $\inf_\pi \inf_{\widehat{\tau}} \sup_{\tau \in \mathcal{F}_{\mathrm{lin}}(S)} \left(\mathbb{E}_\tau^\pi\left[R(\widehat{\tau})^2\right]\right)^{1/2} \geq c\sqrt{\frac{d}{B}}$. *Here the expectation is over the randomness of the adaptive experiment, the treatment assignments, outcomes, and any internal randomness of the estimator.*

Theorem 4.9 formalizes the key point: if $\tau(\cdot)$ has $d$ independent degrees of freedom, such as $d$ user segments with genuinely different causal responses, then a budget of $B$ samples cannot yield a root-mean-square PEHE rate better than order $\sqrt{d/B}$ in general. Active learning can reallocate samples to reduce constants, but it cannot create information where none exists. It does not say that active sampling is useless. It says that the best possible scaling in $B$ is $1/\sqrt{B}$ for this class. Active sampling is still valuable because it improves the information matrix and prevents degeneracy, especially when OBS coverage is heavily imbalanced.

**Corollary 4.10** (Near-minimax optimality of the orthogonalized estimator). *Under Assumptions 4.1–4.3, a well-conditioned design $V_0 \succeq \kappa B I_d$, and the nontrivial radius regime $S \gtrsim d/\sqrt{B}$, the upper bound in (24) matches the root-mean-square minimax lower bound in Theorem 4.9 up to logarithmic factors and the conditioning constant $\kappa$. In this sense, the estimator in Algorithm 2 is minimax-rate optimal for $\mathcal{F}_{\mathrm{lin}}(S)$.*

Also, we justify using $f_k(x) \equiv 1/2$ as a default in Algorithm 1 could minimize the conditional variance, and using clipping $[f_{\min}, f_{\max}]$ as a principled safeguard when $p$ must vary by covariates.

## 5. Experiments

We evaluate the framework in two complementary settings. We first use a **synthetic benchmark** with known ground-truth CATE to measure PEHE and verify the theory-aligned behavior of active RCT design. We then conduct **real-world industrial experiments** to test whether the same active experimentation principle extends to practical uplift modeling and yields reliable gains under deployment conditions.

**Synthetic benchmark.** We construct synthetic benchmarks with dimensions $d \in \{20, 50, 100, 150\}$, biased OBS data $\mathcal{D}_{\mathrm{obs}}$, and a shifted candidate pool $\mathcal{D}_{\mathrm{pool}}$. The OBS treatment follows a strong historical policy, creating imbalance and weak overlap, while queried pool samples are randomized to form $\mathcal{D}_{\mathrm{rct}}$. OBS is used only to build design signals, and the final CATE estimator is fitted on selected RCT samples via pseudo-outcome ridge regression.

**Real-world industrial dataset.** We further evaluate the method on a large-scale ride-hailing dataset collected at the order level. Each instance contains 468 features. The treatment $T \in \{0, 1\}$ indicates whether a "free upgrade"

service was triggered, and the outcome $Y \in \{0, 1\}$ denotes order completion. We use an Out-of-Time (OOT) protocol over 44 days. The first 30 days provide the RCT pool $\mathcal{D}_{\text{pool}}$ and observational training data, either $\mathcal{D}_{\text{obs}}^{\text{bias}}$ or $\mathcal{D}_{\text{obs}}^{\text{full}}$. The remaining 14 days are used as an unbiased RCT test set $\mathcal{D}_{\text{rct}}^{\text{test}}$. Additional analyses, including ablation studies, sensitivity analysis, and experiments with different proportions of observational data, are reported in the appendix C.6.

**Baselines & Metrics.** *(i) For the synthetic benchmark*, we compare active sampling with random sampling under the same RCT budget and report $\sqrt{\text{PEHE}}$. Each synthetic experiment is repeated 30 times, and results are reported as mean $\pm$ standard deviation. *(ii) For the real-world industrial experiments*, we evaluate DragonNet, DESCN (Zhong et al., 2022), and DRCFR under active versus random sampling. Since PEHE cannot be directly computed without counterfactual outcomes, we report AUUC (Gutierrez & Gérardy, 2017). Following (Zhong et al., 2022), each real-world experiment is repeated 5 times, and results are reported as mean $\pm$ standard deviation.

**Main results on the industrial dataset.** As shown in Figure 2, active sampling yields higher mean AUUC than random sampling in most cases. For example, DRCFR with 50k active RCT samples reaches an AUUC of 0.6534, exceeding random sampling with 300k samples (0.6466). This indicates better sample efficiency under the evaluated experimental protocol.

**Main results on the Synthetic dataset.** Across the 20D, 50D, 100D, and 150D settings, $\sqrt{\text{PEHE}}$ decreases as the RCT budget grows, while errors increase with dimension. This trend is consistent with the $\mathcal{O}(\sqrt{d/B})$ dependence in our finite-sample bound. Active sampling gives lower $\sqrt{\text{PEHE}}$ than random sampling in most cases, especially for small budgets or high-dimensional covariates, suggesting that active design reduces finite-sample constants by improving the conditioning of the information matrix.

## 6. Conclusion

We study heterogeneous treatment effect estimation under a pervasive real-world constraint: randomized controlled trials are scarce and costly, while OBS data are abundant but biased. In our framework, OBS data inform *where* randomized experimentation should occur, while causal identification is carried exclusively by randomized experiments. We formalize this perspective as a ***budgeted active experimentation*** problem and propose a ***multi-criteria acquisition strategy*** that leverages OBS data to diagnose uncertainty, overlap deficits, and covariate shift, thereby directing limited RCT budget to the most informative regions.

Our theoretical analysis shows that randomization induces a martingale structure, which preserves unbiasedness un-

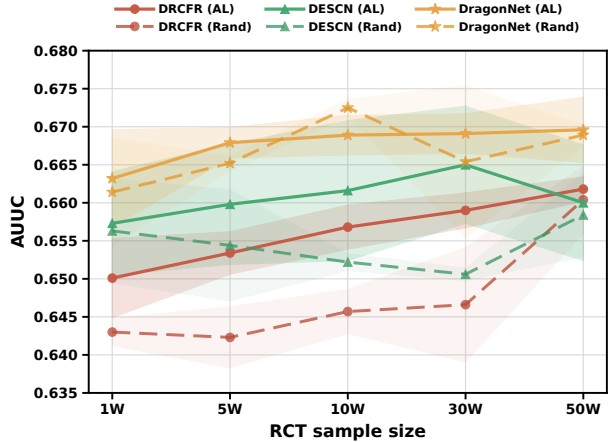

*Figure 2.* Performance comparison of Active Learning (AL) versus Random Sampling (Rand) strategies across varying RCT sample sizes (10k to 500k). The curves display the AUUC scores for DRCFR, DESCN, and DragonNet models. The experiments are conducted on the biased observational set $\mathcal{D}_{\text{obs}}^{\text{bias}}$, demonstrating the improved sample efficiency of active sampling (solid lines) compared to random sampling (dashed lines) under distribution shifts.

der adaptive sampling and leads to finite-sample deviation bounds and asymptotic normality. We further establish minimax lower bounds showing that the achieved $\sqrt{d/B}$ rate is near-optimal in the linear representation setting. These results clarify the role of active experimentation: it improves finite-sample efficiency by producing a better experimental design, rather than changing the fundamental statistical rate.

The experiments support these theoretical findings. On synthetic benchmarks, active sampling achieves lower $\sqrt{\text{PEHE}}$ than random sampling, especially under small RCT budgets and high-dimensional settings. We further apply the framework to industrial neural uplift models, where active sampling improves AUUC across multiple CATE backbones and can reduce the RCT budget required to reach comparable performance. These results suggest that the proposed design is effective for improving RCT sample efficiency in large-scale OBS–RCT fusion scenarios.

**Future work.** A natural direction is to sharpen the theory for nonparametric and deep function classes, for example by making the approximation and sequential-complexity terms explicit beyond the linear-head analysis used here. Practical deployment also calls for designs that balance unit selection, treatment randomization, and operational constraints. These goals become harder in multi-source settings, where randomized trials, observational records, and historical logs may differ in bias, coverage, and treatment support, especially when covariates and treatment responses shift over time. Addressing these issues would make active experimentation more useful for cost-efficient causal learning.

## Acknowledgements

Zhiheng Zhang is supported by the Fundamental Research Funds for the Central Universities (Grant No. 2025110602) of Shanghai University of Finance and Economics, and Independent Research Project (Grant No. 2026110081) funded by the School of Statistics and Data Science. This work was supported by the Shanghai Engineering Research Center of Finance Intelligence (Grant No. 19DZ2254600).

In addition, we thank Professor Chengchun Shi from the London School of Economics and Political Science for his valuable guidance and support. We also gratefully acknowledge DiDi Chuxing for providing the data and computational resources used in this work.

## Impact Statement

This paper presents work whose goal is to advance the field of machine learning. There are many potential societal consequences of our work, none of which we feel must be specifically highlighted here.

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

# A. Related Work

Industrial CATE estimation often faces a central challenge: limited RCT budgets versus abundant but biased OBS data. This motivates us to develop an OBS & RCT data fusion framework that casts experiment design as a budgeted active learning problem. In this section, we discuss related work on (i) *active sampling for budget-limited experiments*, (ii) *active sampling in observational studies*, and (iii) *active learning with neural networks*, which is also based on our previous work (Zhang et al., 2025; Wang et al., 2026; Zhang & Wang, 2026)

**(i) Active sampling for budget-limited experiments.** In sample-constrained RCT settings, the unlabeled pool is available but only a small subset of individuals can be enrolled, and the goal is to minimize estimation error via selective recruitment. Addanki et al. (2022) utilize leverage-based selection and balanced assignment to optimize ATE deviation under fixed budgets. Similarly, Zhang et al. (2025) propose adaptive sampling with reweighting (RWAS) to achieve stable, sample-efficient ATE estimation within a limited draw budget.

**(ii) Active sampling in observational studies.** In observational settings, treatment is pre-assigned, and the budget restricts label queries. Wen et al. (2025) prioritize enhancing *factual and counterfactual coverage* to improve overlap and reduce PEHE. Alternatively, Causal-EPIG (Gao et al., 2025) employs an information-theoretic approach, selecting samples that maximize expected predictive information gain to minimize CATE uncertainty under a limited labeling budget.

**(iii) Active learning with neural networks.** In neural settings, acquisition is often driven by disagreement or uncertainty. Zhu & Nowak (2022) propose NeuralCAL, which queries points in the *disagreement region* of plausible predictors to reduce label complexity. Its extension, NeuralCAL++, further integrates *abstention* mechanisms and confidence intervals to reach target error rates with significantly fewer queries.

*Table 1.* Comparison of related works in Active Learning (AL) for Treatment Effect Estimation.

| AL Methods | Data | Objective & Budget | Guides Exp. Design? | Acquisition Strategy | Evaluation Metric |
|---|---|---|---|---|---|
| Ours. **(AL for RCT&OBS TE)** | OBS + RCT | **Obj:** Min. CATE Error **Budget:** RCT Experiment Budget | ✓ | **Ours:** Top-$k$ by uncertainty + discrepancy + overlap deficit. | **PEHE** (For CATE) **AUUC** (For CATE) |
| Sample-Constrained TE Est. (Addanki et al., 2022) | RCT only | **Obj:** Min. ATE/ITE Error **Budget:** Limited Sample Size | ✓ | **SCTE:** Top-$k$ informative units with covariate-balanced assignment. | **RMSE** (For ITE) **Deviation** (For ATE) |
| Active TE Est. (Zhang et al., 2025) | RCT only | **Obj:** Min. ATE Error **Budget:** Limited Sample Size | ✓ | **RWAS:** Probability sampling + inverse-propensity reweighting. | **Deviation** (For ATE) |
| Enhancing TE Est. (Wen et al., 2025) | OBS only | **Obj:** Min. CATE Error **Budget:** Labeling Budget | ✗ | **Coverage:** Top-$k$ maximizing factual/counterfactual coverage. | **RMSE** (For ITE) **PEHE** (For CATE) |
| Causal-EPIG (Gao et al., 2025) | OBS only | **Obj:** Min. CATE Error **Budget:** Labeling Budget | ✗ | **Causal-EPIG:** Top-$k$ by expected predictive information gain. | **PEHE** (For CATE) |
| AL with Neural Networks (Zhu & Nowak, 2022) | Supervised | **Obj:** Optim. Label Complexity and Excess Error **Budget:** Query Count | ✗ | **NeuralCAL(++):** Top-$k$ by disagreement/ambiguity (with abstention). | **Excess Error Label Complexity** |

**Note:** OBS = Observational Data, RCT = Randomized Controlled Trial. **Guides Exp. Design?**: Indicates if the method actively assigns treatments (✓) or only queries labels (✗).

# B. Theoretical Analysis

This section gives theoretical guarantees for the *estimation component* of Algorithm 1 under adaptively collected RCT data. Our goal is to state three results precisely:

(i) **Finite-sample validity**: a nearly unbiased effect estimator, together with a non-asymptotic error bound that remains valid under *adaptive, non-i.i.d.* RCT sampling.

(ii) **Asymptotic inference**: a central limit theorem (CLT) that establishes asymptotic normality and supports confidence intervals.

(iii) **Fundamental limits**: a minimax lower bound showing that the achieved rate is information-theoretically optimal up to logarithmic factors. The central observation is that randomization protects unbiasedness, whereas active selection shapes the information matrix and therefore the error rate. We first formalize an analyzable estimator that follows the RCT protocol in Algorithm 1 and is standard in modern causal ML: it regresses a *pseudo-outcome* onto a representation.

## B.1. Setup: adaptive RCT stream and pseudo-outcome regression

Let $B$ denote the total RCT budget and index queried units in chronological order $t = 1, \ldots, B$. Write the $t$-th queried sample as $(X_t, T_t, Y_t, p_t)$ where $p_t \in [f_{\min}, f_{\max}]$ is the (known) assignment probability used when unit $t$ was experimented. Let $\{\mathcal{F}_t\}_{t \geq 0}$ be the filtration where $\mathcal{F}_t$ contains $\mathcal{D}_{\mathrm{obs}}$, the full covariate pool $\mathcal{D}_{\mathrm{pool}}$, and all RCT data up to time $t$. The adaptive querying policy (via $\mathcal{C}_k$, $f_k$) implies that $X_t$ and $p_t$ may depend on $\mathcal{F}_{t-1}$.

We define the RCT pseudo-outcome

$$\widetilde{Y}_t \triangleq \frac{T_t Y_t}{p_t} - \frac{(1 - T_t) Y_t}{1 - p_t}. \tag{29}$$

Intuitively, $\widetilde{Y}_t$ converts each single randomized trial into a noisy but unbiased "label" of the treatment effect at $X_t$.

Let $\phi(x) \in \mathbb{R}^d$ be a fixed feature map (e.g., the learned embedding $\phi$ in Algorithm 1). For theory, we analyze a realizable linear CATE model in this representation:

$$\tau(x) = \langle \theta_\star, \phi(x) \rangle, \qquad \theta_\star \in \mathbb{R}^d. \tag{30}$$

Given $\{(X_t, T_t, Y_t, p_t)\}_{t=1}^B$, we estimate $\theta_\star$ by (optionally regularized) least squares on pseudo-outcomes:

$$\widehat{\theta}_\lambda \triangleq \arg\min_{\theta \in \mathbb{R}^d} \sum_{t=1}^B \left( \widetilde{Y}_t - \langle \theta, \phi(X_t) \rangle \right)^2 + \lambda \|\theta\|_2^2, \tag{31}$$

and output $\widehat{\tau}_\lambda(x) \triangleq \langle \widehat{\theta}_\lambda, \phi(x) \rangle$. Algorithm 2 summarizes this estimator.

Algorithm 1 specifies *how the $X_t$'s are chosen* (active sampling) and *how $p_t$ is assigned* (bounded randomization). Algorithm 2 isolates a transparent estimator for analyzing the statistical consequences of that adaptive design. In practice, one may replace the linear regressor by a neural CATE learner; the key objects in the proofs below are (i) the unbiased pseudo-outcome, and (ii) an information matrix that quantifies how well the queried points cover the representation space.

## B.2. Unbiasedness and finite-sample error bounds

We now state the first main result: despite *adaptive*, non-i.i.d. selection of $X_t$, the RCT pseudo-outcome yields a martingale estimating equation. This leads to a finite-sample self-normalized deviation bound for the ridge estimator with $\lambda > 0$, and recovers conditional unbiasedness of the OLS estimator when $\lambda = 0$ and the queried design is fixed in advance or generated independently of the RCT outcomes.

**Assumption B.1.** (Predictable adaptive design). For each $t$, the queried covariate $X_t$ and assignment probability $p_t$ are $\mathcal{F}_{t-1}$-measurable.

**Assumption B.2.** (RCT randomization and boundedness). Conditioned on $(X_t, p_t, \mathcal{F}_{t-1})$, treatment is randomized as $T_t \sim \mathrm{Bern}(p_t)$ and $T_t \perp (Y_t(0), Y_t(1)) \mid (X_t, p_t, \mathcal{F}_{t-1})$. Outcomes are bounded: $Y_t(0), Y_t(1) \in [0, 1]$ almost surely, and $p_t \in [f_{\min}, f_{\max}]$ with $0 < f_{\min} \leq f_{\max} < 1$.

**Assumption B.3.** (Linear realizability in representation space). There exists $\theta_\star \in \mathbb{R}^d$ such that $\tau(x) = \langle \theta_\star, \phi(x) \rangle$ for all $x$. Moreover, $\|\theta_\star\|_2 \leq S$ and $\|\phi(x)\|_2 \leq L$ for all $x$.

**Lemma B.4** (Unbiased pseudo-outcome under adaptive sampling). *Under Assumptions 4.1–4.2,*

$$\mathbb{E}\Big[\widetilde{Y}_t \mid X_t, p_t, \mathcal{F}_{t-1}\Big] \;=\; \tau(X_t), \qquad t = 1, \dots, B. \tag{32}$$

*Moreover, $\widetilde{Y}_t$ is uniformly bounded:*

$$|\widetilde{Y}_t| \;\leq\; \max\left\{\frac{1}{f_{\min}}, \frac{1}{1 - f_{\max}}\right\} \;\triangleq\; L_p. \tag{33}$$

*Proof.* Fix $t \in \{1, \dots, B\}$. By consistency and $Y_t = T_t Y_t(1) + (1 - T_t)Y_t(0)$,

$$T_t Y_t = T_t Y_t(1), \qquad (1 - T_t)Y_t = (1 - T_t)Y_t(0).$$

Hence,

$$\widetilde{Y}_t = \frac{T_t Y_t(1)}{p_t} - \frac{(1 - T_t)Y_t(0)}{1 - p_t}.$$

Taking conditional expectation given $(X_t, p_t, \mathcal{F}_{t-1})$ and using Assumption 4.2,

$$\mathbb{E}\left[\frac{T_t Y_t(1)}{p_t} \,\bigg|\, X_t, p_t, \mathcal{F}_{t-1}\right] = \mathbb{E}\left[\mathbb{E}\left[\frac{T_t}{p_t} \,\bigg|\, X_t, p_t, \mathcal{F}_{t-1}, Y_t(1)\right] Y_t(1) \,\bigg|\, X_t, p_t, \mathcal{F}_{t-1}\right]$$

$$= \mathbb{E}\left[\frac{\mathbb{E}[T_t \mid X_t, p_t, \mathcal{F}_{t-1}]}{p_t} Y_t(1) \,\bigg|\, X_t, p_t, \mathcal{F}_{t-1}\right]$$

$$= \mathbb{E}[Y_t(1) \mid X_t, p_t, \mathcal{F}_{t-1}],$$

where the second line uses conditional independence $T_t \perp Y_t(1) \mid (X_t, p_t, \mathcal{F}_{t-1})$ and the last line uses $\mathbb{E}[T_t \mid X_t, p_t, \mathcal{F}_{t-1}] = p_t$. Similarly,

$$\mathbb{E}\left[\frac{(1 - T_t)Y_t(0)}{1 - p_t} \,\bigg|\, X_t, p_t, \mathcal{F}_{t-1}\right] = \mathbb{E}[Y_t(0) \mid X_t, p_t, \mathcal{F}_{t-1}].$$

Therefore,

$$\mathbb{E}\Big[\widetilde{Y}_t \mid X_t, p_t, \mathcal{F}_{t-1}\Big] = \mathbb{E}[Y_t(1) - Y_t(0) \mid X_t, p_t, \mathcal{F}_{t-1}] = \mathbb{E}[Y_t(1) - Y_t(0) \mid X_t] = \tau(X_t),$$

where the second equality uses the randomization/ignorability in Assumption 4.2 and the last equality is the definition of CATE.

For boundedness, note that exactly one of the two terms in the pseudo outcome is nonzero: if $T_t = 1$, then $\widetilde{Y}_t = Y_t/p_t \in [0, 1/p_t]$; if $T_t = 0$, then $\widetilde{Y}_t = -Y_t/(1 - p_t) \in [-1/(1 - p_t), 0]$. Using $p_t \geq f_{\min}$ and $1 - p_t \geq 1 - f_{\max}$ gives (33). $\qquad\square$

Equation (18) shows that selection need not be random. If treatment is randomized for each selected unit, the queried unit gives an unbiased single-sample estimate of its own CATE. For example, we may target "hard" coupon users, but a randomized coupon/no-coupon decision still yields a valid causal contrast for that user. The lemma does not make the observational estimator or neural training unbiased. It relies only on RCT randomization and bounded outcomes. Adaptive selection changes which $X_t$'s we query, not whether the queried contrast is valid.

A natural concern is that, because the learner chooses $X_t$ based on past outcomes, the resulting estimator may suffer from selection bias and no longer be unbiased. The resolution is that unbiasedness in (18) is *conditional* on $(X_t, p_t, \mathcal{F}_{t-1})$: the selection may change the distribution of $X_t$, but $T_t$ is randomized *after* selection and is conditionally independent of potential outcomes. Technically, $\widetilde{Y}_t - \tau(X_t)$ forms a martingale difference sequence.

**A key martingale object.** Define the centered pseudo-outcome noise

$$\varepsilon_t \;\triangleq\; \widetilde{Y}_t - \tau(X_t), \tag{34}$$

and the (predictable) feature vector $\phi_t \triangleq \phi(X_t)$. By Lemma 4.4, $\mathbb{E}[\varepsilon_t \mid \mathcal{F}_{t-1}, X_t, p_t] = 0$ and $|\varepsilon_t| \leq 2L_p$ (since both $\widetilde{Y}_t$ and $\tau(X_t) \in [-1, 1]$ are bounded). Let

$$V_\lambda \triangleq \lambda I_d + \sum_{t=1}^{B} \phi_t \phi_t^\top, \qquad M_B \triangleq \sum_{t=1}^{B} \phi_t \varepsilon_t. \tag{35}$$

Then $\{M_t\}_{t\geq 0}$ with $M_t = \sum_{s=1}^{t} \phi_s \varepsilon_s$ is a vector-valued martingale with respect to $\{\mathcal{F}_t\}$.

**A self-normalized concentration lemma (proved in full).** The finite-sample bound relies on a vector-valued self-normalized martingale inequality. Because this step is often cited as a black box, we provide a complete proof to make the non-i.i.d. aspect fully transparent.

**Lemma B.5** (Conditional sub-Gaussianity of $\varepsilon_t$). *Under Assumptions 4.1–4.2, for each $t$ and any $\lambda \in \mathbb{R}$,*

$$\mathbb{E}\big[\exp(\lambda \varepsilon_t) \,\big|\, \mathcal{F}_{t-1}, X_t, p_t\big] \leq \exp\left(\frac{\lambda^2 \sigma^2}{2}\right), \qquad \text{with } \sigma \triangleq 2L_p. \tag{36}$$

*Proof.* Condition on $(\mathcal{F}_{t-1}, X_t, p_t)$. By Lemma 4.4, $\mathbb{E}[\varepsilon_t \mid \mathcal{F}_{t-1}, X_t, p_t] = 0$. Moreover, $Y_t \in [0, 1]$ and $p_t \in [f_{\min}, f_{\max}]$ imply $|\widetilde{Y}_t| \leq L_p$ by (33). Also $\tau(X_t) = \mathbb{E}[Y(1) - Y(0) \mid X_t] \in [-1, 1]$ because $Y(1), Y(0) \in [0, 1]$. Hence $|\varepsilon_t| \leq |\widetilde{Y}_t| + |\tau(X_t)| \leq L_p + 1 \leq 2L_p$ since $L_p \geq 1$.

Now apply Hoeffding's lemma in conditional form: if $Z$ is zero-mean and almost surely lies in $[a, b]$ given a sigma-field, then $\mathbb{E}[e^{\lambda Z} \mid \cdot] \leq \exp(\lambda^2 (b-a)^2/8)$. Here $Z = \varepsilon_t$ and $[a, b] = [-2L_p, 2L_p]$, so $(b-a)^2/8 = (4L_p)^2/8 = 2L_p^2$. Thus (36) holds with $\sigma = 2L_p$. $\square$

**Lemma B.6** (Self-normalized martingale inequality (vector form)). *Assume $\lambda > 0$ and let $V_\lambda$ and $M_B$ be defined in (35). Suppose $\varepsilon_t$ satisfies the conditional sub-Gaussian property (36) with parameter $\sigma$ and $\phi_t$ is $\mathcal{F}_{t-1}$-measurable. Then for any $\delta \in (0, 1)$, with probability at least $1 - \delta$,*

$$\|M_B\|_{V_\lambda^{-1}} \leq \sigma \sqrt{2 \log\left(\frac{\det(V_\lambda)^{1/2}}{\det(\lambda I_d)^{1/2} \, \delta}\right)}. \tag{37}$$

*Here $\|z\|_A \triangleq \sqrt{z^\top A z}$ for any positive semidefinite $A$.*

*Proof.* The proof proceeds by constructing an exponential supermartingale and applying a mixture (Gaussian integration) argument.

**Step 1: exponential supermartingale for a fixed direction.** Fix any $u \in \mathbb{R}^d$ and define for $t = 0, 1, \ldots, B$,

$$Z_t(u) \triangleq \exp\left(\frac{1}{\sigma} u^\top M_t - \frac{1}{2} u^\top \left(\sum_{s=1}^{t} \phi_s \phi_s^\top\right) u\right), \tag{38}$$

with the convention $M_0 = 0$ and the empty sum equals 0.

We claim that $\{Z_t(u)\}_{t=0}^{B}$ is a nonnegative supermartingale w.r.t. $\{\mathcal{F}_t\}$. Indeed, using $M_t = M_{t-1} + \phi_t \varepsilon_t$ and $\phi_t$ being $\mathcal{F}_{t-1}$-measurable,

$$Z_t(u) = Z_{t-1}(u) \cdot \exp\left(\frac{1}{\sigma} u^\top \phi_t \varepsilon_t - \frac{1}{2} u^\top \phi_t \phi_t^\top u\right)$$

$$= Z_{t-1}(u) \cdot \exp\left(\frac{1}{\sigma} (u^\top \phi_t) \varepsilon_t - \frac{1}{2} (u^\top \phi_t)^2\right).$$

Taking conditional expectation given $\mathcal{F}_{t-1}$ and applying (36) with $\lambda = (u^\top \phi_t)/\sigma$ yields

$$\mathbb{E}[Z_t(u) \mid \mathcal{F}_{t-1}] = Z_{t-1}(u) \cdot \exp\left(-\frac{1}{2} (u^\top \phi_t)^2\right) \cdot \mathbb{E}\left[\exp\left(\frac{1}{\sigma} (u^\top \phi_t) \varepsilon_t\right) \,\bigg|\, \mathcal{F}_{t-1}\right]$$

$$\leq Z_{t-1}(u) \cdot \exp\left(-\frac{1}{2} (u^\top \phi_t)^2\right) \cdot \exp\left(\frac{1}{2} (u^\top \phi_t)^2\right) = Z_{t-1}(u).$$

Thus $\{Z_t(u)\}$ is a supermartingale and in particular $\mathbb{E}[Z_B(u)] \leq Z_0(u) = 1$.

**Step 2: mixture over $u$ to obtain a uniform bound.** Let $U \sim \mathcal{N}(0, \lambda^{-1} I_d)$ be independent of everything else. Define the mixture random variable

$$\overline{Z}_B \triangleq \mathbb{E}[Z_B(U) \,|\, \mathcal{F}_B].$$

Since $Z_B(u) \geq 0$ for all $u$, by Fubini's theorem and the supermartingale property,

$$\mathbb{E}[\overline{Z}_B] = \mathbb{E}\big[\mathbb{E}[Z_B(U) \mid \mathcal{F}_B]\big] = \mathbb{E}[Z_B(U)] = \mathbb{E}\big[\mathbb{E}[Z_B(U)]\big] \leq 1.$$

**Step 3: explicit evaluation of the Gaussian integral.** Write $\Sigma_B \triangleq \sum_{t=1}^B \phi_t \phi_t^\top$ so that $V_\lambda = \lambda I_d + \Sigma_B$. Using the definition (38) at $t = B$,

$$Z_B(u) = \exp\left(\frac{1}{\sigma} u^\top M_B - \frac{1}{2} u^\top \Sigma_B u\right).$$

The density of $U \sim \mathcal{N}(0, \lambda^{-1} I_d)$ is

$$(2\pi)^{-d/2} \det(\lambda^{-1} I_d)^{-1/2} \exp\left(-\frac{1}{2} u^\top (\lambda I_d) u\right) = (2\pi)^{-d/2} \det(\lambda I_d)^{1/2} \exp\left(-\frac{1}{2} u^\top (\lambda I_d) u\right).$$

Therefore, conditioning on $\mathcal{F}_B$ (so $M_B$ and $\Sigma_B$ are fixed),

$$\begin{aligned}
\overline{Z}_B &= \int_{\mathbb{R}^d} \exp\left(\frac{1}{\sigma} u^\top M_B - \frac{1}{2} u^\top \Sigma_B u\right) \cdot (2\pi)^{-d/2} \det(\lambda I_d)^{1/2} \exp\left(-\frac{1}{2} u^\top (\lambda I_d) u\right) du \\
&= (2\pi)^{-d/2} \det(\lambda I_d)^{1/2} \int_{\mathbb{R}^d} \exp\left(\frac{1}{\sigma} u^\top M_B - \frac{1}{2} u^\top (\Sigma_B + \lambda I_d) u\right) du \\
&= (2\pi)^{-d/2} \det(\lambda I_d)^{1/2} \int_{\mathbb{R}^d} \exp\left(\frac{1}{\sigma} u^\top M_B - \frac{1}{2} u^\top V_\lambda u\right) du.
\end{aligned}$$

Applying the standard Gaussian integral identity

$$\int_{\mathbb{R}^d} \exp\left(b^\top u - \frac{1}{2} u^\top A u\right) du = (2\pi)^{d/2} \det(A)^{-1/2} \exp\left(\frac{1}{2} b^\top A^{-1} b\right), \quad A \succ 0,$$

with $A = V_\lambda$ and $b = M_B / \sigma$ gives

$$\overline{Z}_B = \frac{\det(\lambda I_d)^{1/2}}{\det(V_\lambda)^{1/2}} \exp\left(\frac{1}{2\sigma^2} M_B^\top V_\lambda^{-1} M_B\right) = \frac{\det(\lambda I_d)^{1/2}}{\det(V_\lambda)^{1/2}} \exp\left(\frac{1}{2\sigma^2} \|M_B\|_{V_\lambda^{-1}}^2\right). \tag{39}$$

**Step 4: apply Markov's inequality.** Since $\mathbb{E}[\overline{Z}_B] \leq 1$, for any $\delta \in (0, 1)$,

$$\mathbb{P}\left(\overline{Z}_B \geq \frac{1}{\delta}\right) \leq \delta.$$

On the complement event (probability at least $1 - \delta$), combine with (39) to obtain

$$\frac{\det(\lambda I_d)^{1/2}}{\det(V_\lambda)^{1/2}} \exp\left(\frac{1}{2\sigma^2} \|M_B\|_{V_\lambda^{-1}}^2\right) \leq \frac{1}{\delta}.$$

Taking logarithms and rearranging yields exactly (37). $\qquad\square$

**Main finite-sample theorem.** We can now prove the finite-sample guarantee in full detail.

**Theorem B.7** (Finite-sample martingale validity and deviation bound). *Assume B.1–B.3 and define $V_\lambda$ as in (35). Let $\widehat{\theta}_\lambda$ be defined by (31). Then:*

1. **(Martingale estimating equation).** *Let $\epsilon_t = \widetilde{Y}_t - \tau(X_t)$ and $\phi_t = \phi(X_t)$. Then*

$$\mathbb{E}[\phi_t \epsilon_t \mid \mathcal{F}_{t-1}, X_t, p_t] = 0, \qquad t = 1, \ldots, B.$$

*Consequently,*

$$\mathbb{E}\left[\sum_{t=1}^{B} \phi_t \epsilon_t\right] = 0.$$

*If, in addition, the queried covariates $\{X_t\}_{t=1}^{B}$ and assignment probabilities $\{p_t\}_{t=1}^{B}$ are fixed in advance, or are generated independently of the RCT outcomes, then the OLS estimator with $\lambda = 0$ is conditionally unbiased given the design.*

2. **(High-probability self-normalized bound).** *Fix $\delta \in (0, 1)$ and any $\lambda > 0$. With probability at least $1 - \delta$,*

$$\|\widehat{\theta}_\lambda - \theta_\star\|_{V_\lambda} \leq \underbrace{\sigma\sqrt{2\log\left(\frac{\det(V_\lambda)^{1/2}}{\det(\lambda I_d)^{1/2}\,\delta}\right)}}_{\text{stochastic term}} + \underbrace{\sqrt{\lambda}S}_{\text{regularization bias}}, \tag{40}$$

*where $\sigma$ can be taken as $2L_p$ (Lemma B.5). Consequently, for every $x \in \mathcal{X}$,*

$$\left|\widehat{\tau}_\lambda(x) - \tau(x)\right| \leq \beta_B(\delta) \cdot \sqrt{\phi(x)^\top V_\lambda^{-1}\phi(x)}, \qquad \beta_B(\delta) \triangleq \sigma\sqrt{2\log\left(\frac{\det(V_\lambda)^{1/2}}{\det(\lambda I_d)^{1/2}\,\delta}\right)} + \sqrt{\lambda}S. \tag{41}$$

*Proof.* **Part (i): martingale estimating equation.** By Lemma B.4,

$$\mathbb{E}[\epsilon_t \mid \mathcal{F}_{t-1}, X_t, p_t] = 0.$$

Since $\phi_t = \phi(X_t)$ is measurable with respect to $\sigma(\mathcal{F}_{t-1}, X_t, p_t)$, it follows that

$$\mathbb{E}[\phi_t \epsilon_t \mid \mathcal{F}_{t-1}, X_t, p_t] = 0.$$

Summing over $t$ gives

$$\mathbb{E}\left[\sum_{t=1}^{B} \phi_t \epsilon_t\right] = 0.$$

If the design is fixed in advance, or is generated independently of the RCT outcomes, then conditioning on the full design does not reveal any information about the martingale increments. In that special case,

$$\mathbb{E}[\epsilon \mid \{X_t, p_t\}_{t=1}^{B}] = 0,$$

and the usual OLS conditional-unbiasedness calculation applies.

**Part (ii): high-probability self-normalized bound ($\lambda > 0$).** From the ridge normal equations, the closed form is

$$\widehat{\theta}_\lambda = V_\lambda^{-1} \sum_{t=1}^{B} \phi_t \widetilde{Y}_t = V_\lambda^{-1} \sum_{t=1}^{B} \phi_t(\phi_t^\top \theta_\star + \varepsilon_t) = V_\lambda^{-1}\left(\sum_{t=1}^{B} \phi_t \phi_t^\top\right)\theta_\star + V_\lambda^{-1} \sum_{t=1}^{B} \phi_t \varepsilon_t.$$

Since $\sum_{t=1}^{B} \phi_t \phi_t^\top = V_\lambda - \lambda I_d$, we get the exact identity

$$\widehat{\theta}_\lambda - \theta_\star = V_\lambda^{-1} M_B - \lambda V_\lambda^{-1}\theta_\star, \tag{42}$$

where $M_B = \sum_{t=1}^{B} \phi_t \varepsilon_t$ as in (35). Take the $V_\lambda$-norm:

$$\|\widehat{\theta}_\lambda - \theta_\star\|_{V_\lambda} \leq \|V_\lambda^{-1} M_B\|_{V_\lambda} + \lambda\|V_\lambda^{-1}\theta_\star\|_{V_\lambda}.$$

Using $\|V_\lambda^{-1} M_B\|_{V_\lambda} = \|M_B\|_{V_\lambda^{-1}}$ and $\lambda\|V_\lambda^{-1}\theta_\star\|_{V_\lambda} = \lambda\sqrt{\theta_\star^\top V_\lambda^{-1}\theta_\star} \le \lambda\sqrt{\theta_\star^\top (\lambda I_d)^{-1}\theta_\star} = \sqrt{\lambda}\,\|\theta_\star\|_2 \le \sqrt{\lambda}\,S$, we obtain

$$\|\widehat{\theta}_\lambda - \theta_\star\|_{V_\lambda} \le \|M_B\|_{V_\lambda^{-1}} + \sqrt{\lambda}S. \tag{43}$$

By Lemma B.5 and Lemma B.6, with probability at least $1 - \delta$,

$$\|M_B\|_{V_\lambda^{-1}} \le \sigma\sqrt{2\log\left(\frac{\det(V_\lambda)^{1/2}}{\det(\lambda I_d)^{1/2}\,\delta}\right)}.$$

Plugging into (43) proves (20).

Finally, for any $x \in \mathcal{X}$,

$$|\widehat{\tau}_\lambda(x) - \tau(x)| = |\phi(x)^\top(\widehat{\theta}_\lambda - \theta_\star)| \le \|\widehat{\theta}_\lambda - \theta_\star\|_{V_\lambda} \cdot \|\phi(x)\|_{V_\lambda^{-1}} = \|\widehat{\theta}_\lambda - \theta_\star\|_{V_\lambda} \cdot \sqrt{\phi(x)^\top V_\lambda^{-1}\phi(x)},$$

which combined with (20) yields (21). $\qquad\square$

Theorem 4.5 quantifies a clean "budget $\Rightarrow$ error" tradeoff under *adaptive* experimentation: the estimation error is controlled by an *information matrix* $V_\lambda$ built from the queried covariates. The crucial difference from classical i.i.d. regression is that the $X_t$'s may be chosen adaptively based on past outcomes; nevertheless, the deviation bound remains valid because the centered pseudo-outcome noise forms a martingale difference sequence.

**Corollary B.8** (Finite-sample PEHE bound). *Under the conditions of Theorem 4.5, with probability at least $1 - \delta$,*

$$\mathcal{R}(\widehat{\tau}_\lambda) \le \beta_B(\delta) \cdot \sqrt{\mathbb{E}_{X\sim\mathbb{P}_X}\left[\phi(X)^\top V_\lambda^{-1}\phi(X)\right]}. \tag{44}$$

*In particular, if the queried design is* well-conditioned *in the sense that $V_0 \succeq \kappa B\, I_d$ for some $\kappa > 0$, and $\Sigma_X \triangleq \mathbb{E}_{\mathbb{P}_X}[\phi(X)\phi(X)^\top] \preceq L^2 I_d$, then (taking $\lambda = 0$ for simplicity and assuming $V_0$ is invertible),*

$$\mathcal{R}(\widehat{\tau}_0) \le \beta_B(\delta) \cdot \sqrt{\frac{\mathrm{Tr}(\Sigma_X)}{\kappa B}} \lesssim \frac{L\,\sigma}{\sqrt{\kappa}}\sqrt{\frac{d\log(B/\delta)}{B}}. \tag{45}$$

*Proof.* For any fixed $x$, by Cauchy–Schwarz in the $V_\lambda$-inner product,

$$(\widehat{\tau}_\lambda(x) - \tau(x))^2 = (\phi(x)^\top(\widehat{\theta}_\lambda - \theta_\star))^2 \le \|\widehat{\theta}_\lambda - \theta_\star\|_{V_\lambda}^2 \cdot \phi(x)^\top V_\lambda^{-1}\phi(x).$$

Taking expectation over $X \sim \mathbb{P}_X$ gives

$$\mathbb{E}_{X\sim\mathbb{P}_X}\left[(\widehat{\tau}_\lambda(X) - \tau(X))^2\right] \le \|\widehat{\theta}_\lambda - \theta_\star\|_{V_\lambda}^2 \cdot \mathbb{E}_{X\sim\mathbb{P}_X}\left[\phi(X)^\top V_\lambda^{-1}\phi(X)\right].$$

Taking square roots and using the high-probability bound $\|\widehat{\theta}_\lambda - \theta_\star\|_{V_\lambda} \le \beta_B(\delta)$ from Theorem 4.5 yields (23).

For the "well-conditioned" simplification, $V_0 \succeq \kappa B I_d$ implies $V_0^{-1} \preceq (\kappa B)^{-1} I_d$. Hence

$$\mathbb{E}_{X\sim\mathbb{P}_X}\left[\phi(X)^\top V_0^{-1}\phi(X)\right] = \mathrm{Tr}\left(V_0^{-1}\Sigma_X\right) \le \mathrm{Tr}\left((\kappa B)^{-1}I_d \cdot \Sigma_X\right) = \frac{\mathrm{Tr}(\Sigma_X)}{\kappa B}.$$

If additionally $\Sigma_X \preceq L^2 I_d$, then $\mathrm{Tr}(\Sigma_X) \le dL^2$, which yields the final scaling in (24) (up to the log factor hidden in $\beta_B(\delta)$). $\qquad\square$

Corollary 4.6 shows that the PEHE risk is governed by an *integrated leverage* term $\mathbb{E}_{\mathbb{P}_X}[\phi(X)^\top V_\lambda^{-1}\phi(X)]$. Active sampling is useful precisely because it can shape $V_\lambda$: selecting points that increase eigenvalues of $V_\lambda$ reduces this term. This provides a direct theoretical rationale for using an uncertainty proxy (Algorithm 1, $v_u$): in linear models, predictive variance is proportional to $\phi(u)^\top V_\lambda^{-1}\phi(u)$, and ensemble disagreement is a practical surrogate for that quantity.

## B.3. Asymptotic normality under adaptive sampling

Finite-sample concentration ensures small error with high probability. For statistical inference (e.g., confidence intervals), we further need a CLT. The non-i.i.d. nature of active sampling prevents a direct appeal to the classical i.i.d. CLT, but a martingale CLT applies.

**Assumption B.9.** (Stabilizing design and moments). As $B \to \infty$, the normalized information matrix converges in probability:

$$\frac{1}{B} \sum_{t=1}^{B} \phi_t \phi_t^\top \xrightarrow{p} \Sigma_\pi, \qquad \Sigma_\pi \succ 0. \tag{46}$$

Moreover, the conditional quadratic variation stabilizes:

$$\frac{1}{B} \sum_{t=1}^{B} \mathbb{E}\big[\varepsilon_t^2 \, \phi_t \phi_t^\top \mid \mathcal{F}_{t-1}\big] \xrightarrow{p} \Omega_\pi, \tag{47}$$

and a Lindeberg condition holds: for every $\eta > 0$,

$$\frac{1}{B} \sum_{t=1}^{B} \mathbb{E}\Big[\varepsilon_t^2 \|\phi_t\|_2^2 \cdot \mathbf{1}\big\{|\varepsilon_t|\|\phi_t\|_2 > \eta\sqrt{B}\big\} \,\Big|\, \mathcal{F}_{t-1}\Big] \xrightarrow{p} 0. \tag{48}$$

**Theorem B.10** (Asymptotic normality (martingale CLT)). *Suppose Assumptions 4.1–4.7 hold and $\lambda = 0$. Then*

$$\sqrt{B}\big(\widehat{\theta}_0 - \theta_\star\big) \xrightarrow{d} \mathcal{N}\big(0, \, \Sigma_\pi^{-1} \Omega_\pi \Sigma_\pi^{-1}\big). \tag{49}$$

*Consequently, for any fixed $x \in \mathcal{X}$,*

$$\sqrt{B}\big(\widehat{\tau}_0(x) - \tau(x)\big) \xrightarrow{d} \mathcal{N}\big(0, \, \phi(x)^\top \Sigma_\pi^{-1} \Omega_\pi \Sigma_\pi^{-1} \phi(x)\big). \tag{50}$$

*Proof.* Recall $M_B = \sum_{t=1}^{B} \phi_t \varepsilon_t$ from (35) and $V_0 = \sum_{t=1}^{B} \phi_t \phi_t^\top$. From (42) with $\lambda = 0$ we have the exact representation

$$\widehat{\theta}_0 - \theta_\star = V_0^{-1} M_B. \tag{51}$$

**Step 1: multivariate martingale CLT for $B^{-1/2} M_B$.** We apply the Cramér–Wold device. Fix any deterministic vector $a \in \mathbb{R}^d$ and define the scalar martingale difference array

$$\xi_{t,B}(a) \triangleq a^\top \phi_t \, \varepsilon_t, \qquad S_B(a) \triangleq \sum_{t=1}^{B} \xi_{t,B}(a) = a^\top M_B.$$

Since $\phi_t$ is $\mathcal{F}_{t-1}$-measurable and $\mathbb{E}[\varepsilon_t \mid \mathcal{F}_{t-1}] = 0$, we have $\mathbb{E}[\xi_{t,B}(a) \mid \mathcal{F}_{t-1}] = 0$. The predictable quadratic variation of $S_B(a)$ is

$$\langle S(a) \rangle_B = \sum_{t=1}^{B} \mathbb{E}\big[\xi_{t,B}(a)^2 \mid \mathcal{F}_{t-1}\big] = \sum_{t=1}^{B} \mathbb{E}\big[\varepsilon_t^2 \, (a^\top \phi_t)^2 \mid \mathcal{F}_{t-1}\big] = a^\top \left(\sum_{t=1}^{B} \mathbb{E}\big[\varepsilon_t^2 \phi_t \phi_t^\top \mid \mathcal{F}_{t-1}\big]\right) a.$$

By Assumption 4.7 (specifically (47)),

$$\frac{1}{B} \langle S(a) \rangle_B \xrightarrow{p} a^\top \Omega_\pi a.$$

Moreover, the Lindeberg condition (48) implies the (scalar) Lindeberg condition for $\xi_{t,B}(a)$: for any $\eta > 0$,

$$\frac{1}{B} \sum_{t=1}^{B} \mathbb{E}\Big[\xi_{t,B}(a)^2 \cdot \mathbf{1}\big\{|\xi_{t,B}(a)| > \eta\sqrt{B}\big\} \,\Big|\, \mathcal{F}_{t-1}\Big] \leq \|a\|_2^2 \cdot \frac{1}{B} \sum_{t=1}^{B} \mathbb{E}\Big[\varepsilon_t^2 \|\phi_t\|_2^2 \cdot \mathbf{1}\big\{|\varepsilon_t|\|\phi_t\|_2 > \eta\sqrt{B}/\|a\|_2\big\} \,\Big|\, \mathcal{F}_{t-1}\Big] \xrightarrow{p} 0.$$

Therefore, by a martingale central limit theorem for triangular arrays (e.g., the Hall–Heyde martingale CLT),

$$\frac{1}{\sqrt{B}} S_B(a) = a^\top \frac{1}{\sqrt{B}} M_B \overset{d}{\Rightarrow} \mathcal{N}\big(0, a^\top \Omega_\pi a\big).$$

Since this holds for every fixed $a$, the Cramér–Wold device yields the multivariate convergence

$$\frac{1}{\sqrt{B}} M_B \overset{d}{\Rightarrow} \mathcal{N}(0, \Omega_\pi). \tag{52}$$

**Step 2: Slutsky with the stabilizing design.** By Assumption 4.7 in (46), we have $B^{-1} V_0 \overset{p}{\to} \Sigma_\pi$ and $\Sigma_\pi \succ 0$. By the continuous mapping theorem, $(B^{-1} V_0)^{-1} \overset{p}{\to} \Sigma_\pi^{-1}$. Now multiply (51) by $\sqrt{B}$:

$$\sqrt{B}(\widehat{\theta}_0 - \theta_\star) = \left(\frac{1}{B} V_0\right)^{-1} \left(\frac{1}{\sqrt{B}} M_B\right).$$

Combine the convergence in probability of $(B^{-1} V_0)^{-1}$ with the distributional convergence (52) and apply Slutsky's theorem to obtain

$$\sqrt{B}(\widehat{\theta}_0 - \theta_\star) \overset{d}{\Rightarrow} \Sigma_\pi^{-1} Z, \qquad Z \sim \mathcal{N}(0, \Omega_\pi),$$

which implies $\Sigma_\pi^{-1} Z \sim \mathcal{N}(0, \Sigma_\pi^{-1} \Omega_\pi \Sigma_\pi^{-1})$.

**Step 3: pointwise CATE normality.** For any fixed $x$, $\widehat{\tau}_0(x) - \tau(x) = \phi(x)^\top (\widehat{\theta}_0 - \theta_\star)$. Apply the continuous mapping theorem to the linear functional $z \mapsto \phi(x)^\top z$ to conclude the second statement. $\qquad\square$

Theorem 4.8 says: *even though the RCT data are collected adaptively and are non-i.i.d., the final estimator still satisfies the usual $\sqrt{B}$-asymptotic normality. The CLT invoked here does not require i.i.d. observations; In this setting, i.i.d. assumptions are sufficient but not necessary. The deeper requirement is the martingale structure created by randomization (mean-zero increments) and a non-degenerate stabilizing design.*

### B.4. Minimax lower bound and (near) optimality

Upper bounds alone do not establish whether a method is optimal. We therefore complement them with a minimax lower bound. The lower bound shows that: *even with perfect adaptivity and access to unlimited observational logs and unlabeled pools, one cannot beat the $\sqrt{d/B}$ scaling in general.*

**A hard instance family.** We consider a discrete covariate space $\{x^{(1)}, \ldots, x^{(d)}\}$ with

$$\mathbb{P}_X(x^{(j)}) = \frac{1}{d}, \qquad j = 1, \ldots, d, \qquad \text{and} \qquad \phi(x^{(j)}) = e_j \in \mathbb{R}^d. \tag{53}$$

Define the parameter set $\Theta_\Delta \triangleq \{-\Delta, +\Delta\}^d$ for some $\Delta \in (0, 1)$. For each $\theta \in \Theta_\Delta$, define potential outcomes (conditionally independent across units and over time) by

$$Y(1) \mid X = x^{(j)} \sim \text{Bern}\left(\frac{1}{2} + \frac{\theta_j}{2}\right), \qquad Y(0) \mid X = x^{(j)} \sim \text{Bern}\left(\frac{1}{2} - \frac{\theta_j}{2}\right). \tag{54}$$

Then $\tau(x^{(j)}) = \theta_j$ and $\|\theta\|_2 = \sqrt{d}\Delta$. (If the function class enforces $\|\theta\|_2 \leq S$, we choose $\Delta \leq S/\sqrt{d}$ so that $\Theta_\Delta \subseteq \{\|\theta\|_2 \leq S\}$.) This family lies entirely within the binary-outcome potential-outcomes model and is compatible with bounded randomization.

**A basic two-point inequality.** We will use the following standard reduction from estimation to testing.

**Lemma B.11** (Two-point lower bound via total variation). *Let $P$ and $Q$ be two probability measures and let $a \neq b$ be two real numbers. For any (possibly randomized) estimator $\widehat{u}$ based on an observation distributed as either $P$ or $Q$,*

$$\max\left\{\mathbb{E}_P\big[(\widehat{u} - a)^2\big], \mathbb{E}_Q\big[(\widehat{u} - b)^2\big]\right\} \geq \frac{(a-b)^2}{8}\big(1 - \text{TV}(P, Q)\big), \tag{55}$$

*where* $\text{TV}(P, Q) \triangleq \sup_A |P(A) - Q(A)|$.

*Proof.* Let $\psi$ be the test $\psi = 1$ if $|\widehat{u} - a| \le |\widehat{u} - b|$ and $\psi = 0$ otherwise. If $\psi = 1$, then $|\widehat{u} - b| \ge |a - b|/2$; if $\psi = 0$, then $|\widehat{u} - a| \ge |a - b|/2$. Hence,

$$(\widehat{u} - a)^2 \ge \frac{(a-b)^2}{4}\mathbf{1}\{\psi = 0\}, \qquad (\widehat{u} - b)^2 \ge \frac{(a-b)^2}{4}\mathbf{1}\{\psi = 1\}.$$

Therefore,

$$\mathbb{E}_P[(\widehat{u} - a)^2] + \mathbb{E}_Q[(\widehat{u} - b)^2] \ge \frac{(a-b)^2}{4}\Big(\mathbb{P}_P(\psi = 0) + \mathbb{P}_Q(\psi = 1)\Big).$$

By the Neyman–Pearson characterization of total variation, for any test $\psi$, $\mathbb{P}_P(\psi = 0) + \mathbb{P}_Q(\psi = 1) \ge 1 - \mathrm{TV}(P, Q)$. Thus

$$\mathbb{E}_P[(\widehat{u} - a)^2] + \mathbb{E}_Q[(\widehat{u} - b)^2] \ge \frac{(a-b)^2}{4}\Big(1 - \mathrm{TV}(P, Q)\Big).$$

Taking the maximum of the two terms on the left-hand side yields (55). $\qquad\square$

**A chain rule for KL under adaptive designs.** Let $Z_{1:B}$ denote the entire observed data stream generated by a fixed adaptive policy $\pi$: $Z_t$ includes $(X_t, p_t, T_t, Y_t)$ (and can include any extra bookkeeping variables). Write $P_\theta^\pi$ for the induced law of $Z_{1:B}$ under parameter $\theta$ and policy $\pi$.

**Lemma B.12** (Sequential chain rule for KL under a fixed policy). *For any $\theta, \theta' \in \Theta_\Delta$ and any fixed policy $\pi$,*

$$\mathrm{KL}(P_\theta^\pi \| P_{\theta'}^\pi) = \sum_{t=1}^{B} \mathbb{E}_\theta^\pi \big[\mathrm{KL}\big(P_\theta^\pi(Z_t \mid \mathcal{F}_{t-1}) \,\big\|\, P_{\theta'}^\pi(Z_t \mid \mathcal{F}_{t-1})\big)\big]. \tag{56}$$

*Proof.* This is the standard KL chain rule for a joint law written as a product of conditionals. Because the policy $\pi$ is fixed, both $P_\theta^\pi$ and $P_{\theta'}^\pi$ admit the same filtration $\{\mathcal{F}_t\}$, and their joint densities (w.r.t. a suitable product dominating measure) factorize as $p_\theta(z_{1:B}) = \prod_{t=1}^{B} p_\theta(z_t \mid z_{1:t-1})$ and similarly for $\theta'$. Then

$$\begin{aligned}
\mathrm{KL}(P_\theta^\pi \| P_{\theta'}^\pi) &= \mathbb{E}_\theta^\pi\left[\log\frac{p_\theta(Z_{1:B})}{p_{\theta'}(Z_{1:B})}\right] = \mathbb{E}_\theta^\pi\left[\sum_{t=1}^{B}\log\frac{p_\theta(Z_t \mid Z_{1:t-1})}{p_{\theta'}(Z_t \mid Z_{1:t-1})}\right] \\
&= \sum_{t=1}^{B}\mathbb{E}_\theta^\pi\left[\mathbb{E}_\theta^\pi\left[\log\frac{p_\theta(Z_t \mid \mathcal{F}_{t-1})}{p_{\theta'}(Z_t \mid \mathcal{F}_{t-1})}\,\bigg|\,\mathcal{F}_{t-1}\right]\right] \\
&= \sum_{t=1}^{B}\mathbb{E}_\theta^\pi\big[\mathrm{KL}\big(P_\theta^\pi(Z_t \mid \mathcal{F}_{t-1}) \,\big\|\, P_{\theta'}^\pi(Z_t \mid \mathcal{F}_{t-1})\big)\big],
\end{aligned}$$

which is (56). $\qquad\square$

**Theorem B.13** (Minimax lower bound for active RCT (rate optimality)). *Consider the hard covariate construction in (53). Suppose that $B \ge c_0 d$ and $S \ge c_S d/\sqrt{B}$ for sufficiently large universal constants $c_0, c_S > 0$. Then there exists a hypercube subfamily $\Theta_\Delta \subseteq \mathcal{F}_{\mathrm{lin}}(S)$ of the form (54), with $\Delta^2 \asymp d/B$, such that for any adaptive querying policy $\pi$ possibly using $D_{\mathrm{obs}}$ and $D_{\mathrm{pool}}$, and any estimator $\widehat{\tau}$ based on $B$ RCT samples,*

$$\inf_\pi \inf_{\widehat{\tau}} \sup_{\tau \in \mathcal{F}_{\mathrm{lin}}(S)} \big(\mathbb{E}_\tau^\pi\big[R(\widehat{\tau})^2\big]\big)^{1/2} \ge c_1\sqrt{\frac{d}{B}},$$

*for a universal constant $c_1 > 0$. In particular, the $\sqrt{d/B}$ scaling in Corollary 4.6 is information-theoretically unimprovable in root-mean-square PEHE in this regime.*

*Proof.* We prove a lower bound on the *expected squared* PEHE risk and then take square roots.

**Step 1: reduce minimax risk to a Bayes risk over a hypercube prior.** Let $\Pi$ be the uniform prior over $\Theta_\Delta = \{-\Delta, +\Delta\}^d$. For any policy $\pi$ and estimator $\widehat{\tau}$, by Jensen and the fact that $\sup_\theta \ge \mathbb{E}_{\theta\sim\Pi}$,

$$\sup_{\theta\in\Theta_\Delta} \mathbb{E}_\theta^\pi\big[\mathcal{R}(\widehat{\tau})^2\big] \ge \mathbb{E}_{\theta\sim\Pi}\mathbb{E}_\theta^\pi\big[\mathcal{R}(\widehat{\tau})^2\big].$$

Thus it suffices to lower bound the Bayes risk under $\Pi$ uniformly over $\pi$ and $\widehat{\tau}$.

Under (53) and $\phi(x^{(j)}) = e_j$, the PEHE squared is

$$\mathcal{R}(\widehat{\tau})^2 = \mathbb{E}_{X \sim \mathbb{P}_X}\left[(\widehat{\tau}(X) - \tau(X))^2\right] = \frac{1}{d}\sum_{j=1}^{d}\left(\widehat{\tau}(x^{(j)}) - \theta_j\right)^2.$$

Define $\widehat{\theta}_j \triangleq \widehat{\tau}(x^{(j)})$. Then

$$\mathcal{R}(\widehat{\tau})^2 = \frac{1}{d}\sum_{j=1}^{d}(\widehat{\theta}_j - \theta_j)^2. \tag{57}$$

**Step 2: apply the two-point bound coordinate-wise (Assouad-style).** For each $j$, define the "flipped" parameter $\theta^{(j)}$ by $\theta_j^{(j)} = -\theta_j$ and $\theta_\ell^{(j)} = \theta_\ell$ for $\ell \neq j$. Let $P_\theta^\pi$ and $P_{\theta^{(j)}}^\pi$ be the laws of the entire adaptive data stream under these two parameters.

Apply Lemma B.11 with $a = \Delta$ and $b = -\Delta$ to the estimation of coordinate $j$, and then average over the uniform prior on $\theta$. A standard symmetrization argument yields

$$\mathbb{E}_{\theta \sim \Pi}\mathbb{E}_\theta^\pi\left[(\widehat{\theta}_j - \theta_j)^2\right] \geq \frac{\Delta^2}{8}\left(1 - \mathbb{E}_{\theta \sim \Pi}\left[\mathrm{TV}\left(P_\theta^\pi, P_{\theta^{(j)}}^\pi\right)\right]\right). \tag{58}$$

Summing (58) over $j = 1, \ldots, d$ and dividing by $d$, and using (57), we get

$$\mathbb{E}_{\theta \sim \Pi}\mathbb{E}_\theta^\pi\left[\mathcal{R}(\widehat{\tau})^2\right] \geq \frac{\Delta^2}{8}\left(1 - \frac{1}{d}\sum_{j=1}^{d}\mathbb{E}_{\theta \sim \Pi}\left[\mathrm{TV}\left(P_\theta^\pi, P_{\theta^{(j)}}^\pi\right)\right]\right). \tag{59}$$

**Step 3: bound total variation by KL and control KL under adaptivity.** By Pinsker's inequality, $\mathrm{TV}(P, Q) \leq \sqrt{\mathrm{KL}(P\|Q)/2}$. Thus

$$\frac{1}{d}\sum_{j=1}^{d}\mathbb{E}_{\theta \sim \Pi}\left[\mathrm{TV}\left(P_\theta^\pi, P_{\theta^{(j)}}^\pi\right)\right] \leq \frac{1}{d}\sum_{j=1}^{d}\mathbb{E}_{\theta \sim \Pi}\left[\sqrt{\frac{1}{2}\mathrm{KL}\left(P_\theta^\pi \,\|\, P_{\theta^{(j)}}^\pi\right)}\right].$$

Since $x \mapsto \sqrt{x}$ is concave on $\mathbb{R}_+$, Jensen gives

$$\frac{1}{d}\sum_{j=1}^{d}\mathbb{E}_{\theta \sim \Pi}\left[\sqrt{\frac{1}{2}\mathrm{KL}\left(P_\theta^\pi \,\|\, P_{\theta^{(j)}}^\pi\right)}\right] \leq \sqrt{\frac{1}{2d}\sum_{j=1}^{d}\mathbb{E}_{\theta \sim \Pi}\left[\mathrm{KL}\left(P_\theta^\pi \,\|\, P_{\theta^{(j)}}^\pi\right)\right]}. \tag{60}$$

It remains to bound the average KL. Fix $j$ and apply Lemma B.12. Because $\theta$ and $\theta^{(j)}$ differ only at coordinate $j$, the conditional distributions of $Y_t$ coincide whenever $X_t \neq x^{(j)}$. Hence the conditional KL contribution at time $t$ is zero if $X_t \neq x^{(j)}$. If $X_t = x^{(j)}$, then the conditional distribution of $(T_t, Y_t)$ differs only through the Bernoulli parameters $(1/2 \pm \Delta/2)$ in (54). A direct computation (or the standard Bernoulli KL bound) shows that for $\Delta \leq 1/2$ there exists a universal constant $C_{\mathrm{KL}} > 0$ such that

$$\mathrm{KL}\left(\mathrm{Bern}\left(\frac{1}{2} + \frac{\Delta}{2}\right) \,\Big\|\, \mathrm{Bern}\left(\frac{1}{2} - \frac{\Delta}{2}\right)\right) \leq C_{\mathrm{KL}}\Delta^2, \qquad \mathrm{KL}\left(\mathrm{Bern}\left(\frac{1}{2} - \frac{\Delta}{2}\right) \,\Big\|\, \mathrm{Bern}\left(\frac{1}{2} + \frac{\Delta}{2}\right)\right) \leq C_{\mathrm{KL}}\Delta^2. \tag{61}$$

(For instance, one may take $C_{\mathrm{KL}} = 16/3$ since the Bernoulli parameters lie in $[1/4, 3/4]$ and $\mathrm{KL}(\mathrm{Bern}(p)\|\mathrm{Bern}(q)) \leq (p - q)^2/(q(1 - q))$.)

Therefore, the one-step KL between the conditional laws of $(T_t, Y_t)$ under $\theta$ and $\theta^{(j)}$ is at most $C_{\mathrm{KL}}\Delta^2$ whenever $X_t = x^{(j)}$. Let $N_j \triangleq \sum_{t=1}^{B}\mathbf{1}\{X_t = x^{(j)}\}$ be the (random) number of times type $j$ is queried. Combining with the chain rule (56) yields

$$\mathrm{KL}\left(P_\theta^\pi \,\|\, P_{\theta^{(j)}}^\pi\right) \leq C_{\mathrm{KL}}\Delta^2\,\mathbb{E}_\theta^\pi[N_j]. \tag{62}$$

Now average (62) over $\theta \sim \Pi$ and sum over $j$: since $\sum_{j=1}^{d} N_j = B$ deterministically (exactly one $X_t$ is queried per round), we have for every $\theta$, $\sum_{j=1}^{d} \mathbb{E}_{\theta}^{\pi}[N_j] = B$, and thus

$$\frac{1}{d} \sum_{j=1}^{d} \mathbb{E}_{\theta \sim \Pi} \left[ \mathrm{KL}\big(P_\theta^\pi \,\|\, P_{\theta^{(j)}}^\pi\big) \right] \;\leq\; \frac{C_{\mathrm{KL}}\Delta^2}{d} \sum_{j=1}^{d} \mathbb{E}_{\theta \sim \Pi}\mathbb{E}_{\theta}^{\pi}[N_j] = C_{\mathrm{KL}}\Delta^2 \cdot \frac{B}{d}. \tag{63}$$

Plugging (63) into (60) gives

$$\frac{1}{d} \sum_{j=1}^{d} \mathbb{E}_{\theta \sim \Pi} \left[ \mathrm{TV}\big(P_\theta^\pi, P_{\theta^{(j)}}^\pi\big) \right] \;\leq\; \sqrt{\frac{C_{\mathrm{KL}}\Delta^2 B}{2d}}. \tag{64}$$

**Step 4: choose $\Delta$.** Choose $\Delta^2 = \frac{d}{16 C_{\mathrm{KL}} B}$. Since $B \geq c_0 d$ for a sufficiently large universal constant $c_0$, we have $\Delta \leq 1/2$. Since $S \geq c_S d/\sqrt{B}$ for a sufficiently large universal constant $c_S$, we also have $\sqrt{d}\,\Delta \leq S$. Hence $\Theta_\Delta \subseteq \mathcal{F}_{\mathrm{lin}}(S)$.

With this choice of $\Delta$, it gives

$$\frac{1}{d} \sum_{j=1}^{d} \mathbb{E}_{\theta \sim \Pi} \left[ \mathrm{TV}\big(P_\theta^\pi, P_{\theta^{(j)}}^\pi\big) \right] \leq \sqrt{\frac{C_{\mathrm{KL}}\Delta^2 B}{2d}} \leq \frac{1}{4},$$

after adjusting constants if necessary. Plugging this bound into the above yields

$$\mathbb{E}_{\theta \sim \Pi}\mathbb{E}_{\theta}^{\pi} \left[ R(\widehat{\tau})^2 \right] \geq \frac{\Delta^2}{8} \left( 1 - \frac{1}{4} \right) \geq c\,\frac{d}{B}$$

for a universal constant $c > 0$.

Since the supremum over $\theta \in \Theta_\Delta$ is at least the Bayes average under the uniform prior $\Pi$, and since $\Theta_\Delta \subseteq \mathcal{F}_{\mathrm{lin}}(S)$, we obtain for arbitrary $\pi$ and $\widehat{\tau}$,

$$\sup_{\tau \in \mathcal{F}_{\mathrm{lin}}(S)} \mathbb{E}_{\tau}^{\pi} \left[ R(\widehat{\tau})^2 \right] \geq c\,\frac{d}{B}.$$

Taking the infimum over $\pi$ and $\widehat{\tau}$, and then taking square roots by monotonicity of the square-root map, gives

$$\inf_{\pi} \inf_{\widehat{\tau}} \sup_{\tau \in \mathcal{F}_{\mathrm{lin}}(S)} \left( \mathbb{E}_{\tau}^{\pi} \left[ R(\widehat{\tau})^2 \right] \right)^{1/2} \geq c_1 \sqrt{\frac{d}{B}}.$$

This proves the claimed root-mean-square PEHE lower bound.

$\square$

Theorem 4.9 formalizes a simple but important reality: if there are $d$ independent degrees of freedom in $\tau(\cdot)$, then with budget $B$, no adaptive design can achieve a root-mean-square PEHE rate faster than order $\sqrt{d/B}$ in general.

**Corollary B.14** (Near-minimax optimality of the orthogonalized estimator)**.** *Under Assumptions 4.1–4.3, a well-conditioned design $V_0 \succeq \kappa B I_d$, and the nontrivial radius regime $S \gtrsim d/\sqrt{B}$, the upper bound in (26) matches the root-mean-square minimax lower bound in Theorem B.13 up to logarithmic factors and the conditioning constant $\kappa$. In this sense, the estimator in Algorithm 2 is minimax-rate optimal, up to logarithmic factors, for the linear class.*

*Proof.* Corollary 4.6 gives an upper bound of order $\sqrt{d/B}$, up to logarithmic factors and the conditioning constant $\kappa$, under a well-conditioned queried design. Theorem B.13 shows that no adaptive querying policy and no estimator can improve the $\sqrt{d/B}$ scaling in root-mean-square PEHE over $\mathcal{F}_{\mathrm{lin}}(S)$ in the nontrivial radius regime $S \gtrsim d/\sqrt{B}$. Combining the two results gives the claimed near-minimax optimality. $\square$

## B.5. How the main results connect

We close the theory section by clarifying the logical flow:

- Lemma 4.4 is the foundation: it isolates the causal "truth" created by randomization, even under adaptive selection.

- Lemmas B.5–B.6 provide the technical engine: self-normalized martingale concentration that remains valid for non-i.i.d. adaptive designs.

- Theorem 4.5 is the strongest finite-sample guarantee: a deviation bound controlled by the information matrix $V_\lambda$.

- Corollary 4.6 translates this into the paper's target metric (PEHE risk), revealing the central role of integrated leverage.

- Theorem 4.8 upgrades consistency to inference: asymptotic normality under a stabilizing adaptive design.

- Theorem 4.9 provides the impossibility result: in root-mean-square PEHE, no method can beat the $\sqrt{d/B}$ scaling in general in the nontrivial radius regime $S \gtrsim d/\sqrt{B}$.

- Corollary 4.10 combines the finite-sample upper bound and the root-mean-square minimax lower bound to conclude near-minimax optimality up to logarithmic factors and conditioning constants.

**Choosing the randomization probability.** If operational constraints permit, one may ask how to choose $p$ to minimize the pseudo-outcome variance. The exact variance-optimal choice depends on second moments of potential outcomes:

**Proposition B.15** (Variance-optimal randomization for the pseudo-outcome)**.** *Fix $x$ and denote $A(x) \triangleq \mathbb{E}[Y(1)^2 \mid X = x]$ and $B(x) \triangleq \mathbb{E}[Y(0)^2 \mid X = x]$. Under Assumption 4.2, the conditional variance $\mathrm{Var}(\widetilde{Y} \mid X = x, p)$ is minimized at*

$$p^\star(x) \;=\; \frac{\sqrt{A(x)}}{\sqrt{A(x)} + \sqrt{B(x)}}. \tag{65}$$

*In particular, if $A(x) = B(x)$ (e.g., comparable outcome scales across arms), then $p^\star(x) = 1/2$.*

*Proof.* Condition on $X = x$ and $p \in (0, 1)$. Using the definition of pseudo outcome and the randomization $T \sim \mathrm{Bern}(p)$,

$$\widetilde{Y} = \begin{cases} Y(1)/p, & T = 1, \\ -Y(0)/(1-p), & T = 0. \end{cases}$$

Hence

$$\mathbb{E}[\widetilde{Y}^2 \mid X = x, p] = p \cdot \mathbb{E}\left[\frac{Y(1)^2}{p^2} \,\middle|\, X = x\right] + (1-p) \cdot \mathbb{E}\left[\frac{Y(0)^2}{(1-p)^2} \,\middle|\, X = x\right] = \frac{A(x)}{p} + \frac{B(x)}{1-p}.$$

Since $\mathbb{E}[\widetilde{Y} \mid X = x, p] = \tau(x)$ does not depend on $p$ (Lemma 4.4), minimizing $\mathrm{Var}(\widetilde{Y} \mid X = x, p)$ over $p$ is equivalent to minimizing $f(p) = A(x)/p + B(x)/(1-p)$ over $p \in (0, 1)$. This is a strictly convex function on $(0, 1)$ with derivative $f'(p) = -A(x)/p^2 + B(x)/(1-p)^2$. Setting $f'(p) = 0$ yields $(1-p)/p = \sqrt{B(x)/A(x)}$, which rearranges to (65). The special case $A(x) = B(x)$ gives $p^\star(x) = 1/2$. $\square$

Proposition B.15 justifies $p \approx 1/2$ as a robust default when outcome scales are comparable across arms, and motivates clipping $[f_{\min}, f_{\max}]$ as a principled safeguard when $p$ must vary by covariates.

# C. Real-world Industrial Experiment Details

In the real-world industrial experiments, we further *extend the proposed active experimentation strategy to a production-oriented hybrid neural estimator*. Specifically, we train a neural CATE estimator using both the observational data and the actively collected randomized data:

$$\hat{\tau}_\pi^{\mathrm{hyb}} = \mathcal{A}_{\mathrm{nn}}\left(\mathcal{D}_{\mathrm{obs}} \cup \mathcal{D}_{\mathrm{rct}}^{(K)}; w_i\right). \tag{66}$$

Unlike the estimator $\hat{\tau}_\pi$ used in the theoretical analysis, this hybrid version is designed to **better match large-scale industrial settings**, where flexible neural uplift models are commonly used for CATE estimation. The basic idea is to

initialize the model using abundant OBS data and then iteratively augment its training set with RCT samples selected by the proposed active experimentation strategy. As more randomized data are collected, the neural estimator is repeatedly updated, allowing the RCT data to progressively **correct observational bias** and **strengthen model performance** in regions where OBS provides limited causal support. Here, $\mathcal{D}_{\mathrm{rct}}^{(K)}$ denotes the randomized data collected after $K$ active experimentation rounds, and $w_i$ denotes the alignment weight assigned to sample $i$, which gives higher importance to randomized samples that repair observational blind spots.

## C.1. Industry-Oriented Active Learning Extension

---

**Algorithm 3** Industry-Oriented Active Learning for OBS–RCT Fusion

---

**Require:** Observational data $\mathcal{D}_{\mathrm{obs}}$, unlabeled pool $\mathcal{D}_{\mathrm{pool}}$, maximum query batch size $M$, budget $B$, maximum rounds $K$.
**Ensure:** Final CATE estimator $\widehat{\tau}_\pi^{\mathrm{hyb}}$.
 1: **Initialize:** $\mathcal{D}_{\mathrm{rct}} \leftarrow \emptyset$, $k \leftarrow 1$.
 2: **Representation learning:** Train feature encoder $\phi : \mathcal{X} \to \mathcal{H}$ on $\mathcal{D}_{\mathrm{obs}} \cup \mathcal{D}_{\mathrm{pool}}$.
 3: **while** $|\mathcal{D}_{\mathrm{rct}}| < B$ and $k \leq K$ **do**
 4:     $m_k \leftarrow \min(M,\ B - |\mathcal{D}_{\mathrm{rct}}|)$.
 5:     **1. Update scoring components.**
 6:     Train domain classifier $g_\xi(\phi(x))$ to discriminate $\mathcal{D}_{\mathrm{pool}}$ from $\mathcal{D}_{\mathrm{obs}} \cup \mathcal{D}_{\mathrm{rct}}$.
 7:     Update CATE prediction ensemble $\mathcal{M} = \{\widehat{\tau}_j(x)\}_{j=1}^E$.          {$E$: number of stochastic passes}
 8:     Estimate observational propensity

$$\widehat{e}_{\mathrm{obs}}(\phi(x)) \approx \mathbb{P}_{\mathrm{obs}}\Big(T^{\mathrm{obs}} = 1 \mid \phi(x)\Big).$$

 9:     **2. Acquisition scoring.**
10:     **for** candidate $u \in \mathcal{D}_{\mathrm{pool}}$ **do**
11:         $v_u \leftarrow \mathrm{Var}\big(\{\widehat{\tau}_j(\phi(u))\}_{j=1}^E\big)$.              {model uncertainty}
12:         $d_u \leftarrow \sigma(g_\xi(\phi(u)))$.                  {domain discrepancy}
13:         $o_u \leftarrow 2\,|\widehat{e}_{\mathrm{obs}}(\phi(u)) - 0.5|$.             {observational imbalance}
14:

$$S(u) \leftarrow \alpha\,\eta(v_u) + \beta\,\eta(d_u) + \gamma\,\eta(o_u),$$

                    {$\eta(\cdot)$: normalization function}

15:     **end for**
16:     **3. Selection and experimentation.**
17:     Select candidates

$$\mathcal{C}_k \leftarrow \text{Top-}m_k\big(S(\cdot), \mathcal{D}_{\mathrm{pool}}\big).$$

18:     **for** $X_i \in \mathcal{C}_k$ **do**
19:         Perform RCT: assign $T_i^{\mathrm{rct}} \sim \mathrm{Bern}(p_k(X_i))$ and observe $Y_i^{\mathrm{rct}}$.
20:         $\mathcal{D}_{\mathrm{rct}} \leftarrow \mathcal{D}_{\mathrm{rct}} \cup \{(X_i, T_i^{\mathrm{rct}}, Y_i^{\mathrm{rct}}, p_k(X_i))\}$.
21:     **end for**
22:     **4. Update pool and round index.**
23:     $\mathcal{D}_{\mathrm{pool}} \leftarrow \mathcal{D}_{\mathrm{pool}} \setminus \mathcal{C}_k$,   $k \leftarrow k + 1$.
24: **end while**
25: **Return** $\widehat{\tau}_\pi^{\mathrm{hyb}}$ trained on $\mathcal{D}_{\mathrm{obs}} \cup \mathcal{D}_{\mathrm{rct}}$.

---

## C.2. Detailed Design of the Acquisition Strategy

In this section, we provide a rigorous formulation of our proposed active learning framework, specifically designed to enhance the Disentangled Representations for CounterFactual Regression (DRCFR) model. We adopt DRCFR as the foundational backbone not merely for its theoretical properties, but due to its proven efficacy as our *deployed production engine*, where it demonstrates superior stability in uplift estimation across diverse industrial scenarios. Building on this robust baseline, our methodology integrates two core components: a multi-faceted acquisition function for strategic sample selection and a reverse alignment weighting mechanism for model optimization.

### C.2.1. MULTI-FACETED ACQUISITION FUNCTION

To select the most informative samples from the unlabeled candidate pool $\mathcal{D}_{\mathrm{pool}}$, we design a composite acquisition function that addresses three challenges in causal inference: *epistemic uncertainty*, *domain discrepancy*, and *overlap deficit*. For any candidate sample $u \in \mathcal{D}_{\mathrm{pool}}$, the total acquisition score $S(u)$ is defined as:

$$S(u) = \alpha \cdot \eta(v_u) + \beta \cdot \eta(d_u) + \gamma \cdot \eta(o_u), \tag{67}$$

where $\alpha, \beta, \gamma$ are hyperparameters balancing the trade-off between uncertainty reduction, domain discrepancy, and overlap deficit. We detail each component below.

**1. Epistemic Uncertainty Score ($v_u$).**    To measure the model's lack of knowledge regarding the Conditional Average Treatment Effect (CATE), we employ Monte Carlo (MC) Dropout (Gal & Ghahramani, 2016) as a Bayesian approximation. Unlike standard output variance, we focus specifically on the variance of the predicted *uplift*. Let $f_\theta$ denote the inference network with dropout enabled. We perform $E$ stochastic forward passes for each sample $u$, obtaining a set of CATE predictions $\{\hat{\tau}_j(u)\}_{j=1}^E$. The uncertainty score is quantified as the variance of these predictions:

$$v_u = \mathrm{Var}\left(\{\hat{\tau}_j(u)\}_{j=1}^E\right) = \frac{1}{E}\sum_{j=1}^E \left(\hat{\tau}_j(u) - \bar{\tau}(u)\right)^2, \tag{68}$$

where $\bar{\tau}(u)$ is the empirical mean. A higher variance implies high epistemic uncertainty, suggesting that observing a randomized outcome for $u$ would yield significant information gain.

**2. Domain Discrepancy Score ($d_u$).**    To ensure the selected samples effectively cover the feature space of the target pool (exploration) and mitigate domain shift, we introduce a Domain Classifier. This module, parameterized by $\xi$, discriminates between the *current* training distribution ($\mathcal{D}_{\mathrm{current}} = \mathcal{D}_{\mathrm{obs}} \cup \mathcal{D}_{\mathrm{rct}}$) and the unselected pool ($\mathcal{D}_{\mathrm{target}} = \mathcal{D}_{\mathrm{pool}}$).

Crucially, the classifier operates on the disentangled representation $\phi(u) \in \mathbb{R}^{d_{rep}}$ extracted from the DRCFR backbone, rather than raw features. Structurally, $g_\xi$ is designed as a two-layer Multilayer Perceptron (MLP). It consists of a linear transformation to a hidden dimension $d_{hidden}$, followed by an Exponential Linear Unit (ELU) activation and a Dropout layer. The final layer maps the hidden features to a scalar logit. Formally, the forward pass for a sample $u$ is defined as:

$$\mathbf{h} = \mathrm{Dropout}(\mathrm{ELU}(\mathbf{W}_1\phi(u) + \mathbf{b}_1)). \tag{69}$$

$$g_\xi(\phi(u)) = \mathbf{W}_2\mathbf{h} + b_2. \tag{70}$$

where $\mathbf{W}_1$ and $\mathbf{W}_2$ are learnable weights. The discrepancy score is then defined as the predicted probability of belonging to the target pool (label 1):

$$d_u = \mathbb{P}(\mathrm{domain} = 1 \mid \phi(u); \xi) = \sigma(g_\xi(\phi(u))) \tag{71}$$

where $\sigma(\cdot)$ is the sigmoid function. A higher $d_u$ indicates that $u$ lies in a region under-represented by the current training set, necessitating active sampling for distribution alignment.

**3. Overlap Deficit Score ($o_u$).**    To explicitly target samples heavily affected by selection bias in the observational data, we define a metric based on propensity extremity. We utilize a propensity head $\hat{e}_{\mathrm{obs}}(\phi(u))$, pre-trained strictly on $\mathcal{D}_{\mathrm{obs}}$, to capture the historical assignment mechanism. The score targets samples where the observational policy is deterministic (i.e., propensity close to 0 or 1):

$$o_u = |\hat{e}_{\mathrm{obs}}(\phi(u)) - 0.5| \times 2. \tag{72}$$

Samples with $o_u \approx 1$ represent individuals who, historically, almost exclusively received one specific treatment (either control or treated). Querying these samples in an RCT context (where $e(\phi(u)) = 0.5$) provides counterfactual evidence to correct the learned bias.

C.2.2. OPTIMIZATION VIA COUNTERFACTUAL ALIGNMENT AND SAMPLE REWEIGHTING

After selecting and labeling a batch of samples to form $\mathcal{D}_{\mathrm{rct}}$, we add these samples to $\mathcal{D}_{\mathrm{obs}}$ and retrain the uplift models. Directly pooling the two datasets does not make full use of the RCT samples. We therefore use a **counterfactual alignment** strategy, which assigns higher weights to instances that conflict with the bias in the observational data. This update places greater emphasis on counterfactual evidence that is underrepresented in $\mathcal{D}_{\mathrm{obs}}$.

We define the *Propensity-Assignment Gap*, $\Delta(u,t)$, as the absolute divergence between the realized randomized treatment assignment $t_{\mathrm{rct}}$ and the historical observational propensity $\hat{e}_{\mathrm{obs}}(\phi(u))$:

$$\Delta(u, t_{\mathrm{rct}}) = |t_{\mathrm{rct}} - \hat{e}_{\mathrm{obs}}(\phi(u))|. \tag{73}$$

A large gap (e.g., $\Delta > 0.5$) signifies that the RCT assignment acts as a counter-balance to the observational tendency. For instance, assigning control ($t = 0$) to a user who historically had a high probability of being treated ($\hat{e}_{\mathrm{obs}} \approx 1$) exposes outcomes in a region of the joint distribution that was previously a blind spot.

To focus model training on these information-rich samples and debias the uplift estimator under a finite RCT budget, we apply the following weights $w_i$ to the loss function:

$$w_i = \begin{cases} 1.0, & \text{if } \Delta(u_i, t_i) > 0.5 \quad \text{(Counterfactual-aligned sample)}. \\ 0.2, & \text{otherwise} \quad \text{(Lower-gap complementary sample)}. \end{cases} \tag{74}$$

**Motivation**. Samples with larger counterfactual-alignment gaps provide information that is largely missing from $\mathcal{D}_{\mathrm{obs}}$. These samples are therefore given higher weights. Samples with smaller gaps are closer to the historical assignment pattern and mainly reinforce information already present in the observational data. We assign them lower weights so that the gradient updates depend more on the counterfactual-alignment signal.

Taking DRCFR as an example, the total training objective incorporates these alignment weights:

$$\mathcal{L}_{DRCFR} = \sum_{i \in \mathcal{D}_{\mathrm{current}}} w_i \cdot \ell_{\mathrm{pred}}(y_i, \hat{y}_i) + \lambda_{\mathrm{disc}} \cdot \mathrm{MMD}(\phi(u)_{|t=0}, \phi(u)_{|t=1}) + \lambda_\pi \cdot \mathcal{L}_{\mathrm{prop}}. \tag{75}$$

where the first term is the weighted factual prediction loss, the second term enforces representation balance via Maximum Mean Discrepancy (MMD), and the third term is the propensity regularization. In our experiments, we set the regularization coefficients $\lambda_{\mathrm{disc}} = \lambda_\pi = 1.0$.

### C.3. Detailed Description of the Dataset

Our empirical evaluation utilizes a large-scale *real-world industrial dataset* derived from a ride-hailing platform. The dataset is collected at the order level, where each instance corresponds to an individual order characterized by 468 features. Formally, let $T \in \{0, 1\}$ denote the treatment indicator, where $T = 1$ means that a "free upgrade" service was triggered for the passenger, and let $Y \in \{0, 1\}$ denote the binary outcome indicating whether the order was completed.

To comprehensively assess the proposed active learning framework under extreme distribution shifts, realistic deployment scenarios, and temporal variations, we organize the data into distinct components following a strict **Out-of-Time (OOT) evaluation protocol** spanning 44 days, from Nov. 25 to Jan. 07. During the initial 30-day phase, we construct the RCT Pool $\mathcal{D}_{\mathrm{pool}}$ and concurrently collect observational samples to form the training set, denoted as either $\mathcal{D}_{\mathrm{obs}}^{\mathrm{bias}}$ or $\mathcal{D}_{\mathrm{obs}}^{\mathrm{full}}$ depending on the observational sampling strategy. Generalization is then strictly evaluated on the Unbiased RCT Test Set $\mathcal{D}_{\mathrm{rct}}^{\mathrm{test}}$, which consists of RCT data collected during the remaining 14 days.

- **Biased Observational Set** ($\mathcal{D}_{\mathrm{obs}}^{\mathrm{bias}}$)**:** Collected from the training phase (**Nov. 25 – Dec. 25**), we curate a subset of approximately 4.5 million historical OBS logs. Generated by deterministic production policies, this dataset exhibits severe selection bias and violations of the *overlap assumption*. We utilize this dataset for: (i) comparative analysis between active sampling strategies and random baselines, and (ii) ablation studies on our proposed acquisition functions to evaluate robustness against sample imbalance.

- **Full-Scale Observational Set** ($\mathcal{D}_{\text{obs}}^{\text{full}}$)**:** To validate the framework's efficacy in a real business environment, we employ a larger dataset $\mathcal{D}_{\text{obs}}^{\text{full}}$ of approximately 6.5 million logs from the same training period (**Nov. 25 – Dec. 25**). Compared to $\mathcal{D}_{\text{obs}}^{\text{bias}}$, it shows much lower distributional shift and sample imbalance. Experiments on this set focus on evaluating performance sensitivity across varying ratios of OBS data.

- **RCT Candidate Pool** ($\mathcal{D}_{\text{pool}}$)**:** Coinciding with the training phase (**Nov. 25 – Dec. 25**), this dataset comprises approximately 1.2 million samples representing the candidate pool for RCTs. Under our experimental setting, only covariate information $X$ is available beforehand, while treatment $T$ and outcome $Y$ remain *masked* until explicitly queried. The active learner selectively queries labels subject to budgets $B \in \{10\text{k}, 50\text{k}, \dots, 500\text{k}\}$ to rectify covariate distribution shifts, yielding the labeled dataset $\mathcal{D}_{\text{rct}}$.

- **RCT Test Set** ($\mathcal{D}_{\text{rct}}^{\text{test}}$)**:** To evaluate generalization capability beyond historical biases, we employ a dedicated test set of approximately 1.3 million samples collected from the future period (**Dec. 26 – Jan. 07**). Unlike the training data, this set consists exclusively of randomized experimental data, representing an unbiased sample of the full population's covariate distribution.

### C.4. Training Settings

**Hyperparameter Settings.** In our real-world setting, excessive training epochs often lead to overfitting or result in the model learning to distinguish between the treatment and control groups rather than capturing the true uplift signal. Thus, we train all models for **5 epochs** to ensure robustness and use the **Adam** optimizer with **learning rate 0.001** and **batch size 512**. We detail the other hyperparameter settings used in our experiments in Table 2.

*Table 2.* Hyperparameter specifications for the proposed active learning framework.

| Parameter | Value | Description |
|---|---|---|
| ***Acquisition Function*** | | |
| $\alpha$ (Uncertainty weight) | 0.5 | Controls epistemic uncertainty $v_u$ |
| $\beta$ (Discrepancy weight) | 1.0 | Controls distributional discrepancy $d_u$ |
| $\gamma$ (Bias weight) | 0.7 | Controls propensity-based bias $o_u$ |
| $K$ (MC-Dropout Passes) | 15 | Stochastic passes for uncertainty estimation |
| $M$ (Acquisition Batch Size) | $\{2\text{k}, 10\text{k}, 20\text{k}, 60\text{k}, 100\text{k}\}$ | Number of queried samples per round |
| ***Reverse Alignment Weighting*** | | |
| $w_{\text{gold}}$ | 1.0 | Weight for high-gap counterfactual samples |
| $w_{\text{silver}}$ | 0.2 | Weight for low-gap counterfactual samples |
| $w_{\text{obs}}$ | 0.3 | Weight for observational training samples |
| ***Domain Classifier & Optimization*** | | |
| Representation Dim ($\mathbb{R}^{d_{rep}}$) | 128 | Dimension of shared representation $\phi(u)$ |
| Hidden Width | 64 | Hidden-layer size of the classifier |
| Dropout Rate | 0.2 | Dropout for main and domain models |
| Domain Classifier Learning Rate | 0.001 | Learning rate for classifier updates |
| Domain Classifier Updates per Round | 100 | Maximum classifier updates per round |
| ***Active Learning Loop*** | | |
| RCT Budget ($B$) | $\{10\text{k}, 50\text{k}, 100\text{k}, 300\text{k}, 500\text{k}\}$ | Total randomized samples allowed |
| Maximum Rounds | 5 | Maximum active-learning iterations |
| Training Epochs per Round | 1 | Training epochs after each acquisition |

**Active Sampling and Training Protocol.** During the iterative learning phase, the model selects a subset of informative samples from $\mathcal{D}_{\text{pool}}$ based on the acquisition function. We denote this actively queried subset as $\mathcal{D}_{\text{rct}}$, representing the Randomized Controlled Trial data where unbiased treatment effects are observed. Consequently, the final model is trained on the union of the biased historical data and the unbiased queried data:

$$\mathcal{D}_{\text{train}} = \mathcal{D}_{\text{obs}} \cup \mathcal{D}_{\text{rct}}. \tag{76}$$

This hybrid construction allows the model to leverage the scale of $\mathcal{D}_{\text{obs}}$ while correcting for bias using the high-quality signals from $\mathcal{D}_{\text{rct}}$.

**Computational Infrastructure.** All experiments are conducted on NVIDIA GPUs with mixed-precision support. We deploy computationally intensive active sampling strategies on high-performance units (L20 and RTX A6000), while assigning random sampling baselines to P40s.

## C.5. Evaluation Metrics

### C.5.1. NORMALIZED AREA UNDER THE UPLIFT CURVE (AUUC)

To evaluate the performance of our uplift model, we assess the ranking quality across the **entire test set** $\mathcal{D}_{test}$ of size $N$. Let $T_i \in \{0, 1\}$ represent the treatment assignment and $Y_i^{obs}$ denote the observed outcome for individual $i$.

The evaluation process begins by ranking the test instances based on their predicted uplift scores, $\hat{\tau}(x)$, from highest to lowest. Let the sequence of indices $i_1, i_2, \ldots, i_N$ correspond to this sorted order, satisfying $\hat{\tau}(x_{i_1}) \geq \hat{\tau}(x_{i_2}) \geq \cdots \geq \hat{\tau}(x_{i_N})$.

The **Uplift Curve** value at rank $k$ ($1 \leq k \leq N$), denoted as $f(k)$, quantifies the cumulative incremental gain obtained by targeting the top $k$ individuals. It is defined as (Gutierrez & Gérardy, 2017):

$$f(k) = \left( \frac{Y_k^T}{N_k^T} - \frac{Y_k^C}{N_k^C} \right) (N_k^T + N_k^C), \tag{77}$$

where $Y_k^T = \sum_{j=1}^k Y_{i_j}^{obs} T_{i_j}$ and $N_k^T = \sum_{j=1}^k T_{i_j}$ are the cumulative outcome sum and the count of treated individuals among the top $k$ samples, respectively (with $Y_k^C$ and $N_k^C$ defined analogously for the control group using $1 - T_{i_j}$).

To facilitate model comparison, we calculate the **Normalized AUUC**. This metric approximates the integral of the normalized uplift curve (over the population percentile from 0 to 1) by averaging the scaled gains across all ranks:

$$\text{AUUC} \approx \int_0^1 \frac{f(x)}{|f(N)|} \, dx \approx \frac{1}{N} \sum_{k=1}^N \frac{f(k)}{|f(N)|}. \tag{78}$$

Here, $f(N)$ corresponds to the global lift observed on the entire test set. A higher normalized AUUC indicates that the model effectively prioritizes individuals with the highest treatment effects.

## C.6. Experimental Results and Analysis

In this appendix, we detail the experimental protocols for the three settings introduced in the main text. Furthermore, we report supplementary findings regarding data scalability and conduct a comprehensive ablation analysis.

### C.6.1. EXP 1: PERFORMANCE UNDER EXTREME SELECTION BIAS

In this scenario, we focus on evaluating the model's ability to rectify learned policies under severe selection bias using a limited budget of active feedback. We initialize all base models using the **Biased Observational Set** ($\mathcal{D}_{\text{obs}}^{\text{bias}}$).

To simulate the active learning process, we iteratively query labels from the **RCT Candidate Pool** ($\mathcal{D}_{\text{pool}}$). The acquisition budget $B$ is varied across the range $\{10k, 50k, 100k, 300k, 500k\}$ to analyze the performance trajectory as the ratio of experimental data increases. **Upon completion of retraining on the augmented dataset in each round**, the updated model is evaluated on the out-of-sample **RCT Test Set** ($\mathcal{D}_{\text{rct}}^{\text{test}}$). We report the performance of various base learners across these budget levels. The comparative results are visualized in Figure 2 and detailed numerical metrics are provided in Table 3.

### C.6.2. EXP 2: ROBUSTNESS TO OBSERVATIONAL DATA SCALE

To validate the criticality of active sampling within our real-world production environment, we conduct a sensitivity analysis using the **Full-Scale Set** ($\mathcal{D}_{\text{obs}}^{\text{full}}$). We simulate varying data availability scenarios by down-sampling the historical business logs at different ratios $\rho$ and mixing them with the actively acquired samples:

$$\mathcal{D}_{\text{train}} = \text{Sample}(\mathcal{D}_{\text{obs}}^{\text{full}}, \rho) \cup \mathcal{D}_{\text{rct}} \tag{79}$$

Table 4 reports the AUUC of DRCFR under different observational-data mixing ratios and RCT budgets. Active Sampling gives the clearest gains when observational data are limited. In the $0.2\mathcal{D}_{\text{obs}}^{\text{full}} + \mathcal{D}_{\text{rct}}$ setting, it outperforms Random Sampling

*Table 3.* **AUUC of all backbones on the real-world dataset under active and random sampling.** Results are reported as mean ± standard deviation. Bold values indicate that *Active Learning* outperforms *Random Sampling* under the same model and budget.

| Model | Sampling | 1W | 5W | 10W | 30W | 50W |
|---|---|---|---|---|---|---|
| DRCFR | Active | **0.6501 ± 0.0052** | **0.6534 ± 0.0028** | **0.6568 ± 0.0029** | **0.6590 ± 0.0023** | **0.6618 ± 0.0016** |
| DRCFR | Random | 0.6430 ± 0.0018 | 0.6423 ± 0.0040 | 0.6457 ± 0.0029 | 0.6466 ± 0.0075 | 0.6604 ± 0.0044 |
| DESCN | Active | **0.6573 ± 0.0068** | **0.6598 ± 0.0079** | **0.6616 ± 0.0092** | **0.6650 ± 0.0077** | **0.6600 ± 0.0076** |
| DESCN | Random | 0.6563 ± 0.0068 | 0.6544 ± 0.0073 | 0.6522 ± 0.0011 | 0.6506 ± 0.0008 | 0.6584 ± 0.0055 |
| DragonNet | Active | **0.6632 ± 0.0064** | **0.6679 ± 0.0020** | 0.6689 ± 0.0026 | **0.6691 ± 0.0026** | **0.6696 ± 0.0043** |
| DragonNet | Random | 0.6614 ± 0.0071 | 0.6652 ± 0.0009 | 0.6725 ± 0.0011 | 0.6654 ± 0.0100 | 0.6689 ± 0.0013 |

at every budget. The improvement is especially clear from 50k to 500k, with the largest gap at 100k, where AUUC increases from 0.6215 to 0.6428.

When more observational data are available, the gap between the two sampling strategies becomes smaller. In the 0.3 and 0.5 settings, Active Sampling improves AUUC only at some budgets, and several differences are small. With $0.8\mathcal{D}_{\text{obs}}^{\text{full}}$, Active Sampling still gives higher mean AUUC in most cases, but the gains are modest. In the Group Sampling setting, where the observational data are biased, Active Sampling shows a clear improvement at 100k. These results suggest that active acquisition is most useful when the observational data have poor coverage or strong selection bias, since it can direct RCT samples toward regions that are not well represented by the historical data.

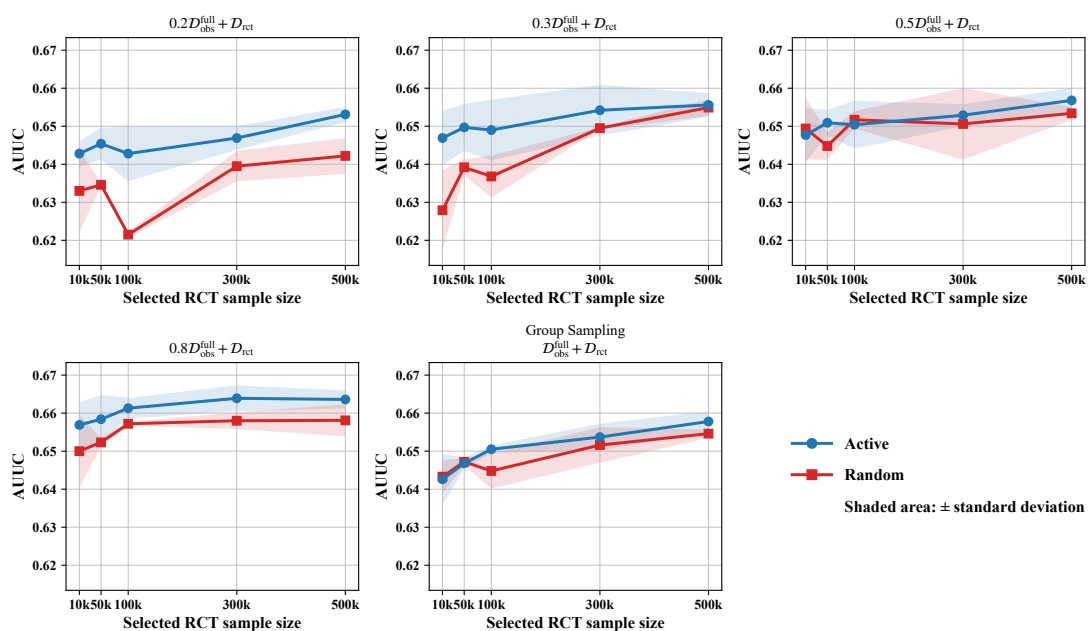

*Figure 3.* AUUC comparison between Active Sampling and Random Sampling under varying OBS data mixing ratios.

### C.7. Exp 3: Ablation Study on Acquisition Components

We study the contribution of each term in the acquisition score $S(u)$ (Eq. 14) through an ablation experiment on DRCFR. The comparison includes single-component scores, pairwise combinations, and the full score. Table 5 shows that no single component is uniformly best across all RCT budgets. Sampling by uncertainty alone is weak in the low-budget setting, which suggests that epistemic uncertainty by itself can place too much weight on noisy or poorly represented samples.

Adding distributional and bias-related information gives more stable performance. The overlap term is competitive at medium budgets, indicating that samples from weak-overlap regions are useful for reducing bias from historical assignments. Pairwise scores further improve the results, with $v_u + o_u$ performing best at 10k and 50k.

The full score $v_u + d_u + o_u$ is strongest, or close to strongest, at larger budgets, and gives the best AUUC at 100k and 500k.

*Table 4.* AUUC results of the DRCFR model across varying observational (OBS) data mixing ratios, reported as mean±standard deviation. Bold values indicate that Active is significantly higher than Random in the corresponding Active/Random pair under a one-sided paired $t$-test at the 5% significance level.

| Data Setting | Sampling | 10k | 50k | 100k | 300k | 500k |
|---|---|---|---|---|---|---|
| $0.2\mathcal{D}_{\text{obs}}^{\text{full}} + \mathcal{D}_{\text{rct}}$ | Active | 0.6428±0.0033 | **0.6454±0.0042** | **0.6428±0.0072** | **0.6469±0.0031** | **0.6531±0.0020** |
| | Random | 0.6330±0.0108 | 0.6346±0.0004 | 0.6215±0.0008 | 0.6395±0.0040 | 0.6422±0.0047 |
| $0.3\mathcal{D}_{\text{obs}}^{\text{full}} + \mathcal{D}_{\text{rct}}$ | Active | **0.6469±0.0072** | **0.6497±0.0062** | **0.6490±0.0080** | 0.6542±0.0067 | 0.6556±0.0031 |
| | Random | 0.6279±0.0105 | 0.6392±0.0021 | 0.6368±0.0055 | 0.6495±0.0003 | 0.6549±0.0022 |
| $0.5\mathcal{D}_{\text{obs}}^{\text{full}} + \mathcal{D}_{\text{rct}}$ | Active | 0.6477±0.0071 | **0.6509±0.0034** | 0.6504±0.0063 | 0.6529±0.0029 | **0.6568±0.0033** |
| | Random | 0.6494±0.0082 | 0.6448±0.0036 | 0.6517±0.0023 | 0.6506±0.0094 | 0.6534±0.0017 |
| $0.8\mathcal{D}_{\text{obs}}^{\text{full}} + \mathcal{D}_{\text{rct}}$ | Active | **0.6569±0.0059** | 0.6584±0.0064 | **0.6613±0.0027** | **0.6639±0.0034** | **0.6636±0.0024** |
| | Random | 0.6500±0.0096 | 0.6523±0.0011 | 0.6572±0.0005 | 0.6580±0.0022 | 0.6581±0.0042 |
| Group Sampling $\mathcal{D}_{\text{obs}}^{\text{full}} + \mathcal{D}_{\text{rct}}$ | Active | 0.6426±0.0067 | 0.6468±0.0015 | **0.6505±0.0010** | 0.6537±0.0035 | 0.6578±0.0027 |
| | Random | 0.6433±0.0044 | 0.6472±0.0010 | 0.6448±0.0046 | 0.6516±0.0047 | 0.6546±0.0011 |

These results suggest that uncertainty, domain discrepancy, and overlap deficit capture different aspects of sample value. Combining them gives a sampling rule that balances informativeness, coverage of the target distribution, and counterfactual evidence.

*Table 5.* **Ablation study of DRCFR score components on the real-world dataset.** Entries report mean ± standard deviation AUUC at each RCT sample size. Bold values indicate the best-performing strategy at the corresponding budget.

| Acquisition Strategy | RCT Sample Size | | | | |
|---|---|---|---|---|---|
| | 10k | 50k | 100k | 300k | 500k |
| **Baseline (Random)** | 0.6430 ± 0.0018 | 0.6423 ± 0.0040 | 0.6457 ± 0.0029 | 0.6466 ± 0.0075 | 0.6604 ± 0.0044 |
| *Single-Component* | | | | | |
| Uncertainty ($v_u$) | 0.6377 ± 0.0062 | 0.6421 ± 0.0075 | 0.6490 ± 0.0046 | 0.6561 ± 0.0047 | 0.6586 ± 0.0035 |
| Discrepancy ($d_u$) | 0.6328 ± 0.0112 | 0.6460 ± 0.0051 | 0.6498 ± 0.0049 | **0.6596 ± 0.0041** | 0.6578 ± 0.0043 |
| Overlap Deficit ($o_u$) | 0.6382 ± 0.0025 | 0.6485 ± 0.0026 | 0.6551 ± 0.0017 | 0.6565 ± 0.0038 | 0.6588 ± 0.0035 |
| *Pairwise Combinations* | | | | | |
| $v_u + d_u$ | 0.6468 ± 0.0053 | 0.6524 ± 0.0040 | 0.6533 ± 0.0037 | 0.6578 ± 0.0041 | 0.6615 ± 0.0012 |
| $v_u + o_u$ | **0.6538 ± 0.0024** | **0.6569 ± 0.0019** | 0.6561 ± 0.0006 | 0.6593 ± 0.0016 | 0.6582 ± 0.0043 |
| $d_u + o_u$ | 0.6456 ± 0.0044 | 0.6538 ± 0.0019 | 0.6549 ± 0.0022 | 0.6582 ± 0.0037 | 0.6601 ± 0.0039 |
| *Full Strategy* | | | | | |
| Full Score ($v_u + d_u + o_u$) | 0.6501 ± 0.0052 | 0.6534 ± 0.0028 | **0.6568 ± 0.0029** | 0.6590 ± 0.0023 | **0.6618 ± 0.0016** |

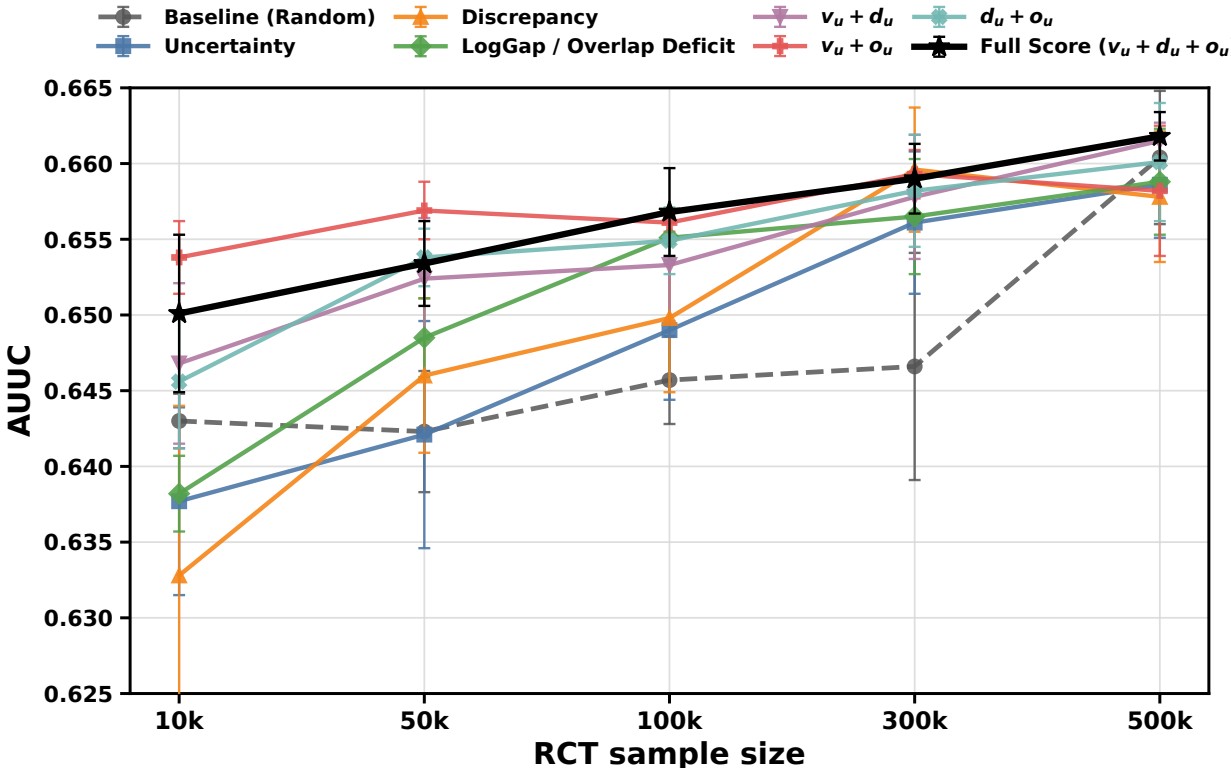

*Figure 4.* DRCFR ablation study on the real-world dataset: AUUC comparison of each score component variant against Random Sampling across different RCT sample sizes. Shaded areas denote $\pm$ standard deviation.

### C.8. Exp 4: Sensitivity Analysis

To examine the robustness of our method to hyperparameter choices, we conduct a sensitivity analysis on DRCFR under a fixed RCT budget of 100k. We separately vary the acquisition-score weights $(\alpha, \beta, \gamma)$ and the sample reweighting parameters $(w_{\text{gold}}, w_{\text{silver}})$, while keeping the remaining parameters fixed at their default values. The evaluated parameter ranges and default settings are summarized in Table 6, and the corresponding AUUC results are reported in Table 7.

*Table 6.* **Hyperparameter settings for the DRCFR sensitivity analysis.** Bold values denote the default configuration used in the main experiments.

| Module | Parameter | Values Tested | Role |
|---|---|---|---|
| Acquisition Score | $\alpha$ | $\{0.0, 0.25, \mathbf{0.5}, 0.75, 1.0\}$ | Uncertainty weight. |
| | $\beta$ | $\{0.0, \mathbf{1.0}, 1.5, 2.0\}$ | Domain discrepancy weight. |
| | $\gamma$ | $\{0.0, 0.35, \mathbf{0.7}, 1.0, 1.4\}$ | Overlap-deficit weight. |
| Sample Reweighting | $w_{\text{gold}}$ | $\{0.5, 0.75, \mathbf{1.0}, 1.5, 2.0\}$ | Weight assigned to high-gap counterfactual samples. |
| | $w_{\text{silver}}$ | $\{0.0, 0.1, \mathbf{0.2}, 0.5, 1.0\}$ | Weight assigned to lower-gap complementary samples. |
| | $w_{\text{obs}}$ | $\{\mathbf{0.3}\}$ | Observational sample weight. |

The results show that the proposed method remains stable across a range of parameter settings.

*(i) For the acquisition score*, moderate values of $\alpha$, $\beta$, and $\gamma$ tend to give competitive performance. This suggests that uncertainty, domain discrepancy, and overlap-related signals each help select useful samples. Reducing any one of these terms too much can hurt performance, which indicates that the three components capture complementary information.

*(ii) For sample reweighting*, the method is also not very sensitive to the choices of $w_{\text{gold}}$ and $w_{\text{silver}}$. A small but nonzero weight for silver samples gives stable results, while a larger $w_{\text{gold}}$ can improve AUUC in some settings. The default

configuration used in the main experiments falls in a stable and competitive region, which supports the proposed acquisition and reweighting design.

*Table 7.* **Sensitivity analysis of DRCFR on the real-world dataset at a fixed RCT budget of 100k.** The left table studies the weight parameters $(\alpha, \beta, \gamma)$ in the score function. The right table studies the weights $(w_{\text{gold}}, w_{\text{silver}})$ used in Counterfactual Alignment and Sample Reweighting. The default active parameter setting is $(\alpha, \beta, \gamma) = (0.5, 1.0, 0.7)$, and the default reweighting setting is $(w_{\text{gold}}, w_{\text{silver}}) = (1.0, 0.2)$ with $w_{\text{obs}} = 0.3$, following the same configuration as in the main paper. Results are reported as mean $\pm$ standard deviation.

| Parameter | Value | AUUC |
|---|---|---|
| Baseline (Random) | — | $0.6457 \pm 0.0029$ |
| $\alpha$ | 0.0 | $0.6523 \pm 0.0060$ |
| $\alpha$ | 0.25 | $0.6533 \pm 0.0074$ |
| $\alpha$ | 0.75 | $0.6544 \pm 0.0059$ |
| $\alpha$ | 1.0 | $0.6538 \pm 0.0038$ |
| $\beta$ | 0.0 | $0.6519 \pm 0.0058$ |
| $\beta$ | 1.5 | $0.6535 \pm 0.0042$ |
| $\beta$ | 2.0 | $0.6515 \pm 0.0067$ |
| $\gamma$ | 0.0 | $0.6524 \pm 0.0023$ |
| $\gamma$ | 0.35 | $0.6552 \pm 0.0031$ |
| $\gamma$ | 1.0 | $0.6522 \pm 0.0041$ |
| $\gamma$ | 1.4 | $0.6536 \pm 0.0048$ |

| Parameter | Value | AUUC |
|---|---|---|
| Baseline (Random) | — | $0.6457 \pm 0.0029$ |
| $w_{\text{silver}}$ | 0.0 | $0.6541 \pm 0.0031$ |
| $w_{\text{silver}}$ | 0.1 | $0.6530 \pm 0.0078$ |
| $w_{\text{silver}}$ | 0.5 | $0.6534 \pm 0.0050$ |
| $w_{\text{silver}}$ | 1.0 | $0.6519 \pm 0.0053$ |
| $w_{\text{gold}}$ | 0.5 | $0.6511 \pm 0.0055$ |
| $w_{\text{gold}}$ | 0.75 | $0.6517 \pm 0.0049$ |
| $w_{\text{gold}}$ | 1.5 | $0.6535 \pm 0.0043$ |
| $w_{\text{gold}}$ | 2.0 | $0.6547 \pm 0.0043$ |

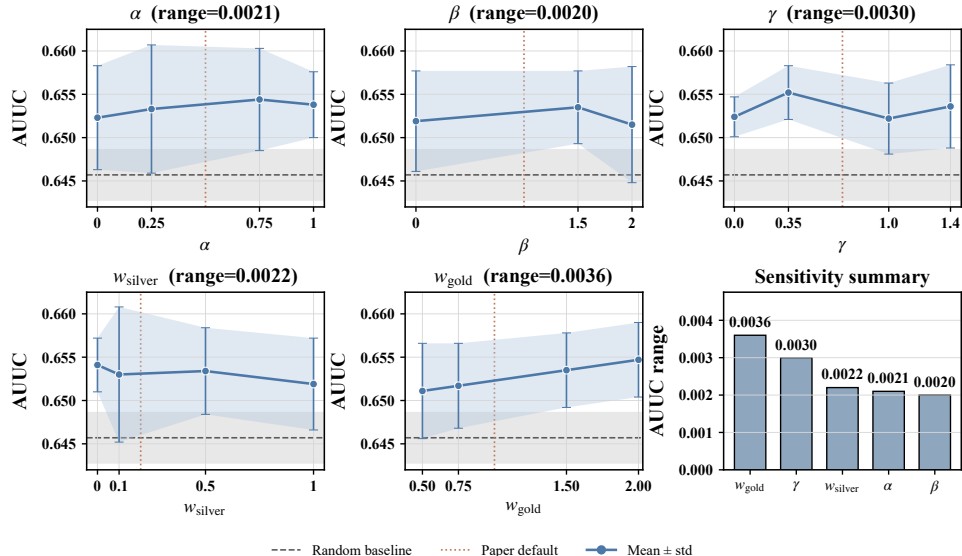

*Figure 5.* **Sensitivity analysis of DRCFR on the real-world dataset at a fixed RCT budget of 100k.** The five line plots show the AUUC under different settings of $\alpha$, $\beta$, $\gamma$, $w_{\text{silver}}$, and $w_{\text{gold}}$, respectively, with error bars and shaded bands indicating mean $\pm$ standard deviation. The horizontal dashed line denotes the Random Sampling baseline.

# D. Synthetic Dataset Experiments

We include the synthetic benchmark used to validate the theory under controlled ground-truth treatment effects. All results in this appendix are reported over 30 independent repeats. All tables report mean $\pm$ standard deviation of $\sqrt{\text{PEHE}}$, and lower values indicate better individual treatment effect estimation. Figure bands are generated from mean $\pm$ standard error.

## D.1. Experimental Protocol and Reproducibility

**Data-generating process.** For each feature dimension $d \in \{20, 50, 100, 150\}$, we generate an observational set, an RCT candidate pool, and a held-out test set. Observational covariates are sampled as $X_{\text{obs}} \sim \mathcal{N}(0, \Sigma_{0.3})$, while candidate-pool and test covariates are sampled as $X_{\text{pool}}, X_{\text{test}} \sim \mathcal{N}(\mu_{\text{shift}}, \Sigma_{0.5})$, where $\Sigma_{\rho,ij} = \rho^{|i-j|}$. The shift vector is nonzero only in the first ten coordinates, with values $(1.2, 1.0, 0.8, 0.6, 0, -0.3, 0.5, -0.2, 0.4, -0.1)$.

Given randomly drawn linear coefficients $\theta_0, \theta_\tau, \theta_e \sim \mathcal{N}(0, I_d)$, the conditional means are

$$\mu_0(x) = 0.5 + 0.15 \frac{x^\top \theta_0}{\max_{x' \in \mathcal{X}_{\text{obs}} \cup \mathcal{X}_{\text{pool}}} |x'^\top \theta_0|}, \quad \tau(x) = 0.15 \frac{x^\top \theta_\tau}{\max_{x' \in \mathcal{X}_{\text{obs}} \cup \mathcal{X}_{\text{pool}}} |x'^\top \theta_\tau|}, \quad \mu_1(x) = \mu_0(x) + \tau(x). \quad (80)$$

The scaling reference intentionally excludes the test covariates, so the held-out test set remains unused during data-generation calibration. Potential outcomes are then sampled as $Y(0) \sim \text{Bernoulli}(\mu_0(X))$ and $Y(1) \sim \text{Bernoulli}(\mu_1(X))$. Observational treatment follows $e(X) = \text{clip}(\sigma(8X^\top \theta_e / \sqrt{d}), 0.01, 0.99)$, while queried RCT labels use randomized assignment with probability $0.5$.

*Table 8.* Synthetic benchmark data and evaluation settings.

| Item | Value | Description |
|---|---|---|
| Feature dimensions | $\{20, 50, 100, 150\}$ | Low- to high-dimensional synthetic covariates |
| Observational samples | 1,200,000 | Biased historical data used for propensity and warm-start estimation |
| RCT candidate pool | 500,000 | Unlabeled pool from which active or random sampling queries labels |
| Held-out test set | 20,000 | Test covariates are generated but not used for DGP scaling |
| Acquisition budgets | $\{10\text{k}, 50\text{k}, 100\text{k}, 200\text{k}, 300\text{k}, 400\text{k}, 500\text{k}\}$ | Total queried RCT labels |
| Batch size | 10,000 | Number of newly queried units per active-learning round |
| Repeats and seeds | 30 repeats; base seed $42 + 1000r$ | Active and random samplers use offsets of $+1$ and $+2$, respectively |
| Evaluation metric | $\sqrt{\text{PEHE}}$ | Computed against the known ground-truth CATE $\tau(x)$ on the held-out test set |

*Table 9.* Hyperparameter and computational settings for the synthetic experiments.

| Category | Setting | Value |
|---|---|---|
| Estimator | Ridge regularization $\lambda$ | 1.0 |
| Estimator | Pseudo-outcome | $\widetilde{Y} = TY/p - (1-T)Y/(1-p)$ |
| Propensity / domain models | Classifier | Logistic regression with a maximum of 500 iterations |
| Propensity clipping | $(p_{\min}, p_{\max})$ | $(0.01, 0.99)$ |
| RCT assignment | $p_{\text{rct}}$ | 0.5 |
| Acquisition score | Default $(\alpha, \beta, \gamma)$ | $(0.3, 1.0, 0.7)$ |
| Sensitivity grid | Values tested | $\{0, 0.25, 0.5, 0.75, 1.0\}$ |
| Feature preprocessing | Standardization | Fit on $\mathcal{D}_{\text{obs}} \cup \mathcal{D}_{\text{pool}}$ |
| CPU | Processor | Dual Intel Xeon Gold 6530 CPUs (64 physical cores in total) |
| Memory | RAM | 503 GiB system memory |
| Compute backend | Execution device | CPU |
| Software | Python / key packages | Python 3.10.20; NumPy 2.2.6; scikit-learn 1.7.2; SciPy 1.15.3; Matplotlib 3.10.9 |
| Parallelism | Worker configuration | Up to 64 concurrent workers, each using single-threaded BLAS operations |
| Reproducibility | Environment controls | Python hash seed fixed at 0; BLAS backends restricted to one thread per worker |

## D.2. Fixed-Weight Setting

Figures 6 and 7 compare active sampling and random sampling with fixed acquisition weights $(\alpha, \beta, \gamma) = (0.3, 1.0, 0.7)$ in the 20D, 50D, 100D, and 150D synthetic settings. Across all dimensions, $\sqrt{\text{PEHE}}$ decreases as the RCT budget grows, in line with the theoretical rate $\mathcal{O}(\sqrt{d/B})$. The errors are also larger in higher-dimensional settings, which reflects the increased difficulty of estimating CATE in a higher-dimensional representation space.

Active sampling gives lower $\sqrt{\text{PEHE}}$ than random sampling in most cases. The gains are more visible when the RCT budget is small and when the covariate dimension is high, suggesting that the acquisition rule selects more useful experimental units and improves the conditioning of the design matrix. As the budget increases, the gap between the two sampling strategies becomes smaller. This behavior is consistent with the theory: active sampling improves the finite-sample constants through a better design, but it does not change the underlying $\sqrt{d/B}$ convergence rate.

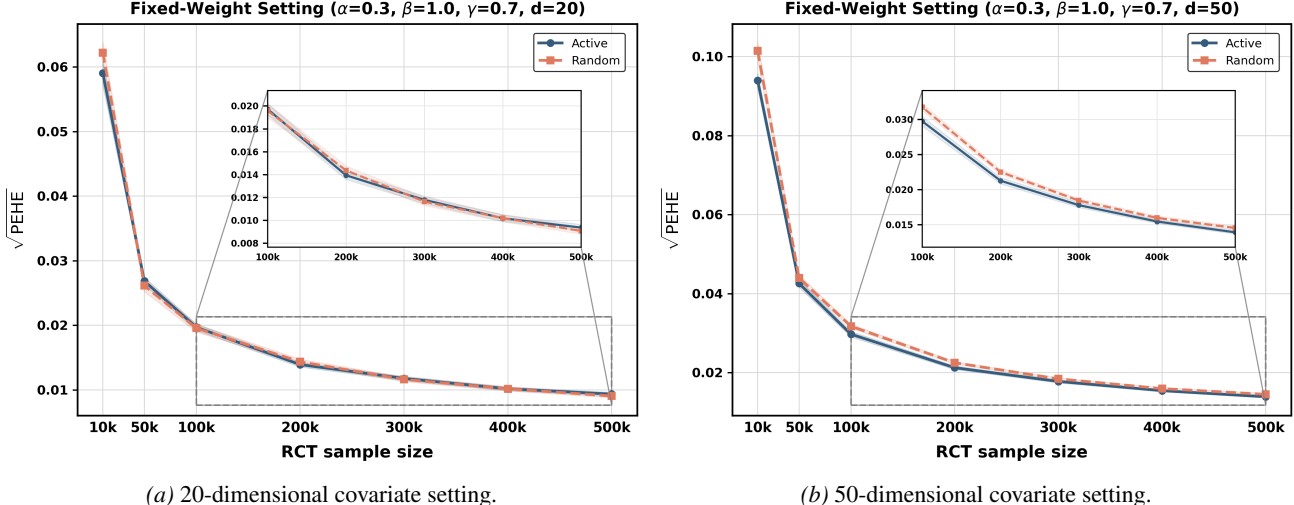

*(a)* 20-dimensional covariate setting.

*(b)* 50-dimensional covariate setting.

*Figure 6.* PEHE comparison between active and random sampling under the 20D and 50D synthetic covariate settings with fixed weights $(\alpha, \beta, \gamma) = (0.3, 1.0, 0.7)$.

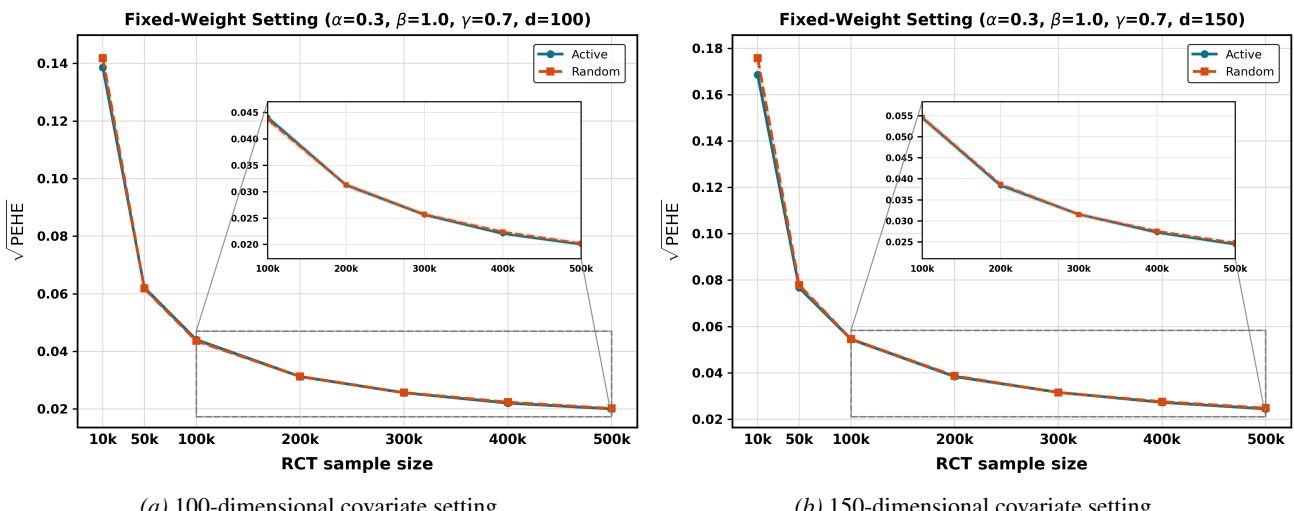

*(a)* 100-dimensional covariate setting.

*(b)* 150-dimensional covariate setting.

*Figure 7.* PEHE comparison between active and random sampling under the 100D and 150D synthetic covariate settings with fixed weights $(\alpha, \beta, \gamma) = (0.3, 1.0, 0.7)$.

*Table 10.* **Fixed-weight PEHE across synthetic dimensions with 30 repeats.** Entries report mean $\pm$ standard deviation of $\sqrt{\text{PEHE}}$ over 30 repeats; lower is better. Bold indicates that Active is lower than Random and significant at the 5% level using the one-sided summary-statistic test.

| Dimension | Strategy | 10k | 50k | 100k | 200k | 300k | 400k | 500k |
|---|---|---|---|---|---|---|---|---|
| 20D | Active | $0.0590 \pm 0.0103$ | $0.0269 \pm 0.0032$ | $0.0197 \pm 0.0027$ | $0.0139 \pm 0.0020$ | $0.0118 \pm 0.0018$ | $0.0102 \pm 0.0016$ | $0.0094 \pm 0.0015$ |
| | Random | $0.0622 \pm 0.0092$ | $0.0261 \pm 0.0050$ | $0.0196 \pm 0.0033$ | $0.0144 \pm 0.0021$ | $0.0116 \pm 0.0019$ | $0.0102 \pm 0.0018$ | $0.0091 \pm 0.0015$ |
| 50D | Active | $\mathbf{0.0939 \pm 0.0097}$ | $0.0426 \pm 0.0051$ | $\mathbf{0.0297 \pm 0.0038}$ | $\mathbf{0.0213 \pm 0.0025}$ | $0.0178 \pm 0.0019$ | $0.0154 \pm 0.0015$ | $\mathbf{0.0139 \pm 0.0014}$ |
| | Random | $0.1014 \pm 0.0127$ | $0.0439 \pm 0.0044$ | $0.0317 \pm 0.0028$ | $0.0225 \pm 0.0018$ | $0.0184 \pm 0.0015$ | $0.0160 \pm 0.0015$ | $0.0145 \pm 0.0015$ |
| 100D | Active | $0.1387 \pm 0.0122$ | $0.0621 \pm 0.0044$ | $0.0441 \pm 0.0025$ | $0.0312 \pm 0.0023$ | $0.0256 \pm 0.0015$ | $0.0220 \pm 0.0016$ | $0.0200 \pm 0.0013$ |
| | Random | $0.1418 \pm 0.0102$ | $0.0618 \pm 0.0039$ | $0.0436 \pm 0.0029$ | $0.0313 \pm 0.0017$ | $0.0257 \pm 0.0019$ | $0.0224 \pm 0.0018$ | $0.0202 \pm 0.0017$ |
| 150D | Active | $\mathbf{0.1686 \pm 0.0097}$ | $0.0767 \pm 0.0042$ | $0.0545 \pm 0.0035$ | $0.0384 \pm 0.0031$ | $0.0316 \pm 0.0024$ | $0.0272 \pm 0.0022$ | $0.0244 \pm 0.0016$ |
| | Random | $0.1758 \pm 0.0123$ | $0.0778 \pm 0.0052$ | $0.0545 \pm 0.0035$ | $0.0387 \pm 0.0019$ | $0.0315 \pm 0.0016$ | $0.0275 \pm 0.0016$ | $0.0247 \pm 0.0015$ |

## D.3. Sensitivity Analysis

We vary the acquisition weights $(\alpha, \beta, \gamma)$ to study the robustness of the proposed sampling rule. The results show that performance is relatively stable around the default setting, and that the three components play complementary roles:

**(i)** $\alpha$ encourages querying samples with high CATE uncertainty, but overly large values may overemphasize noisy regions;

**(ii)** $\beta$ improves coverage by selecting samples underrepresented in the current training distribution;

**(iii)** $\gamma$ directs RCT queries toward weak-overlap regions where observational data provide limited counterfactual support.

Using the three terms together gives the best overall performance, which is consistent with our theory that active sampling improves finite-sample efficiency by producing a better-conditioned experimental design.

### D.3.1. PARAMETER $\alpha$ SENSITIVITY ANALYSIS

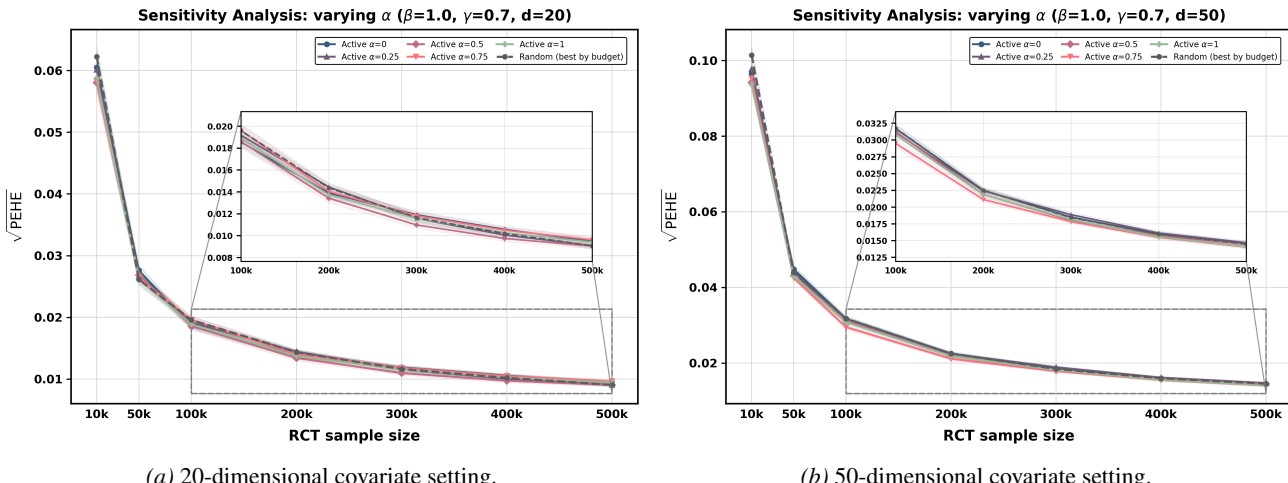

*(a)* 20-dimensional covariate setting.      *(b)* 50-dimensional covariate setting.

*Figure 8.* Sensitivity analysis with respect to $\alpha$ under the 20D and 50D synthetic covariate settings; the remaining acquisition weights are fixed to $\beta = 1.0, \gamma = 0.7$.

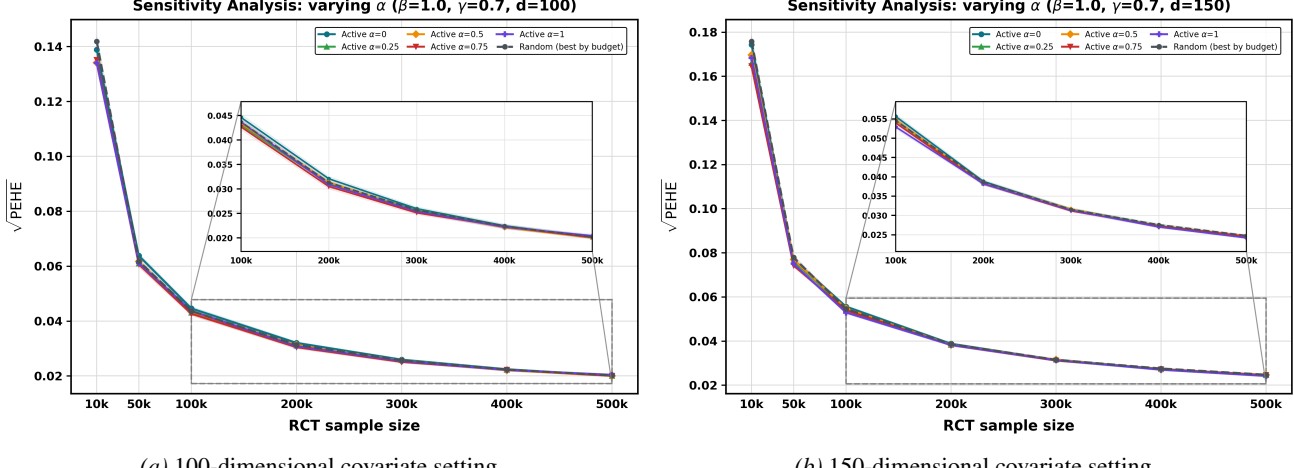

*(a)* 100-dimensional covariate setting.  *(b)* 150-dimensional covariate setting.

*Figure 9.* Sensitivity analysis with respect to $\alpha$ under the 100D and 150D synthetic covariate settings; the remaining acquisition weights are fixed to $\beta = 1.0, \gamma = 0.7$.

*Table 11.* $\alpha$ **sensitivity across synthetic dimensions with 30 repeats.** Entries report mean $\pm$ standard deviation of $\sqrt{\text{PEHE}}$ over 30 repeats; lower is better. Bold marks the best active setting at each budget when it is significantly lower than Random at the 5% level using the one-sided summary-statistic test.

| Dimension | $\alpha$ | 10k | 50k | 100k | 200k | 300k | 400k | 500k |
|---|---|---|---|---|---|---|---|---|
| 20D | 0.00 | $0.0605 \pm 0.0107$ | $0.0276 \pm 0.0047$ | $0.0186 \pm 0.0025$ | $0.0139 \pm 0.0023$ | $0.0119 \pm 0.0017$ | $0.0106 \pm 0.0015$ | $0.0095 \pm 0.0017$ |
| | 0.25 | $0.0601 \pm 0.0089$ | $0.0268 \pm 0.0034$ | $0.0192 \pm 0.0024$ | $0.0144 \pm 0.0019$ | $0.0116 \pm 0.0021$ | $0.0101 \pm 0.0015$ | $0.0091 \pm 0.0012$ |
| | 0.50 | $\mathbf{0.0580 \pm 0.0099}$ | $0.0268 \pm 0.0043$ | $0.0186 \pm 0.0030$ | $\mathbf{0.0134 \pm 0.0023}$ | $0.0110 \pm 0.0017$ | $0.0097 \pm 0.0016$ | $0.0090 \pm 0.0014$ |
| | 0.75 | $0.0582 \pm 0.0076$ | $0.0264 \pm 0.0038$ | $0.0196 \pm 0.0029$ | $0.0140 \pm 0.0028$ | $0.0118 \pm 0.0019$ | $0.0105 \pm 0.0015$ | $0.0096 \pm 0.0015$ |
| | 1.00 | $0.0587 \pm 0.0087$ | $0.0263 \pm 0.0050$ | $0.0189 \pm 0.0029$ | $0.0137 \pm 0.0025$ | $0.0116 \pm 0.0020$ | $0.0102 \pm 0.0013$ | $0.0094 \pm 0.0014$ |
| | Random | $0.0622 \pm 0.0092$ | $0.0261 \pm 0.0050$ | $0.0196 \pm 0.0033$ | $0.0144 \pm 0.0021$ | $0.0116 \pm 0.0019$ | $0.0102 \pm 0.0018$ | $0.0091 \pm 0.0015$ |
| 50D | 0.00 | $0.0965 \pm 0.0085$ | $0.0448 \pm 0.0051$ | $0.0317 \pm 0.0036$ | $0.0225 \pm 0.0025$ | $0.0185 \pm 0.0023$ | $0.0160 \pm 0.0019$ | $0.0145 \pm 0.0016$ |
| | 0.25 | $0.0977 \pm 0.0103$ | $0.0441 \pm 0.0042$ | $0.0311 \pm 0.0028$ | $0.0224 \pm 0.0020$ | $0.0189 \pm 0.0019$ | $0.0161 \pm 0.0016$ | $0.0147 \pm 0.0014$ |
| | 0.50 | $0.0942 \pm 0.0078$ | $0.0431 \pm 0.0041$ | $0.0313 \pm 0.0032$ | $0.0219 \pm 0.0021$ | $0.0180 \pm 0.0017$ | $0.0157 \pm 0.0015$ | $0.0144 \pm 0.0012$ |
| | 0.75 | $0.0953 \pm 0.0077$ | $0.0425 \pm 0.0036$ | $\mathbf{0.0295 \pm 0.0032}$ | $\mathbf{0.0211 \pm 0.0021}$ | $0.0178 \pm 0.0016$ | $0.0155 \pm 0.0017$ | $0.0140 \pm 0.0015$ |
| | 1.00 | $\mathbf{0.0937 \pm 0.0088}$ | $0.0430 \pm 0.0043$ | $0.0308 \pm 0.0026$ | $0.0218 \pm 0.0026$ | $0.0181 \pm 0.0020$ | $0.0155 \pm 0.0016$ | $0.0141 \pm 0.0015$ |
| | Random | $0.1014 \pm 0.0127$ | $0.0439 \pm 0.0044$ | $0.0317 \pm 0.0028$ | $0.0225 \pm 0.0018$ | $0.0184 \pm 0.0015$ | $0.0160 \pm 0.0015$ | $0.0145 \pm 0.0015$ |
| 100D | 0.00 | $0.1388 \pm 0.0094$ | $0.0638 \pm 0.0044$ | $0.0446 \pm 0.0036$ | $0.0320 \pm 0.0024$ | $0.0259 \pm 0.0020$ | $0.0224 \pm 0.0020$ | $0.0201 \pm 0.0016$ |
| | 0.25 | $0.1349 \pm 0.0101$ | $0.0610 \pm 0.0043$ | $0.0431 \pm 0.0026$ | $0.0310 \pm 0.0021$ | $0.0258 \pm 0.0020$ | $0.0222 \pm 0.0015$ | $0.0201 \pm 0.0014$ |
| | 0.50 | $0.1341 \pm 0.0101$ | $0.0617 \pm 0.0047$ | $0.0433 \pm 0.0033$ | $0.0313 \pm 0.0025$ | $0.0254 \pm 0.0021$ | $0.0221 \pm 0.0015$ | $0.0200 \pm 0.0014$ |
| | 0.75 | $0.1351 \pm 0.0111$ | $0.0607 \pm 0.0052$ | $0.0426 \pm 0.0033$ | $0.0305 \pm 0.0027$ | $0.0251 \pm 0.0020$ | $0.0223 \pm 0.0017$ | $0.0201 \pm 0.0016$ |
| | 1.00 | $\mathbf{0.1340 \pm 0.0085}$ | $0.0611 \pm 0.0041$ | $0.0436 \pm 0.0030$ | $0.0309 \pm 0.0025$ | $0.0254 \pm 0.0022$ | $0.0222 \pm 0.0019$ | $0.0204 \pm 0.0016$ |
| | Random | $0.1418 \pm 0.0102$ | $0.0618 \pm 0.0039$ | $0.0436 \pm 0.0029$ | $0.0313 \pm 0.0017$ | $0.0257 \pm 0.0019$ | $0.0224 \pm 0.0018$ | $0.0202 \pm 0.0017$ |
| 150D | 0.00 | $0.1743 \pm 0.0095$ | $0.0768 \pm 0.0045$ | $0.0556 \pm 0.0031$ | $0.0388 \pm 0.0023$ | $0.0314 \pm 0.0019$ | $0.0271 \pm 0.0015$ | $0.0244 \pm 0.0015$ |
| | 0.25 | $0.1660 \pm 0.0095$ | $0.0763 \pm 0.0046$ | $0.0546 \pm 0.0030$ | $0.0383 \pm 0.0023$ | $0.0316 \pm 0.0019$ | $0.0273 \pm 0.0015$ | $0.0246 \pm 0.0012$ |
| | 0.50 | $0.1697 \pm 0.0109$ | $0.0769 \pm 0.0043$ | $0.0547 \pm 0.0030$ | $0.0383 \pm 0.0019$ | $0.0316 \pm 0.0017$ | $0.0270 \pm 0.0014$ | $0.0247 \pm 0.0014$ |
| | 0.75 | $\mathbf{0.1645 \pm 0.0091}$ | $\mathbf{0.0743 \pm 0.0043}$ | $0.0539 \pm 0.0019$ | $0.0381 \pm 0.0016$ | $0.0312 \pm 0.0016$ | $0.0272 \pm 0.0017$ | $0.0245 \pm 0.0014$ |
| | 1.00 | $0.1683 \pm 0.0093$ | $0.0750 \pm 0.0037$ | $\mathbf{0.0529 \pm 0.0028}$ | $0.0382 \pm 0.0021$ | $0.0313 \pm 0.0021$ | $0.0270 \pm 0.0015$ | $0.0242 \pm 0.0015$ |
| | Random | $0.1758 \pm 0.0123$ | $0.0778 \pm 0.0052$ | $0.0545 \pm 0.0035$ | $0.0387 \pm 0.0019$ | $0.0315 \pm 0.0016$ | $0.0275 \pm 0.0016$ | $0.0247 \pm 0.0015$ |

### D.3.2. PARAMETER $\beta$ SENSITIVITY ANALYSIS

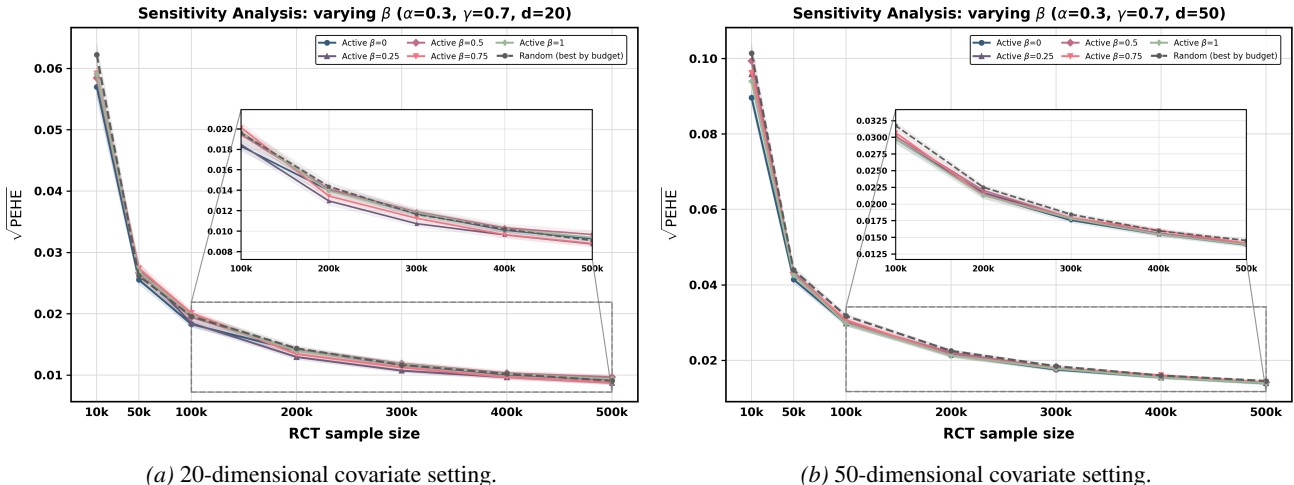

*(a)* 20-dimensional covariate setting.

*(b)* 50-dimensional covariate setting.

*Figure 10.* Sensitivity analysis with respect to $\beta$ under the 20D and 50D synthetic covariate settings; the remaining acquisition weights are fixed to $\alpha = 0.3, \gamma = 0.7$.

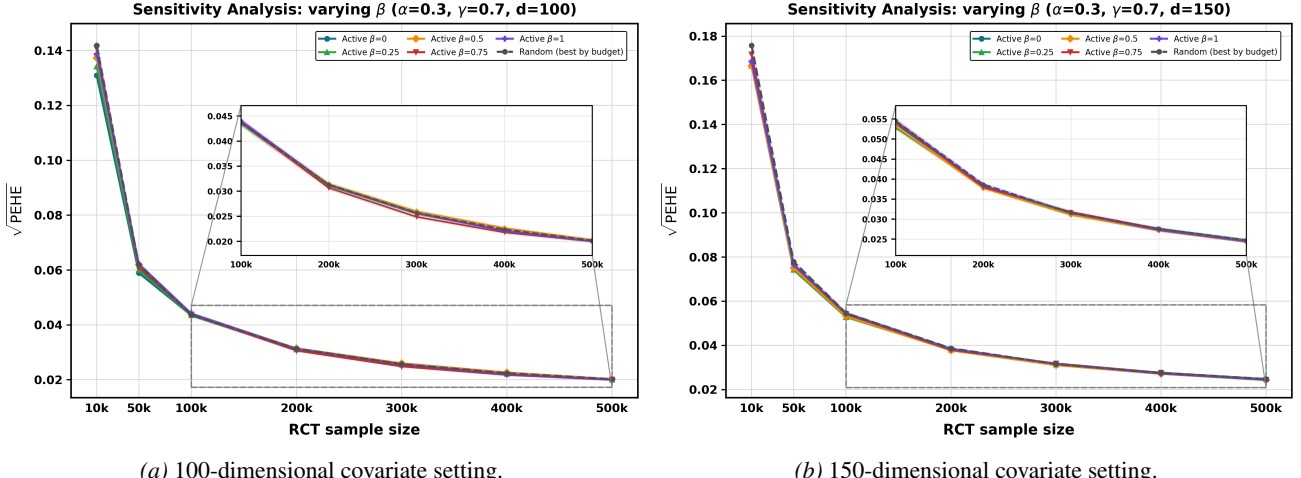

*(a)* 100-dimensional covariate setting.

*(b)* 150-dimensional covariate setting.

*Figure 11.* Sensitivity analysis with respect to $\beta$ under the 100D and 150D synthetic covariate settings; the remaining acquisition weights are fixed to $\alpha = 0.3, \gamma = 0.7$.

*Table 12.* $\beta$ **sensitivity across synthetic dimensions with 30 repeats.** Entries report mean $\pm$ standard deviation of $\sqrt{\text{PEHE}}$ over 30 repeats; lower is better. Bold marks the best active setting at each budget when it is significantly lower than Random at the 5% level using the one-sided summary-statistic test.

| Dimension | $\beta$ | 10k | 50k | 100k | 200k | 300k | 400k | 500k |
|---|---|---|---|---|---|---|---|---|
| 20D | 0.00 | **0.0570 ± 0.0097** | 0.0256 ± 0.0039 | **0.0183 ± 0.0026** | 0.0140 ± 0.0022 | 0.0117 ± 0.0018 | 0.0101 ± 0.0014 | 0.0092 ± 0.0012 |
| | 0.25 | 0.0587 ± 0.0080 | 0.0274 ± 0.0044 | 0.0185 ± 0.0023 | **0.0129 ± 0.0021** | **0.0107 ± 0.0015** | 0.0096 ± 0.0018 | 0.0087 ± 0.0016 |
| | 0.50 | 0.0584 ± 0.0090 | 0.0267 ± 0.0052 | 0.0194 ± 0.0031 | 0.0141 ± 0.0019 | 0.0119 ± 0.0021 | 0.0103 ± 0.0020 | 0.0097 ± 0.0019 |
| | 0.75 | 0.0592 ± 0.0106 | 0.0274 ± 0.0045 | 0.0201 ± 0.0029 | 0.0134 ± 0.0021 | 0.0112 ± 0.0019 | 0.0097 ± 0.0015 | 0.0088 ± 0.0013 |
| | 1.00 | 0.0590 ± 0.0103 | 0.0269 ± 0.0032 | 0.0197 ± 0.0027 | 0.0139 ± 0.0020 | 0.0118 ± 0.0018 | 0.0102 ± 0.0016 | 0.0094 ± 0.0015 |
| | Random | 0.0622 ± 0.0092 | 0.0261 ± 0.0050 | 0.0196 ± 0.0033 | 0.0144 ± 0.0021 | 0.0116 ± 0.0019 | 0.0102 ± 0.0018 | 0.0091 ± 0.0015 |
| 50D | 0.00 | **0.0897 ± 0.0091** | **0.0414 ± 0.0049** | 0.0298 ± 0.0030 | 0.0213 ± 0.0029 | **0.0175 ± 0.0020** | 0.0154 ± 0.0018 | **0.0138 ± 0.0016** |
| | 0.25 | 0.0958 ± 0.0078 | 0.0433 ± 0.0040 | 0.0297 ± 0.0032 | 0.0217 ± 0.0023 | 0.0179 ± 0.0020 | 0.0154 ± 0.0011 | 0.0140 ± 0.0010 |
| | 0.50 | 0.0993 ± 0.0086 | 0.0433 ± 0.0035 | 0.0301 ± 0.0032 | 0.0220 ± 0.0026 | 0.0179 ± 0.0019 | 0.0156 ± 0.0015 | 0.0141 ± 0.0016 |
| | 0.75 | 0.0962 ± 0.0094 | 0.0427 ± 0.0045 | 0.0307 ± 0.0030 | 0.0213 ± 0.0026 | 0.0179 ± 0.0018 | 0.0160 ± 0.0016 | 0.0141 ± 0.0014 |
| | 1.00 | 0.0939 ± 0.0097 | 0.0426 ± 0.0051 | **0.0297 ± 0.0038** | **0.0213 ± 0.0025** | 0.0178 ± 0.0019 | 0.0154 ± 0.0015 | 0.0139 ± 0.0014 |
| | Random | 0.1014 ± 0.0127 | 0.0439 ± 0.0044 | 0.0317 ± 0.0028 | 0.0225 ± 0.0018 | 0.0184 ± 0.0015 | 0.0160 ± 0.0015 | 0.0145 ± 0.0015 |
| 100D | 0.00 | **0.1309 ± 0.0080** | **0.0590 ± 0.0038** | 0.0435 ± 0.0030 | 0.0310 ± 0.0026 | 0.0256 ± 0.0021 | 0.0221 ± 0.0019 | 0.0200 ± 0.0015 |
| | 0.25 | 0.1342 ± 0.0092 | 0.0601 ± 0.0047 | 0.0435 ± 0.0027 | 0.0311 ± 0.0020 | 0.0255 ± 0.0012 | 0.0222 ± 0.0013 | 0.0200 ± 0.0015 |
| | 0.50 | 0.1373 ± 0.0091 | 0.0619 ± 0.0045 | 0.0436 ± 0.0029 | 0.0314 ± 0.0024 | 0.0260 ± 0.0018 | 0.0226 ± 0.0015 | 0.0203 ± 0.0013 |
| | 0.75 | 0.1408 ± 0.0096 | 0.0610 ± 0.0049 | 0.0437 ± 0.0035 | 0.0306 ± 0.0020 | **0.0249 ± 0.0017** | 0.0218 ± 0.0017 | 0.0200 ± 0.0017 |
| | 1.00 | 0.1387 ± 0.0122 | 0.0621 ± 0.0044 | 0.0441 ± 0.0025 | 0.0312 ± 0.0023 | 0.0256 ± 0.0015 | 0.0220 ± 0.0016 | 0.0200 ± 0.0013 |
| | Random | 0.1418 ± 0.0102 | 0.0618 ± 0.0039 | 0.0436 ± 0.0029 | 0.0313 ± 0.0017 | 0.0257 ± 0.0019 | 0.0224 ± 0.0018 | 0.0202 ± 0.0017 |
| 150D | 0.00 | 0.1668 ± 0.0105 | 0.0749 ± 0.0045 | 0.0538 ± 0.0028 | 0.0383 ± 0.0021 | 0.0312 ± 0.0017 | 0.0274 ± 0.0015 | 0.0246 ± 0.0017 |
| | 0.25 | 0.1667 ± 0.0104 | **0.0742 ± 0.0042** | **0.0528 ± 0.0032** | 0.0380 ± 0.0023 | 0.0313 ± 0.0020 | 0.0276 ± 0.0017 | 0.0247 ± 0.0014 |
| | 0.50 | **0.1665 ± 0.0098** | 0.0750 ± 0.0051 | 0.0532 ± 0.0035 | **0.0377 ± 0.0021** | 0.0310 ± 0.0017 | 0.0271 ± 0.0014 | 0.0242 ± 0.0012 |
| | 0.75 | 0.1717 ± 0.0110 | 0.0764 ± 0.0041 | 0.0542 ± 0.0033 | 0.0380 ± 0.0023 | 0.0318 ± 0.0021 | 0.0276 ± 0.0013 | 0.0244 ± 0.0011 |
| | 1.00 | 0.1686 ± 0.0097 | 0.0767 ± 0.0042 | 0.0545 ± 0.0035 | 0.0384 ± 0.0031 | 0.0316 ± 0.0024 | 0.0272 ± 0.0022 | 0.0244 ± 0.0016 |
| | Random | 0.1758 ± 0.0123 | 0.0778 ± 0.0052 | 0.0545 ± 0.0035 | 0.0387 ± 0.0019 | 0.0315 ± 0.0016 | 0.0275 ± 0.0016 | 0.0247 ± 0.0015 |

### D.3.3. PARAMETER $\gamma$ SENSITIVITY ANALYSIS

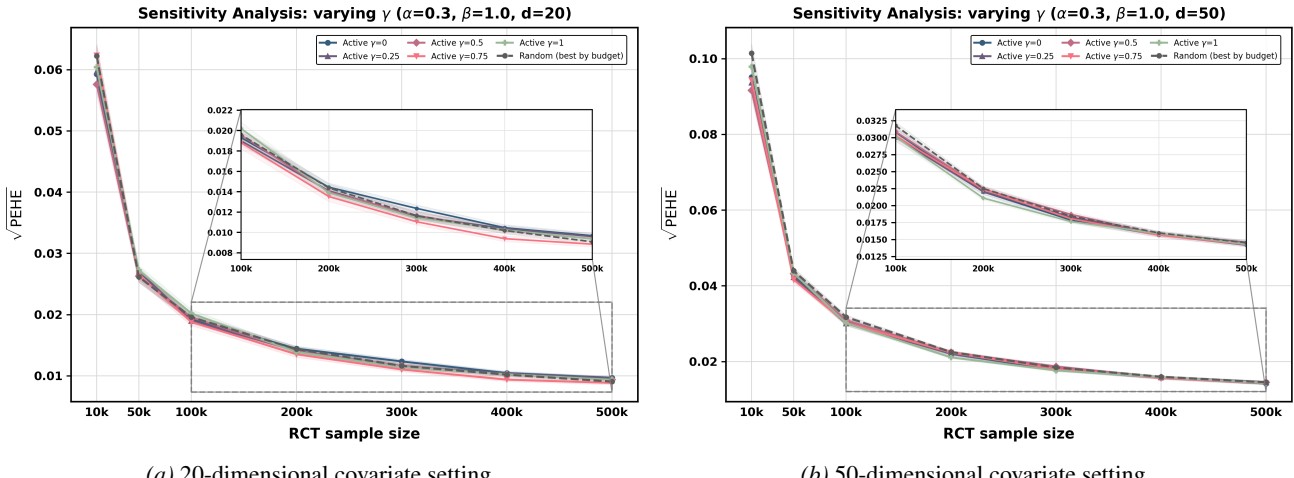

*(a)* 20-dimensional covariate setting.

*(b)* 50-dimensional covariate setting.

*Figure 12.* Sensitivity analysis with respect to $\gamma$ under the 20D and 50D synthetic covariate settings; the remaining acquisition weights are fixed to $\alpha = 0.3, \beta = 1.0$.

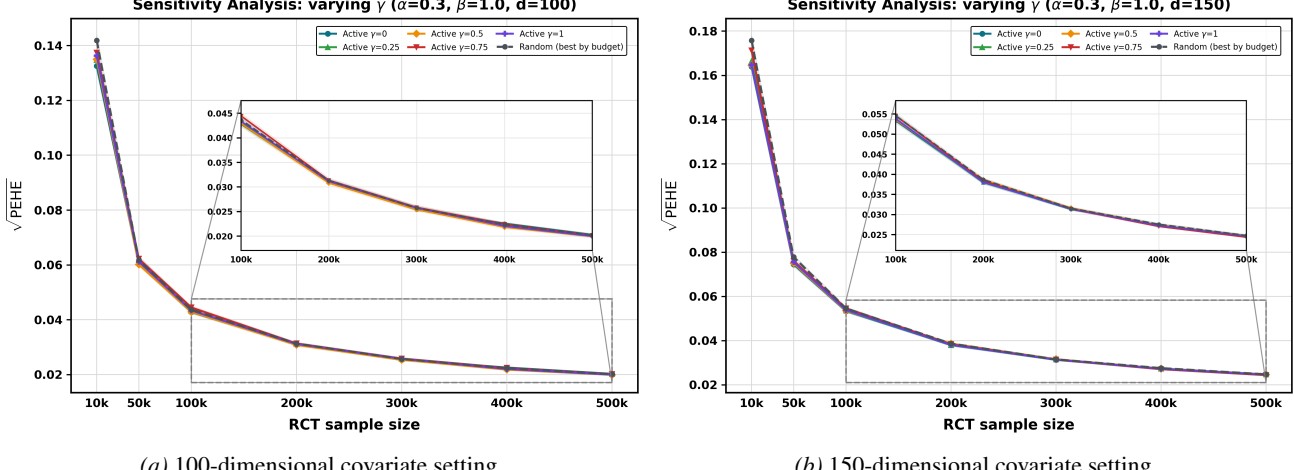

*(a)* 100-dimensional covariate setting.

*(b)* 150-dimensional covariate setting.

*Figure 13.* Sensitivity analysis with respect to $\gamma$ under the 100D and 150D synthetic covariate settings; the remaining acquisition weights are fixed to $\alpha = 0.3, \beta = 1.0$.

*Table 13.* $\gamma$ **sensitivity across synthetic dimensions with 30 repeats.** Entries report mean $\pm$ standard deviation of $\sqrt{\text{PEHE}}$ over 30 repeats; lower is better. Bold marks the best active setting at each budget when it is significantly lower than Random at the 5% level using the one-sided summary-statistic test.

| Dimension | $\gamma$ | 10k | 50k | 100k | 200k | 300k | 400k | 500k |
|---|---|---|---|---|---|---|---|---|
| 20D | 0.00 | $0.0592 \pm 0.0081$ | $0.0264 \pm 0.0035$ | $0.0193 \pm 0.0031$ | $0.0144 \pm 0.0022$ | $0.0123 \pm 0.0018$ | $0.0105 \pm 0.0016$ | $0.0097 \pm 0.0013$ |
| | 0.25 | $0.0597 \pm 0.0099$ | $0.0271 \pm 0.0035$ | $0.0190 \pm 0.0024$ | $0.0141 \pm 0.0023$ | $0.0116 \pm 0.0018$ | $0.0104 \pm 0.0015$ | $0.0096 \pm 0.0014$ |
| | 0.50 | $\mathbf{0.0576 \pm 0.0083}$ | $0.0263 \pm 0.0051$ | $0.0195 \pm 0.0036$ | $0.0140 \pm 0.0021$ | $0.0117 \pm 0.0024$ | $0.0102 \pm 0.0017$ | $0.0094 \pm 0.0015$ |
| | 0.75 | $0.0623 \pm 0.0099$ | $0.0265 \pm 0.0053$ | $0.0188 \pm 0.0033$ | $0.0135 \pm 0.0026$ | $0.0110 \pm 0.0018$ | $\mathbf{0.0094 \pm 0.0016}$ | $0.0088 \pm 0.0015$ |
| | 1.00 | $0.0604 \pm 0.0107$ | $0.0272 \pm 0.0043$ | $0.0202 \pm 0.0034$ | $0.0139 \pm 0.0025$ | $0.0114 \pm 0.0016$ | $0.0102 \pm 0.0013$ | $0.0093 \pm 0.0011$ |
| | Random | $0.0622 \pm 0.0092$ | $0.0261 \pm 0.0050$ | $0.0196 \pm 0.0033$ | $0.0144 \pm 0.0021$ | $0.0116 \pm 0.0019$ | $0.0102 \pm 0.0018$ | $0.0091 \pm 0.0015$ |
| 50D | 0.00 | $0.0951 \pm 0.0100$ | $0.0429 \pm 0.0045$ | $0.0308 \pm 0.0034$ | $0.0220 \pm 0.0022$ | $0.0179 \pm 0.0021$ | $0.0157 \pm 0.0018$ | $0.0142 \pm 0.0015$ |
| | 0.25 | $0.0937 \pm 0.0099$ | $0.0423 \pm 0.0041$ | $\mathbf{0.0300 \pm 0.0033}$ | $0.0220 \pm 0.0023$ | $0.0183 \pm 0.0025$ | $0.0158 \pm 0.0019$ | $0.0146 \pm 0.0016$ |
| | 0.50 | $\mathbf{0.0916 \pm 0.0109}$ | $0.0432 \pm 0.0055$ | $0.0309 \pm 0.0037$ | $0.0224 \pm 0.0022$ | $0.0187 \pm 0.0019$ | $0.0157 \pm 0.0017$ | $0.0145 \pm 0.0016$ |
| | 0.75 | $0.0943 \pm 0.0073$ | $\mathbf{0.0416 \pm 0.0037}$ | $0.0303 \pm 0.0027$ | $0.0224 \pm 0.0024$ | $0.0180 \pm 0.0023$ | $0.0156 \pm 0.0017$ | $0.0142 \pm 0.0018$ |
| | 1.00 | $0.0979 \pm 0.0110$ | $0.0434 \pm 0.0047$ | $0.0301 \pm 0.0032$ | $\mathbf{0.0211 \pm 0.0016}$ | $\mathbf{0.0176 \pm 0.0015}$ | $0.0159 \pm 0.0016$ | $0.0143 \pm 0.0012$ |
| | Random | $0.1014 \pm 0.0127$ | $0.0439 \pm 0.0044$ | $0.0317 \pm 0.0028$ | $0.0225 \pm 0.0018$ | $0.0184 \pm 0.0015$ | $0.0160 \pm 0.0015$ | $0.0145 \pm 0.0015$ |
| 100D | 0.00 | $\mathbf{0.1325 \pm 0.0093}$ | $0.0605 \pm 0.0044$ | $0.0428 \pm 0.0026$ | $0.0310 \pm 0.0013$ | $0.0254 \pm 0.0014$ | $0.0226 \pm 0.0011$ | $0.0203 \pm 0.0012$ |
| | 0.25 | $0.1347 \pm 0.0091$ | $0.0613 \pm 0.0044$ | $0.0432 \pm 0.0030$ | $0.0311 \pm 0.0022$ | $0.0258 \pm 0.0025$ | $0.0223 \pm 0.0019$ | $0.0201 \pm 0.0016$ |
| | 0.50 | $0.1349 \pm 0.0096$ | $\mathbf{0.0601 \pm 0.0032}$ | $0.0429 \pm 0.0033$ | $0.0308 \pm 0.0023$ | $0.0254 \pm 0.0017$ | $0.0218 \pm 0.0018$ | $0.0199 \pm 0.0016$ |
| | 0.75 | $0.1374 \pm 0.0118$ | $0.0622 \pm 0.0048$ | $0.0445 \pm 0.0030$ | $0.0313 \pm 0.0026$ | $0.0258 \pm 0.0019$ | $0.0224 \pm 0.0016$ | $0.0202 \pm 0.0012$ |
| | 1.00 | $0.1363 \pm 0.0079$ | $0.0612 \pm 0.0037$ | $0.0434 \pm 0.0031$ | $0.0311 \pm 0.0021$ | $0.0257 \pm 0.0015$ | $0.0220 \pm 0.0016$ | $0.0200 \pm 0.0015$ |
| | Random | $0.1418 \pm 0.0102$ | $0.0618 \pm 0.0039$ | $0.0436 \pm 0.0029$ | $0.0313 \pm 0.0017$ | $0.0257 \pm 0.0019$ | $0.0224 \pm 0.0018$ | $0.0202 \pm 0.0017$ |
| 150D | 0.00 | $\mathbf{0.1639 \pm 0.0099}$ | $\mathbf{0.0745 \pm 0.0051}$ | $0.0533 \pm 0.0032$ | $0.0381 \pm 0.0023$ | $0.0313 \pm 0.0016$ | $0.0274 \pm 0.0014$ | $0.0246 \pm 0.0013$ |
| | 0.25 | $0.1664 \pm 0.0106$ | $0.0754 \pm 0.0051$ | $0.0536 \pm 0.0032$ | $0.0379 \pm 0.0024$ | $0.0315 \pm 0.0019$ | $0.0274 \pm 0.0016$ | $0.0247 \pm 0.0016$ |
| | 0.50 | $0.1646 \pm 0.0104$ | $0.0750 \pm 0.0044$ | $0.0536 \pm 0.0030$ | $0.0386 \pm 0.0024$ | $0.0316 \pm 0.0016$ | $0.0274 \pm 0.0016$ | $0.0247 \pm 0.0013$ |
| | 0.75 | $0.1713 \pm 0.0116$ | $0.0760 \pm 0.0045$ | $0.0546 \pm 0.0024$ | $0.0384 \pm 0.0021$ | $0.0314 \pm 0.0017$ | $0.0270 \pm 0.0014$ | $0.0244 \pm 0.0011$ |
| | 1.00 | $0.1647 \pm 0.0088$ | $0.0755 \pm 0.0058$ | $0.0538 \pm 0.0034$ | $0.0380 \pm 0.0020$ | $0.0314 \pm 0.0015$ | $0.0273 \pm 0.0013$ | $0.0246 \pm 0.0013$ |
| | Random | $0.1758 \pm 0.0123$ | $0.0778 \pm 0.0052$ | $0.0545 \pm 0.0035$ | $0.0387 \pm 0.0019$ | $0.0315 \pm 0.0016$ | $0.0275 \pm 0.0016$ | $0.0247 \pm 0.0015$ |

