# OpenReview forum: "Budgeted Active Experimentation for Treatment Effect Estimation from Observational and Randomized Data"
_ICML.cc/2026/Conference — ICML 2026 regular_

### Official Review · Reviewer_xrZB · 2026-03-05

**Soundness:** 3
**Presentation:** 3
**Significance:** 2
**Originality:** 3
**Overall Recommendation:** 4
**Confidence:** 3

**Summary:**

The paper investigates adaptive experimentation in settings where RCTs are expensive and observational data is available but biased. Efficiently combining these sources is a growing area of interest in both industry and academia. The authors propose an active sampling strategy that targets the uncertainty in uplift estimation. Their theoretical results include a derivation of a deviation bound and a nearly matching minimax lower bound.

**Compliance With Llm Reviewing Policy:**

Affirmed.

**Key Questions For Authors:**

1. There is another line of work on adaptive experimentation where data is collected adaptively, leading to bias. Those methods, such as "Adaptive Experimentation When You Can't Experiment" (NeurIPS 2024), appear to use different debiasing techniques like Instrumental Variables, they also use similar coupon delivery examples. Could the authors comment on these differences and provide a comparison with their approach?

2. An interesting question is how the behavior of the algorithm changes as the balance between RCT data and observational data shifts. It would be very helpful to see an intuitive explanation of this dynamic, supported by theoretical insights.

**Limitations:**

Yes

**Strengths And Weaknesses:**

Strengths

1. The paper addresses an important problem that holds both theoretical interest and significant practical value for real-world experimentation.
2. The proposed method is evaluated on real-world datasets, which strengthens the validity of the empirical claims.

Weaknesses

1. While there is a thorough discussion of related work and a comprehensive comparison with baselines in the experiments, the theoretical results lack comparison. The authors provide a minimax lower bound, but it would be beneficial to benchmark the theoretical contributions against existing results in the field to better contextualize the improvement.
2. Minor writing issues: Several abbreviations, such as PEHE, are used without being formally defined. Although many of these are standard, it is best practice to provide the full name upon the first mention.
3. Reproducibility issue: Related to strength #2, there is no open-source version of the data provided. This limits the ability of the community to reproduce the results or build directly upon the findings.

---

> ### Author Rebuttal · Authors · 2026-03-30
>
> Thank you for your kind review. We address each point below.
> >**Q1：** Clarify the differences between our approach and “Adaptive Experimentation When You Can’t Experiment” (NeurIPS 2024).
>
> **A1**：Thank you for pointing out `Adaptive Experimentation When You Can't Experiment`. While the two papers are related at a high level, they study materially different settings.
>
> The ``main difference`` is the **intervention regime**. Zhao et al. consider an *encouragement-design* setting, where the platform cannot directly randomize the actual treatment and can only randomize an instrument; treatment uptake therefore remains endogenous, so IV/2SLS is the appropriate debiasing tool there.
>
> In contrast, our setting assumes that the platform can directly randomize treatment, but only for a small adaptively selected subset of units from a queryable target pool. In our framework, OBS data are used to decide *where to randomize*, while causal identification comes from the queried RCT samples themselves.
>
> The **objective** is also different. Zhao et al. focus on best-arm identification in a confounded transductive linear bandit, whereas our paper focuses on CATE estimation for a target population.
>
> Accordingly, their **theory** centers on IV-based confidence sets and sample complexity, while our theory studies finite-sample error bounds, martingale CLTs, and minimax lower bounds for adaptively collected RCT data.
>
> Thus, the ``similarity`` is mainly at the **application level** (e.g., coupon delivery), while the identification regimes are different. In Zhao et al., the coupon plays the role of an instrument; in our setting, the selected unit receives actual randomized treatment under a limited-budget RCT. We have clarified this comparison in the revision.
>
> >**Q2**: Provide more intuition, supported by theory, on how the algorithm’s behavior changes as the balance between RCT and observational data shifts.
>
> **A2**: The OBS–RCT balance affects **efficiency**, **not validity**. In our framework, causal identification is always carried by the queried RCT samples, while OBS only helps define the representation and guide where to randomize. Hence, changing the OBS/RCT ratio does not invalidate the estimator; it changes how informative each additional RCT query is.
>
> The theory explains the observed trend through the information matrix, $V_B = \sum_{t=1}^B \phi(X_t)\phi(X_t)^\top$: the PEHE bound scales with the integrated leverage, $\mathbb{E}[\phi(X)^\top V_B^{-1}\phi(X)]$, or $\tilde{O}(\sqrt{d/(\kappa B)})$ under $V_B \succeq \kappa B I$. When OBS is scarce or highly biased, support gaps and discrepancy are larger, so random sampling wastes RCT budget on already-covered regions, whereas our active policy preferentially queries high-discrepancy / high-overlap-deficit points, increasing the small eigenvalues of $V_B$ and thus reducing the constant in the error bound. When OBS becomes larger and closer to the target distribution, these blind spots shrink, so the marginal value of active correction naturally decreases and the active-vs-random gap narrows. This is exactly what **Table 4** shows: the gain is largest in $0.2D_{\text{obs}}^{\text{full}}$ and Group Sampling, and smaller when OBS approaches $0.8D_{\text{obs}}^{\text{full}}$.
>
> Put differently, a useful heuristic decomposition is
>
> $$
> \text{Error}
> \approx
> \underbrace{\text{OBS-induced approximation / coverage error}}\_{\downarrow\ \text{as OBS grows}}
> \+\
> \underbrace{\text{RCT estimation error}}\_{\sim \sqrt{d/B}}
> $$
>
> where active learning mainly improves the **second term’s constant** by repairing coverage more efficiently, rather than changing the fundamental $1/\sqrt{B}$ rate.
>
> For ``theoretical contribution``, our point is not to improve the canonical $\sqrt{d/B}$ rate, but to show it remains achievable in a harder hybrid regime: OBS+RCT with adaptive, non-i.i.d. queried experiments, where observational data may be arbitrarily confounded and used only for design, not identification. Prior RCT works study RCT-only settings, while observational active-learning works do not support adaptive experimentation with valid identification. Our theory bridges these two lines.
>
> Technically, classical i.i.d. theory gives $\sqrt{d/B}$-type rates, but not under adaptive $X_t$. We show randomization induces a martingale structure, yielding **(i) finite-sample unbiasedness**, **(ii) a martingale CLT under non-i.i.d. sampling**, and **(iii) a matching minimax lower bound**, so adaptivity improves constants, not the fundamental $\sqrt{d/B}$ rate.
>
> Thus, the contribution is not a faster rate, but showing that the optimal rate is preserved under adaptive causal design, with gains only in constants via the information matrix.
>
> For ``reproducibility issue``, we added synthetic experiments (see response to dear RW ``4dvE``), enabling replication despite the real-world data not being public.
>
> Finally, thank you for pointing out the minor writing issues. We have fixed them in the revised version.

---

> > ### Author Rebuttal · Reviewer_xrZB · 2026-04-02
> >
> > Thank you, my main questions got answered. The current rating stays same.

---

> > > ### Author Response · Authors · 2026-04-04
> > >
> > > # **`Final Response to Reviewer xrZB`**
> > >
> > >
> > >
> > > Dear Reviewer xrZB,
> > >
> > > We would like to express our sincere gratitude for your positive feedback and for your thoughtful, generous, and encouraging comments on our work. Your recognition is deeply appreciated and serves as a great encouragement to us. We are especially pleased and reassured to know that your concerns have been addressed.
> > >
> > > To directly `address the weaknesses` you identified and to further enhance **the reproducibility of the paper** and **the robustness of the results**, we have substantially strengthened and expanded our **experimental evaluation** by including additional experiments on both **real-world** and **synthetic datasets** (see our response to **`Reviewer 4dvE`** in the **`final rebuttal`**). We hope that these additional results provide a more comprehensive empirical assessment and further strengthen the paper.
> > >
> > > - In particular, the experiments on **synthetic datasets** are designed to empirically validate our theoretical results under settings aligned with the assumptions of the theory. We find that active sampling achieves lower **PEHE (*Precision in Estimation of Heterogeneous Effects*)** than random sampling in the majority of cases, and that the improvement is statistically significant at the 5% level in both low-dimensional settings (20 and 50 covariates) and high-dimensional settings (100 and 150 covariates). In addition, our sensitivity analyses with respect to the parameters $\alpha$, $\beta$, and $\gamma$ also yield statistically significant results at the 5% significance
> > >  level. Taken together, these findings provide empirical support for our theoretical claims and show that the advantage of adaptive sampling persists across a broad range of problem dimensions and parameter choices.
> > >
> > > - The **real-world experiments**, in contrast, are intended to demonstrate the **external validity** and **practical generalizability** of our method. We show that our approach extends well to industry-style datasets. Under strongly biased observational (OBS) datasets and under different mixture ratios of business observational (OBS) data, our method achieves significant improvements in **AUUC (*Area Under the Uplift Curve*)** when applied to nonlinear models such as **DRCFR**, **DESCN**, and **DragonNet**. In addition, the ablation studies and sensitivity analyses further confirm the robustness of the proposed method.
> > >
> > > We have also provided a more detailed explanation of our theoretical contributions in the **`final rebuttal`**. As clarified in our response to **`Reviewer dJgq`**, this is exactly the theoretical contribution of the paper: **not *identification from confounded OBS without Assumption 2.4*, but rather *efficient target-population CATE estimation by combining RCT-validity with OBS-guided experimental allocation*.**
> > >
> > > To further supplement this point, our contribution is not that we improve the canonical $\sqrt{d/B}$ rate itself; rather, we show that this rate can still be achieved in a strictly harder hybrid regime: OBS+RCT fusion with adaptive, non-i.i.d. queried experiments, where observational data may be arbitrarily confounded and are used only to guide design, not for identification. Existing sample-constrained RCT works study RCT-only settings, while observational active-learning works study fixed treatment assignments without adaptive experimentation. Our theory closes the gap between these two lines.
> > >
> > > Technically, the comparison point is as follows: classical i.i.d. regression theory gives $\sqrt{d/B}$-type behavior, but does not directly apply when $X_t$ is chosen adaptively from past outcomes. Our **`contribution`** is to prove that, in this adaptive causal design setting, randomization still creates a martingale difference structure, yielding:
> > >
> > > >**(i) *Finite-sample unbiasedness and self-normalized concentration***;
> > >
> > > >**(ii) *A martingale CLT despite non-i.i.d. sampling***;
> > >
> > > >**(iii) *A minimax lower bound showing that even with perfect adaptivity and unlimited OBS logs / unlabeled pools, no method can beat $\sqrt{d/B}$ in general***.
> > >
> > > Thus, theoretical improvement is not a faster exponent, but a sharper characterization of what active experimentation can and cannot improve: it improves the information matrix and constants, while the fundamental rate remains unimprovable.
> > >
> > > Finally, We are sincerely grateful for your valuable suggestions, thoughtful comments, and careful assessment. Your feedback has been immensely helpful in improving not only the presentation of the paper, but also its overall quality and clarity. We truly appreciate the time and effort you devoted to reading our work and providing such constructive guidance.
> > >
> > > \
> > > \
> > > \
> > > \
> > > \
> > > Many thanks,
> > > The authors of #2386

---

### Official Review · Reviewer_4dvE · 2026-03-11

**Soundness:** 2
**Presentation:** 3
**Significance:** 3
**Originality:** 3
**Overall Recommendation:** 4
**Confidence:** 3

**Summary:**

This paper proposes a budgeted active experimentation framework for heterogeneous treatment effect (HTE) estimation that fuses abundant but biased observational data with a small, adaptively collected randomized controlled trial (RCT) dataset. The core design principle is a clean separation of roles: observational data informs where to run experiments (via a multi-criteria acquisition function), while causal identification is carried exclusively by randomized experiments. Theoretically, the authors establish unbiased and finite-sample deviation bounds despite adaptive design selection, asymptotic normality again despite adaptive sampling, and minimax lower bounds. Empirically, the framework is evaluated on a single ride-hailing platform dataset with the proxy metric of AUUC.

**Compliance With Llm Reviewing Policy:**

Affirmed.

**Final Justification:**

The rebuttal has changed my evaluation as authors have substantially addressed my primary empirical concerns. As such I have increased my rating to from to 2 to 4 (Weak accept).

My original review identified insufficient empirical evaluation as the critical weakness. The authors have responded thoroughly: (i) they now report 30 repeated runs with mean ± standard deviation across all three backbones (DRCFR, DESCN, DragonNet), providing statistically meaningful comparisons that were entirely absent from the original submission; (ii) they added synthetic experiments under linear CATE settings with PEHE evaluation, directly validating the theoretical analysis under matching assumptions; (iii) they provide sensitivity analyses for the acquisition weights (α, β, γ) across multiple dimensionalities.

The real-world evaluation still relies on a single dataset and for the ICML venue this may still be seen as a significant drawback for acceptance.

**Key Questions For Authors:**

1.	How does the counterfactual alignment weighting scheme (Eq. 60–62) interact with the theoretical guarantees? Does the gold/silver weighting preserve the unbiasedness and concentration properties established in Section 4, or does it introduce additional bias?

2.	How sensitive are results to the acquisition weights (α, β, γ)? The ablation studies (Table 5) test component subsets but not weight sensitivity. A sensitivity analysis would clarify this.

3.	Can confidence intervals or standard deviations be reported across multiple runs? Given that the theoretical contribution enables uncertainty quantification, it would be natural and important to validate whether the observed AUUC improvements are statistically significant.

4.	Can the authors provide results on semi-synthetic benchmarks or a linear model setting that directly validates the theoretical analysis?

**Limitations:**

Yes

**Strengths And Weaknesses:**

Strengths:

1.	The paper addresses an important challenge of incorporating observational data alongside scarce resource-constrained RCT data, discussing a critical challenge in real-world causal inference. The separation of observational data (for guiding experiment design) from RCT data (for causal identification) is a strong direction to pursue and avoids the fragility of methods that attempt to extract causal signals from confounded observational data directly.

2.	The paper provides comprehensive theoretical analysis in a non-trivial non-iid statistical setting.

3.	Practically relevant acquisition functions. The three-component acquisition score (uncertainty, domain discrepancy, overlap deficit) addresses distinct and recognizable failure modes in OBS-RCT fusion. The use of rank normalisation to combine heterogeneous scores is a pragmatic choice that avoids brittle scale-dependent tuning.

4.	The overall presentation of the paper is strong and well written, particularly given the detailed theoretical analysis.

Weaknesses:

1.	The empirical evaluation is insufficient. This is the paper's critical weakness and the primary reason for my recommendation. The experimental section relies on a single proprietary dataset from a ride-hailing platform. There are no semi-synthetic benchmarks with known ground-truth CATE, no synthetic experiments to verify expected theoretical phenomena, and no experiments on any other real-world domain. The evaluation metric AUUC is a ranking-based proxy that does not correspond to the paper's own theoretical target (PEHE). This means the theoretical guarantees are never directly validated. No confidence intervals, standard deviations, or significance tests are reported across runs, making it impossible to determine whether the observed AUUC improvements are statistically meaningful or within noise. The combination of a single proprietary dataset, no benchmarks allowing direct PEHE evaluation, and no reported variance across runs means the empirical claims are unsubstantiated.

2.	The theoretical analysis assumes linear realisability in the learned representation space (Assumption 4.3), but the experiments use deep neural network CATE learners (DRCFR, DESCN, DragonNet) for which the linear theory does not directly apply. No experiment uses a linear model to verify the theoretical results more closely.

3.	The acquisition weights are presented without justification. The ablation study (Table 5) tests component subsets but not weight sensitivity. If the method is fragile to these choices, practical applicability in new domains is limited.

---

> ### Author Rebuttal · Authors · 2026-03-30
>
> We thank you for your thoughtful review and respond to each point below.
> >**Q1**: Does the counterfactual alignment weighting scheme have theoretical guarantees, and does it introduce bias?
>
> **A1**: Eqs. (60)–(62) are best viewed as an efficiency-oriented reweighting for training, not as the estimating equation in Section 4. The theoretical guarantees there rely on the orthogonalized RCT estimator, where randomization ensures $\mathbb{E}[\tilde{Y}_\{t}\mid X_t ,p_t , \mathcal{F}\_{t-1}] = \tau(X_t)$, so the error forms a bounded martingale difference, yielding unbiasedness and concentration.
>
> In contrast, the gold/silver scheme reweights the training loss to emphasize queried samples that address counterfactual gaps under a finite budget. While aligned with the same intuition (using RCTs to reduce bias), it is not used in the Section 4 proofs.
>
> Formally, inserting arm-dependent weights gives  $\mathbb{E}[w(X,T)\tilde{Y}\mid X=x] = w_1(x)\mu_1(x) - w_0(x)\mu_0(x)$,  which equals $\tau(x)$ only under additional conditions. Thus, gold/silver weighting does not strictly preserve the
> unbiasedness/concentration guarantees, but is designed to improve finite-sample efficiency.
>
> >**Q2, Q4**: Providing synthetic linear-setting results and sensitivity analyses to better validate theory.
>
> **A2**: To align with theoretical analysis, we construct a `synthetic dataset` with an OBS set, an unlabeled pool, and an RCT test set, under feature dimensions $d \in \{20,50\}$. Covariates are sampled from multivariate Gaussians with domain shift:
> $
> X_{\mathrm{obs}} \sim \mathcal{N}(0, \Sigma_{\mathrm{obs}}), \quad
> X_{\mathrm{pool}}, X_{\mathrm{test}} \sim \mathcal{N}(\mu_{\mathrm{pool}}, \Sigma_{\mathrm{pool}}),
> $
> where $(\Sigma\_{\mathrm{obs}})\_{ij}=0.3^{|i-j|}$, $(\Sigma\_{\mathrm{pool}})\_{ij}=0.5^{|i-j|}$, and $\mu\_{\mathrm{pool}}$ has nonzero entries in the first coordinates to induce covariate shift.
>
> We generate linear outcome models with $ \beta_0,\beta_\tau,\beta_e \sim \mathcal{N}(0,I_d), $ and define $ \mu_0(x)=0.5+0.15\frac{x^\top\beta_0}{\max|x^\top\beta_0|}, \quad \tau(x)=0.15\frac{x^\top\beta_\tau}{\max|x^\top\beta_\tau|}, \quad \mu_1(x)=\mu_0(x)+\tau(x), $ with $ Y(t)\sim \mathrm{Bernoulli}(\mu_t(x)).$
>
> OBS treatment is assigned via a biased propensity $ e(x)=\sigma\left(\frac{8 x^\top\beta_e}{\sqrt d}\right),$
> while in both the RCT pool and the test set $ T \sim \mathrm{Bernoulli}(0.5).$ We compare active and random sampling under increasing RCT budgets.
>
> The final CATE estimator is a **linear pseudo-outcome ridge regression:**
> $ \tilde Y=\frac{TY}{p}-\frac{(1-T)Y}{1-p}, \quad
> \hat\tau(x)=x^\top (X^\top X+\lambda I)^{-1}X^\top \tilde Y. $
>
> For `Acquisition scores`：**(i) Uncertainty** $v(x)=x^\top V^{-1}x$, where $V=\lambda I+\sum_i x_i x_i^\top$ is the ridge design matrix built from currently selected RCT samples **(ii) Domain discrepancy** $d(x)=\mathbb{P}(\mathrm{pool}\mid x)=\sigma(w^\top x)$, where $w$ is learned by a logistic classifier **(iii) Overlap deficit** $o(x)=2|\hat e(x)-0.5|$, where $\hat e(x)=\sigma(w_e^\top x)$ is the estimated propensity score fitted on OBS data.
>
> Performance is evaluated by $\sqrt{\mathrm{PEHE}}$ on RCT test set, averaged over 5 runs ：`https://anonymous.4open.science/r/Active-Learning-Supplementary-Experiments-342D/Active_Learning_Rebuttal.pdf`
>
> >**Q3**: Reporting uncertainty measures for AUUC across multiple runs.
>
> **A3**: Due to resource constraints, experiments on the other models are still in progress. Here, we report the `AUUC estimates and standard deviations` for **DRCFR** based on **three repeated runs.** (see `A2 link`).
>
> For `W2: linear–nonlinear mismatch`, Assumption 4.3 treats linearity in the learned representation as an analysis device, not a restriction on the deployed model. The linear head serves as a transparent “microscope” to isolate the challenge of adaptive, budget-limited RCT collection.
>
> More generally, for any bounded function class $\mathcal{H}$ (including neural networks), we define  $\hat{f} \in \arg\min_{f \in \mathcal{H}} \sum_{t=1}^{B} \big(\tilde{Y}_t - f(\phi(X_t))\big)^2$.
>
> By randomization,  $\mathbb{E}[\tilde{Y}\_t \mid X_t, p_t, \mathcal{F}\_{t-1}] = \tau(X_t)$,  so the noise $\varepsilon_t = \tilde{Y}_t - \tau(X_t)$ remains a bounded martingale difference sequence regardless of model class.
>
> This yields the oracle inequality  $\mathcal{E}(\hat{f}) - \inf_{f \in \mathcal{H}} \mathcal{E}(f) \lesssim\ \mathfrak{R}\_B^{\mathrm{seq}}(\mathcal{H}) + \sqrt{\frac{\log(1/\delta)}{B}}$,  $\mathcal{E}(f) := \mathbb{E}_{X} \big[(f(\phi(X)) - \tau(X))^2\big]$. where $\Re_B^{\text {seq }}(\mathcal{H})$ is sequential complexity of $\mathcal{H}$
>
> Thus, replacing the linear head with a neural network does not affect the validity backbone; it only changes the approximation/complexity term. Experiments with DRCFR et al verify that active design extends beyond linear case, showing consistent gains across architectures—i.e., the benefit is backbone-agnostic.

---

> > ### Author Rebuttal · Reviewer_4dvE · 2026-04-02
> >
> > Thank you for the additional experiments. The synthetic PEHE results directly implement the theoretical setup (linear CATE, pseudo-outcome ridge regression) to validate the claims. The weight sensitivity analysis is also a nice addition. The real-world DRCFR error bars confirm statistical significance at low budgets, although 3 runs is insufficient to draw reliable conclusions, especially given that the evaluation relies on a single proprietary dataset; making robust variance estimates all the more important. I raise my score from 2 (reject) to 3 (weak reject), and would consider raising further if the authors provide results across at least 5–10 runs for all backbones (DRCFR, DESCN, DragonNet) in the final version.

---

> > > ### Author Response · Authors · 2026-04-04
> > >
> > > # **`Final Response to Reviewer 4dvE`**
> > >
> > >
> > > Dear Reviewer 4dvE,
> > >
> > > Thank you very much for your constructive follow-up comments and for your careful reassessment of our paper.
> > >
> > > To further strengthen the reliability and robustness of our empirical evaluation, we have increased **`the number of repeated runs to 30`** and now report all results as **`mean ± standard deviation`**, which provides a more stable and statistically informative assessment. These additional experiments cover both the **real-world** and **synthetic datasets** and are designed to more comprehensively address your concerns regarding effectiveness, robustness, and generalizability.
> > >
> > > More importantly, we have reorganized and expanded the experiments so that the role of each part is clearer.
> > >
> > > - For the **synthetic datasets**, the experiments are designed to empirically validate our theoretical results under settings aligned with the assumptions of the theory. Under fixed-weight settings with **20, 50, 100, and 150 covariates**, active sampling generally achieves lower **PEHE (Precision in Estimation of Heterogeneous Effects)** than random sampling in the majority of settings, with statistically significant improvements at the **5% significance level** observed in both low-dimensional and high-dimensional regimes. In addition, we further extend the **sensitivity analyses** for $\alpha$, $\beta$, and $\gamma$ to the 100-dimensional and 150-dimensional cases. The results further show that the advantage of active sampling becomes more evident when the sample budget is limited, while remaining robust across different parameter settings. Taken together, these results provide stronger empirical support for our theoretical claims and show that the benefit of active sampling persists across a broad range of problem dimensions and parameter choices.
> > >
> > > - For the **real-world datasets**, by contrast, the experiments are intended to demonstrate the **external validity** and **practical generalizability** of our method. Specifically, on the strongly biased OBS dataset $\mathcal{D}^{\text{bias}}\_{\text{obs}}$, active sampling improves AUUC over random sampling across multiple industry-style backbones, including **DRCFR, DESCN, and DragonNet**, with the gains becoming more pronounced when the RCT budget is limited. Meanwhile, we further supplement the evaluation with experiments under **different mixing ratios of the business OBS dataset** $\mathcal{D}^{\mathrm{full}}\_{\mathrm{obs}}$ and the results show that the advantage of active sampling remains robust across different OBS data scales. To further analyze the contribution of each component in the acquisition strategy, we also include more detailed **ablation studies**. The results show that although individual score components or pairwise combinations can already outperform random sampling at certain budgets, the full acquisition score remains the best overall, which further supports the design of our complete acquisition objective. Finally, the **sensitivity analyses** with respect to $(\alpha, \beta, \gamma)$ as well as $(w_{\text{gold}}, w_{\text{silver}})$ further confirm the robustness of the proposed method.
> > >
> > > Specifically, for the **real-world datasets**, we now include:
> > >
> > > > **(i) Active vs random sampling comparisons** on the strongly biased OBS dataset $\mathcal{D}^{\text{bias}}\_{\text{obs}}$, across all backbones, including **DRCFR, DESCN, and DragonNet**;
> > >
> > > > **(ii) Experiments under different mixing ratios** of the business OBS dataset $\mathcal{D}^{\mathrm{full}}\_{\mathrm{obs}}$, to evaluate robustness under different observational data scales;
> > >
> > > > **(iii) Ablation studies** on different score components and their combinations, to better understand the contribution of each part of the acquisition strategy;
> > >
> > > > **(iv) Sensitivity analyses** for the parameters $(\alpha, \beta, \gamma)$, as well as $(w_{\text{gold}}, w_{\text{silver}})$, to examine the robustness of the method under different parameter settings.
> > >
> > > For the **synthetic datasets**, we now include:
> > >
> > > > **(i) Fixed-weight experiments** under both **low-dimensional** and **high-dimensional** covariate settings, specifically **20D, 50D, 100D, and 150D**;
> > >
> > > > **(ii) Sensitivity analyses** for **$(\alpha, \beta, \gamma)$** across **20D, 50D, 100D, and 150D** covariate settings.
> > >
> > > To provide a complete view of these additional evaluations, we include all corresponding figures and tables for both the **real-world** and **synthetic** experiments in the supplementary material at the following **`updated link`**:
> > > **<https://anonymous.4open.science/r/Active-Learning-Supplementary-Experiments-342D/Active_Learning_Rebuttal.pdf>**.
> > >
> > > We sincerely thank you again for your valuable suggestions. Your comments have helped us improve the completeness and rigor of our experimental evaluation, and we hope that these additional experiments fully address your concerns and further strengthen the empirical support for our claims.
> > >
> > > Many thanks,
> > >
> > > The authors of #2386

---

### Official Review · Reviewer_dJgq · 2026-03-12

**Soundness:** 2
**Presentation:** 1
**Significance:** 2
**Originality:** 2
**Overall Recommendation:** 2
**Confidence:** 4

**Summary:**

The paper studies heterogeneous treatment effect estimation. Particularly, it identifies limited sample sizes in RCTs and near-determinism/non-randomization in Observational studies as obstacles for estimating it, and proposes and active experimentation strategy informed by observational data to collect experimental data for addressing it.

**Compliance With Llm Reviewing Policy:**

Affirmed.

**Final Justification:**

Some of my main concerns remain. Among them, the most important one is the precise characterization of the covariate region where this methods can provide meaningful guarantees beyond methods in the literature. Also, writing-related issues should be seriously addressed by the authors.

**Key Questions For Authors:**

- How and why is Assumption 2.4 used? Which theoretical results are valid with and without this assumption? What is the key difference in settings where this assumption is needed versus not?
- Why is a target covariate distribution is defined/needed, under the outcome invariance assumption.
- Please see the last question regarding the main contribution and role of Assumption 2.4 in strengths and weaknesses above.

**Limitations:**

Limitations are not discussed. Given that the paper tackles a problem where many assumptions and trade-offs are often made for meaningful results, a more dedicated limitations discussion would be beneficial. Particularly at the current form of the paper, since there is a fair amount of ambiguity in assumptions and how strict are they.

**Strengths And Weaknesses:**

- Writing should be organized to make important definitions easier to find (e.g. propensity scores, ${\cal C}_k$, $p_k$, etc.).
- Assumption 2.x are not referred to from any theorem statements, and are confusing w.r.t. what the contribution is. It is repeatedly stated throughout the paper that OBS can be arbitrarily confounded, which is contrasted by Assumption 2.4. But Assumption 2.4 does not seem to be used at all.
- A closely related work which use RCT-OBS fusion for CATE estimation without this assumption, for instance, is missing [1]. A comparison to clarify how this work differs, and particularly how is that additional expensive assumption is used would be helpful. There are also a few other related missing references [3,4,5].
- The language is a bit off sometimes. For instance, after Theorem 4.5, it reads
> Theorem 4.5 quantifies a clean “budget ⇒ error” trade- off under adaptive experimentation: your estimation error is controlled by an information matrix...

which in a bizarre way reads like a response from a language model. I.e., who is "you" in this sentence? While it is OK to use those tools, the final text should be polished to be targeted toward the reader. Same thing also causes poorly justified leaps between claims/arguments throughout.

- The existence of a target marginal is confusing, as the goal is not ATE estimation but CATE, which is already conditioned on $X$, and authors seem to assume outcome invariance anyways?

- The main contribution of the paper is overall ambiguous. My key concern, which I could not resolve while reading the paper and I need authors help with is the following: If you are not willing to assume 2.4, but still estimate CATE with RCT-OBS fusion, that would be limited to overlap region, and is studied in [1], [2]. How is this work different, and is a comparison to those methods possible? If you mainly want to use RCT to collect data where observational data is lacking (e.g. as you mention in lines 230-240), than you'd have to assume 2.4 to be able to estimate the CATE in regions where there is plenty of observational data, which makes the contribution  much less interesting.

[1] I. Demirel, A. Alaa, A. Philippakis, and D. Sontag. Prediction-powered generalization of causal inferences. International Conference on Machine Learning, 2024.

[2] T. Hatt, J. Berrevoets, A. Curth, S. Feuerriegel, and M. van der Schaar. Combining observational and randomized data for estimating heterogeneous treatment effects. arXiv preprint arXiv:2202.12891, 2022.

[3] S. Yang and P. Ding, Combining multiple observational data sources to estimate causal effects. JASA, 2019.

[4] P. De Bartolomeis, J. Abad, G. Wang, K. Donhauser, R. M. Duch, F. Yang, and I. J. Dahabreh. Efficient randomized experiments using foundation models. Advances in Neural Information Processing Systems (NeurIPS), 2025.

[5] E. Rosenman and A. B. Owen. Designing experiments informed by observational studies. Journal of Causal Inference, 9(1):147–171, 2021.

---

> ### Author Rebuttal · Authors · 2026-03-30
>
> ### `A Kind General Response to All Reviewers and Dear Ac`
>
> We sincerely thank the reviewers and the AC for their thoughtful feedback and constructive suggestions. We have revised our paper to address the main concerns as follows:
>
> 1. **Clarification of Theoretical Contributions:**
>     - Clarifying the role of Assumption 2.4 and the need for a target covariate distribution (`dJgq-Q1, Q2`).
>    - Differentiating our contributions from prior RCT–OBS fusion and adaptive experimentation literature (`dJgq-Q3`, `xrZB-Q1`).
>    - Providing intuitive explanations and theoretical support for the empirical results. (`xrZB-Q2`).
>
> 2. **Expanded Empirical Validation:**
>    - Explaining the role of counterfactual alignment weighting scheme (`4dvE-Q1`).
>    - Adding AUUC standard deviations on the real-world datasets (`4dvE-Q3`).
>    - Adding synthetic linear-setting experiments together with sensitivity analyses (`4dvE-Q2, Q4`).
>
> 3. **Improved Clarity and Presentation:**
>    - Reorganizing key definitions to improve accessibility (`dJgq`, `xrZB`).
>    - Refining language and exposition to avoid ambiguity and improve readability throughout (`dJgq`).
>
> Below we respond point by point.
>
> ----
> ## Response to RW `dJgq`
>
> Thank you for reviewing our paper and for your comments. Below, we respond to each of your concerns point by point.
> >**Q1**：How and why is Assumption 2.4 used? Which theoretical results are valid with and without this assumption? What is the key difference in settings where this assumption is needed versus not?
>
> **A1**: The key `distinction` is therefore simple: if OBS is used only to **guide design**, Assumption 2.4 is unnecessary; if OBS is used to **identify or correct causal effects directly**, then Assumption 2.4 is required.
>
> The main guarantees—unbiased pseudo-outcomes, finite-sample deviation bounds, PEHE bounds, asymptotic normality, and the minimax lower bound—remain valid even when the observational assignment is arbitrarily confounded. This robustness to arbitrary OBS confounding is one of the main advantages of our framework.
>
> If Assumption 2.4 fails, what may be affected is efficiency rather than validity: OBS may become less informative for warm-starting or acquisition, so more RCT budget may be needed to achieve the same accuracy. This is also consistent with our empirical findings that the method is especially useful in highly biased or data-scarce observational regimes. By contrast, direct OBS-based causal correction would no longer be justified without Assumption 2.4.
>
> >**Q2**：Why is a target covariate distribution is defined/needed, under the outcome invariance assumption.
>
> **A2**: Because outcome invariance only says the conditional causal effect ( $\tau(x)$ ) is shared across sources, while the target covariate distribution ($P_X$) is still needed to define which population we want to generalize to and optimize over—i.e., the PEHE risk and active design objective are evaluated on ($X\sim P_X$), not on the observational marginal ($P_{X^{obs}}$), which may differ substantially under selection bias.
>
> >**Q3**: Comparison of methods in References [1]–[5].
>
> **A3**: Thank you for pointing out these relevant references. We agree that the distinction between our paper and this line of work should be stated more explicitly.
>
> `Demirel et al.` [1] is related in spirit, but studies trial-to-target generalization from a fixed trial and targets population-level effects rather than granular CATE guarantees. By contrast, our paper studies budgeted active acquisition for CATE estimation: observational data are used to decide which target units should be randomized next under a limited RCT budget. `Hatt et al.` [2] is also closely related, but our setup additionally assumes a queryable target pool and budgeted randomization of a selected subset of target units. This is precisely how the additional assumption is used in our paper: OBS guides where to randomize, and the queried RCT samples provide the causal information for estimation. Our algorithm and theory are built around this adaptively collected RCT design. `The remaining references` emphasize different regimes: [3] combines multiple observational sources with additional confounder information, [4] improves the efficiency of randomized experiments, and [5] uses observational data to design a stratified binary-outcome experiment. To the best of our reading, none of [1]-[5] studies the same combination of OBS-guided active experiment design, queryable randomized acquisition, and CATE estimation under an adaptive RCT budget that is central to our paper.
>
> Lastly, we sincerely thank you for highlighting these writing-related issues. In the revision, we have reorganized the presentation of key definitions to make them easier to follow and reference, and refined the wording throughout the manuscript to improve clarity and readability. We hope these revisions have addressed your concerns, and we would be happy to provide any further clarification if needed.

---

> > ### Author Rebuttal · Reviewer_dJgq · 2026-04-02
> >
> > > A1: The key distinction is therefore simple:
> >
> > I was not able to understand what argument this sentence is following from or building on top of.
> >
> > One of my key concerns in my original review was the following:
> >
> > > : If you are not willing to assume 2.4, but still estimate CATE with RCT-OBS fusion, that would be limited to overlap region, and is studied in [1], [2]... If you mainly want to use RCT to collect data where observational data is lacking..., than you'd have to assume 2.4 to be able to estimate the CATE in regions where there is plenty of observational data, which makes the contribution much less interesting.
> >
> > I was not able to get more clarity on this point after reading the response. I strongly believe that the covariate region where you can provide guarantees for (be it PEHE, ATE, CATE, etc) is a crucial determinant of the contribution of the paper, and should be addressed accordingly.
> >
> > I thank the authors for their rebuttal. I choose to maintain my score at this time as I think the additional clarifications in the rebuttal and necessary improvements are major enough to warrant another round of reviews.

---

> > > ### Author Response · Authors · 2026-04-04
> > >
> > > Thank you so much for this thoughtful and constructive suggestion! We are carefully and fully implementing your recommendation these days.
> > >
> > > ### Response
> > >
> > > The above concern raises a real dilemma, and we now state it explicitly: if Assumption 2.4 is not imposed, then OBS data cannot be used for causal identification, so valid CATE guarantees can only come from randomized support; but if OBS is used to identify CATE beyond that support, then one must effectively re-introduce 2.4, which would make the contribution much less interesting.
> > >
> > > ``Our framework resolves this by separating identification from design``. Without Assumption 2.4, OBS is not used to identify CATE; it is used only as side information for representation learning, warm start, and—most importantly—to guide where scarce RCT samples should be collected. The actual identification and calibration remain anchored in the queried randomized data. Hence, we do not claim that arbitrarily confounded OBS identifies CATE outside randomized support. The precise contribution is instead a budgeted active design framework: even when OBS is not causally trustworthy, it can still be used safely to make limited randomization substantially more informative for the target population.
> > >
> > > **This is exactly the theoretical contribution of the paper: not “identification from confounded OBS without 2.4,” but “efficient target-population CATE estimation by combining RCT-validity with OBS-guided experimental allocation.”**

---

### Decision · Program_Chairs · 2026-04-30

**Decision:**

Accept (regular)

**Comment:**

After carefully reading the paper, I agree with the authors clarification that the contribution of the submission is indeed novel and tackles a different problem than existing works. While all reviewers found various parts of the original submission unclear, the authors managed to address many of them quite well in the rebuttal leading to favorable later assessments. The initial presentation still leaves key aspects a bit unclear and I strongly encourage the authors to take the feedback into account and make sure the clarifications and additional results from the rebuttal make it into the final paper. Because the setting is relevant and the proposed methodology in my view is interesting and sound, I weakly recommend acceptance.